# DHX9 maintains epithelial homeostasis by restraining R-loop-mediated genomic instability in intestinal stem cells

Xingxing Ren [1,2,3], Qiuyuan Liu[4], Peirong Zhou[3], Tingyue Zhou[2], Decai Wang[2], Qiao Mei[4], Richard A. Flavell [5,6], Zhanju Liu [7] ✉, Mingsong Li [3] ✉, Wen Pan [1,2] ✉ & Shu Zhu [1,2,8] ✉

Epithelial barrier dysfunction and crypt destruction are hallmarks of inflammatory bowel disease (IBD). Intestinal stem cells (ISCs) residing in the crypts play a crucial role in the continuous self-renewal and rapid recovery of intestinal epithelial cells (IECs). However, how ISCs are dysregulated in IBD remains poorly understood. Here, we observe reduced DHX9 protein levels in IBD patients, and mice with conditional DHX9 depletion in the intestinal epithelium ($Dhx9^{\Delta IEC}$) exhibit an increased susceptibility to experimental colitis. Notably, $Dhx9^{\Delta IEC}$ mice display a significant reduction in the numbers of ISCs and Paneth cells. Further investigation using ISC-specific or Paneth cell-specific $Dhx9$-deficient mice demonstrates the involvement of ISC-expressed DHX9 in maintaining epithelial homeostasis. Mechanistically, DHX9 deficiency leads to abnormal R-loop accumulation, resulting in genomic instability and the cGAS-STING-mediated inflammatory response, which together impair ISC function and contribute to the pathogenesis of IBD. Collectively, our findings highlight R-loop-mediated genomic instability in ISCs as a risk factor in IBD.

Inflammatory bowel disease (IBD) is prototypical complex intestinal disease, comprises both Crohn's disease (CD) and ulcerative colitis (UC), which are chronic, immunological inflammatory disorders of the gastrointestinal tract. Immune system dysfunction, microbiome imbalances, and epithelial barrier impairments are typical pathophysiological features of IBD[1]. Extensive research has revealed the pivotal involvement of immune cells in the pathogenesis IBD. Although immune-targeting therapies have made significant advancements, their clinical effectiveness is limited due to inconsistent efficacy, side effects, and drug resistance[2]. The attainment of "mucosal healing" has emerged as the standard for achieving long-term remission in IBD[3]. Therefore, a deeper understanding of the etiology of IBD, particularly the regulatory mechanisms governing epithelial function, is crucial for advancing our knowledge of its pathogenesis.

The intestinal epithelial cell (IEC) layer maintains a physical barrier that separates the host from food antigens, commensal microbes, and other cellular insults. Simultaneously, IEC contact with the enteral environment and the immune system of the host, enabling intestinal

[1]Hefei National Research Center for Physical Sciences at the Microscale, School of Basic Medical Sciences, Division of Life Sciences and Medicine, University of Science and Technology of China, 230001 Hefei, China. [2]Key Laboratory of immune response and immunotherapy, Center for Advanced Interdisciplinary Science and Biomedicine of IHM, School of Basic Medical Sciences, Division of Life Sciences and Medicine, University of Science and Technology of China, Hefei, China. [3]Department of Gastroenterology, Third Affiliated Hospital of Guangzhou Medical University, 510145 Guangzhou, China. [4]Department of Gastroenterology, The First Affiliated Hospital of Anhui Medical University, Hefei 230022, China. [5]Department of Immunobiology, Yale University School of Medicine, New Haven, Connecticut, USA. [6]Howard Hughes Medical Institute, Yale University School of Medicine, New Haven, Connecticut, USA. [7]Center for IBD Research, Department of Gastroenterology, Shanghai Tenth People's Hospital, Tongji University School of Medicine, Shanghai 200072, China. [8]School of Data Science, University of Science and Technology of China, Hefei 230026, China. ✉e-mail: zhanjuliu@tongji.edu.cn; lims661216@163.com; wenpan@ustc.edu.cn; zhushu@ustc.edu.cn

homeostasis[4]. To confront the continuous changes in the surrounding environment, IECs rapidly self-renew with a turnover time of 3–5 days. Such rapid renewal is supported by intestinal stem cells (ISCs), which are positioned at the base of invaginated crypt[5]. ISCs differentiate into progenitor cells, which further proliferate and differentiate into specialized IEC lineages with distinct functions, including absorptive enterocytes, mucus-secreting goblet cells, hormone-producing enteroendocrine cells (EECs), chemosensory tuft cells, and antimicrobial peptide-secreting Paneth cells. The diversity and proper function of these specific IEC subsets are vital for maintaining intestinal homeostasis[6]. Dysregulation of the differentiation system responsible for correct IEC formation plays a crucial role in IBD pathogenesis[7,8]. Indeed, several critical genes essential for the differentiation of IECs have been shown to display aberrant expression patterns during IBD[9]. The regulation of ISC differentiation is precisely controlled by external signals that emanate from the niche within the crypts. Multiple cell types provide cellular signals that support ISC function and guarantee adequate ISC turnover, which enables ISCs to differentiate into a healthy epithelial barrier[5]. Moreover, abnormalities in certain intracellular signals or genes can also affect the activity and function of ISCs[10–12]. Therefore, both genetic and environmental factors play critical roles in the regulation of ISC homeostasis.

DEAH-box helicase 9 (DHX9) is an important member of the DEAH-box RNA helicase family. As an ATP-dependent RNA helicase, DHX9 unwinds RNA or DNA duplexes, as well as DNA:RNA hybrids (R-loops), making it an important protein for DNA replication and transcription[13]. DHX9 has been implicated in several cellular processes, including transcription, translation, mRNA decay, and RNA transport[14]. Furthermore, DHX9 has emerged as a significant player in tumorigenesis[15], infection[16], and autoimmune diseases[17]. Notably, our previous research has shed light on the cooperative relationship between DHX9 and Nlrp9b in IECs, where they collaborate to detect rotavirus double-stranded RNA (dsRNA), triggering inflammasome complex formation. This interaction facilitates the maturation of interleukin-18 and GSDMD-induced pyroptosis[18]. Nevertheless, the role and mechanisms of DHX9 in maintaining intestinal homeostasis remain largely unexplored.

In this study, we discover that DHX9 protein levels are significantly reduced in patients with IBD. By employing IEC-specific (*Dhx9*^ΔIEC), Paneth cell-specific (*Dhx9*^ΔPaneth), and ISC-specific (*Dhx9*^iΔISC) *Dhx9*-deficient mice and organoids, we demonstrate that DHX9 plays a pivotal role in epithelial homeostasis by regulating ISC function. Specifically, DHX9 deficiency causes abnormal R-loop formation, leading to increased genomic instability and cGAS-STING pathway activation. This abnormal immune response and DNA damage together result in functional impairment of ISCs and IBD development. Our findings suggest that DHX9 serves as a gatekeeper for maintaining intestinal homeostasis, and further investigation of the dysregulation of DHX9 may contribute to the development of promising therapies for IBD.

## Results

### Epithelial DHX9 deficiency aggravates DSS- or TNBS-induced colitis

Given the potential significance of DHX9 in IBD pathogenesis, we measured DHX9 protein levels in biopsy specimens from patients with UC and CD. Our results indicate that the expression of DHX9 was decreased in both UC and CD patients compared with healthy controls (Fig. 1a, b). Consistently, immunohistochemistry (IHC) results also revealed a significant reduction in DHX9 protein levels in the epithelial cells of UC and CD patients (Supplementary Fig. 1a). However, there was no significant change in *DHX9* mRNA levels (Supplementary Fig. 1b), indicating that the reduction in DHX9 occurs at the protein level. Notably, we observed distinct degradation bands for DHX9, particularly in UC samples (Fig. 1a, b), suggesting that protein cleavage might play a role in the degradation of DHX9. Previous studies have

highlighted that Caspase-7 is specifically highly expressed in IECs[19], with increased levels observed in patients with IBD[20]. Our recent research also demonstrated that Caspase-3 and Caspase-7 were activated in IECs and have functions beyond apoptosis induction in the intestine[21]. Therefore, we investigated whether caspase family proteins mediate the cleavage and degradation of DHX9. Co-expression of Flag-tagged DHX9 with Caspase-1, Caspase-3, and Caspase-7 respectively in 293 T cells revealed a noticeable cleavage band specifically when Caspase-7 was co-expressed with Flag-DHX9 (Supplementary Fig. 1c). Subsequently, we incubated purified Caspase-3 and Caspase-7 proteins with Flag-DHX9, and found that Flag-DHX9 showed significant cleavage bands upon incubation with Caspase-7 (Supplementary Fig. 1d). Moreover, both mouse DHX9 and human DHX9 exhibited increased cleavage over time when incubated with Caspase-7 (Supplementary Fig. 1e, f). Therefore, Caspase-7-mediated cleavage may be the major cause of DHX9 cleavage and subsequent degradation.

To ascertain the specific function of DHX9 in the maintenance of the intestinal epithelium, we generated IEC-specific *Dhx9*-deficient mice (*Dhx9*^fl/fl Villin-Cre, hereafter called *Dhx9*^ΔIEC). The knockout efficiency was confirmed by measuring DHX9 mRNA and protein expression (Supplementary Fig. 2a, b, c). *Dhx9*^ΔIEC mice did not show changes in body weight or colon length compared with their *Dhx9*^fl/fl littermates in steady-state conditions (Supplementary Fig. 2d, e). Subsequently, we evaluated the effect of DHX9 deficiency in colitis using the murine model of DSS- and TNBS-induced colitis, which mimic the features of UC and CD respectively[22]. In the DSS-induced acute colitis model (Fig. 1c), colitis in *Dhx9*^ΔIEC mice was aggravated with rapid loss of body weight (Fig. 1d) and shorter colon lengths (Fig. 1e) compared with their *Dhx9*^fl/fl littermates. Histopathological examination revealed that *Dhx9*^ΔIEC mice showed substantially increased extent of inflammatory cell infiltration and crypt damage (Fig. 1f). Consistently, epithelial DHX9 deficiency also exacerbated disease severity in the TNBS-induced colitis model. During TNBS-induced colitis (Fig. 1g), *Dhx9*^ΔIEC mice lost significantly more body weight (Fig. 1h), increased rectal bleeding, and shortened colon length compared with their littermate controls (Fig. 1i). In addition, the mortality of *Dhx9*^ΔIEC mice was significantly increased in the TNBS model (Fig. 1j). The histopathological analysis showed that epithelial DHX9 deficiency led to increased inflammatory cell infiltration with more severe disruption of the mucosal epithelium (Fig. 1k). Collectively, the results demonstrate that loss of DHX9 in IECs renders mice highly susceptible to intestinal damage, implying a critical role for DHX9 in the maintenance of intestinal homeostasis.

### Ablation of DHX9 leads to a decrease in the numbers of ISCs and intestinal secretory cells

To understand the underlying mechanisms by which DHX9 controls intestinal epithelial biology, we performed bulk RNA sequencing (RNA-seq) of IECs isolated from 2-month-old *Dhx9*^ΔIEC mice and their *Dhx9*^fl/fl littermates. RNA-seq revealed that IEC-specific DHX9 deficiency resulted in substantial alterations in gene expression, including upregulation of 391 genes and downregulation of 384 genes. Gene Ontology analysis revealed that the most highly enriched down-regulated pathways in *Dhx9*^ΔIEC IECs were associated with antimicrobial peptides secretion and epithelial cells migration. such as "Defense response to bacterium", "Ameboidal-type cell migration", "Epithelial cell migration" (Fig. 2a). Notably, many of the top down-regulated genes are markers of ISCs and intestinal secretory cell lineages, including *Lgr5*, *Olfm4*, *Slc12a2* (ISCs), *Ang4*, *Lyz1*, *Defa3*, *Defa5*, *Defa17*, *Defa20*, *Defa26* (Paneth cells), *Itln1*, *Muc2*, *Ccl9* (goblet cells), *Chga*, *Neurog3* (EECs), *Dclk1*, *Sox9*, and *Trpm5* (tuft cells) (Fig. 2b, c). To systematically investigate the effects of DHX9 deficiency on different epithelial cell lineages, we performed a lineage-specific gene set enrichment (GSEA) analysis of the RNA-seq data. We found that DHX9 deficiency caused a marked reduction in the normalized enrichment scores (NESs) for the

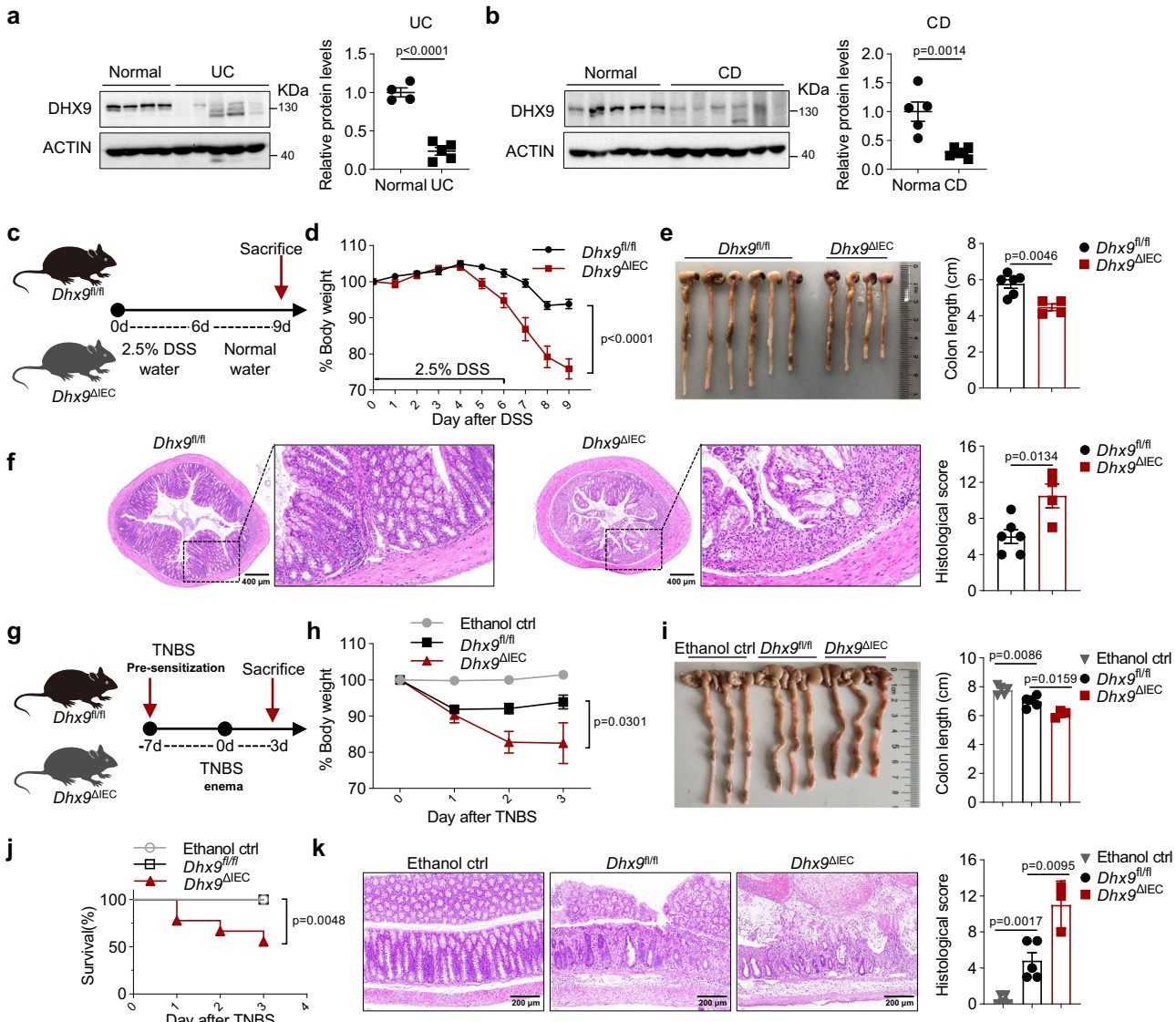

**Fig. 1 | Epithelial DHX9 deficiency aggravates DSS- or TNBS-induced colitis.**
**a** DHX9 protein expression in normal intestinal tissues ($n = 4$) and biopsies from patients with ulcerative colitis (UC) ($n = 5$). Quantifications are presented on the right. **b** DHX9 protein expression in normal intestinal tissues ($n = 5$) and biopsies from patients with Crohn's disease (CD) ($n = 6$). Quantifications are presented on the right. **c-f** Experimental design depicting the induction of colitis using DSS, $Dhx9^{fl/fl}$ ($n = 6$) and $Dhx9^{\Delta IEC}$ ($n = 4$) mice were treated with 2.5% DSS for 6 days (**c**), with monitoring of body weight changes throughout the experimental period (**d**); measurement of colon length upon sacrifice on day 9 (**e**); and representative hematoxylin and eosin (H&E) staining of colonic tissue, Scale bar, 400 μm, histological scores are presented on the right (**f**). **g** Schematic representation of TNBS-induced colitis in $Dhx9^{fl/fl}$ and $Dhx9^{\Delta IEC}$ mice ($n = 5$ per group). **h** Body weight

changes in $Dhx9^{fl/fl}$ and $Dhx9^{\Delta IEC}$ mice following TNBS treatment ($n = 5$ per group). **i** Measurement of colon length upon sacrifice on day 3, with quantifications provided on the right ($Dhx9^{fl/fl}$, $n = 5$; $Dhx9^{\Delta IEC}$, $n = 3$). **j** Survival rates of mice after TNBS treatment (two times results were summarized for statistics, $n = 10$ per group). **k** Representative H&E staining of colon sections. Scale bar, 200 μm. Corresponding histological scores are presented on the right ($Dhx9^{fl/fl}$, $n = 5$; $Dhx9^{\Delta IEC}$, $n = 3$). Results are representative of data generated in at least two independent experiments, error bars show means ± SEM. Statistical analyses of body weight changes were performed using two-way ANOVA analysis with Tukey's multiple comparisons (**d** and **h**). Survival curves was determined by the log-rank test (**j**). Two-tailed unpaired Student's $t$-test were used for other analyses. Source data are provided as a Source Data file.

ISC and Paneth cell associated gene sets. Moreover, the goblet cell, and EEC related gene sets also showed a trend toward reduction (Fig. 2d). We subsequently validated the transcriptome data of intestinal secretory cell lineage-related genes by RT-qPCR. Consistent with the RNA-seq results, *Lgr5*, *Olfm4* (ISCs), *Defa22*, *Ang4* (Paneth cells), *Chga* (EECs), and *Muc2* (goblet cells) expression was significantly reduced in *Dhx9*-deficient IECs (Supplementary Fig. 3). Overall, DHX9 deficiency affected epithelial secretory cell lineages, especially ISCs and Paneth cells.

To further investigate the impact of DHX9 deficiency on different epithelial cell types, we conducted single-cell RNA sequencing (scRNA-seq) on small intestine cells isolated from 2-month-old $Dhx9^{\Delta IEC}$ mice

and their $Dhx9^{fl/fl}$ littermates. This analysis encompassed 30252 cells, with 12108 from the $Dhx9^{fl/fl}$ group and 18144 of $Dhx9^{\Delta IEC}$ group. Unsupervised clustering revealed 15 distinct clusters, each distinguished by unique gene expression patterns (Fig. 2e). A comparative analysis between the $Dhx9^{fl/fl}$ and $Dhx9^{\Delta IEC}$ group cells highlighted a notable difference in cellular composition. Specifically, in the $Dhx9^{\Delta IEC}$ group, there was a significant decrease in the proportion of ISCs, Paneth cells, goblet cells, and EECs, along with an expansion of transit-amplifying (TA) cells (Fig. 2f). These observations are consistent with our RNA-seq data and imply that DHX9 deficiency impacts epithelial secretory cell lineages. Subsequent differential expression analysis across these epithelial subpopulations, including ISCs, Paneth cells, TA

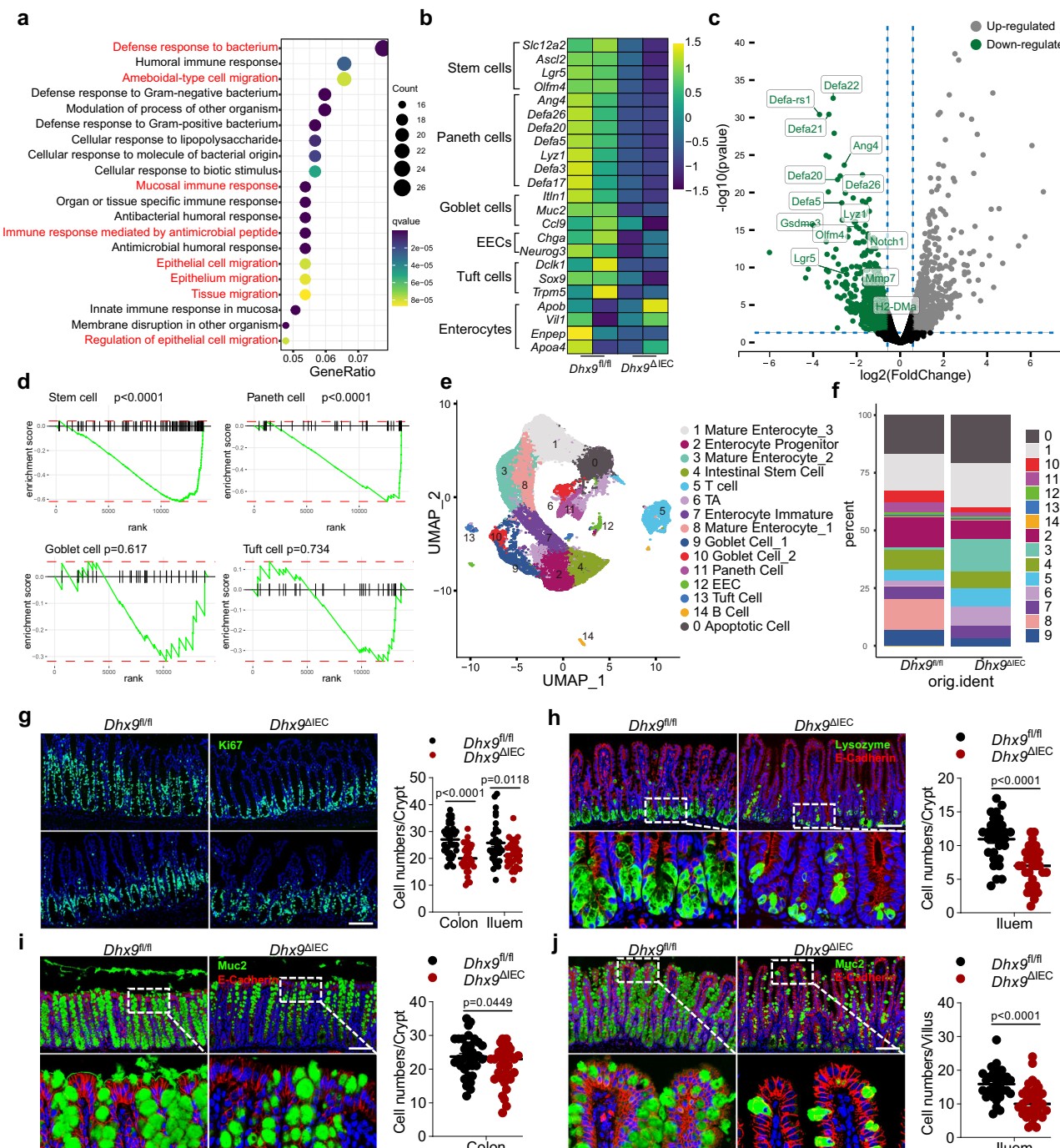

**Fig. 2 | Ablation of DHX9 results in aberrant ISCs and intestinal secretory cells.**
**a** Gene ontology analysis revealing the down-regulated expressed genes in IECs of *Dhx9*^ΔIEC^ mice compared to *Dhx9*^fl/fl^ mice (*P* < 0.05; right-tailed Fischer's exact *t*-test). **b** Heat map analysis of the RNA-seq dataset displaying marker genes for intestinal epithelial cell lines in *Dhx9*^ΔIEC^ mice versus *Dhx9*^fl/fl^ mice. **c** Volcano plot analysis of the RNA-seq dataset illustrating the top upregulated and downregulated genes in IECs of *Dhx9*^ΔIEC^ mice compared to *Dhx9*^fl/fl^ mice (*P* < 0.05; right-tailed Fischer's exact *t*-test). **d** Gene set enrichment analysis (GSEA) comparing the gene expression profiles of intestinal cells based on RNA-seq data in *Dhx9*^ΔIEC^ compared to *Dhx9*^fl/fl^ IECs. Two-tailed *t*-test with the Benjamini–Hochberg correction for an adjusted *P*-value. **e** Single-cell RNA sequencing of IECs from *Dhx9*^ΔIEC^ and their *Dhx9*^fl/fl^ littermates. Visualized as a Uniform Manifold Approximation and Projection (UMAP) plot, categorizing 30,252 cells from the small intestine into 15 distinct clusters based on their gene expression profiles. **f** Graphical representation of the

proportion of each identified cell cluster relative to the total population in the analyzed samples. **g–j** Littermate 8 week-old male *Dhx9*^fl/fl^ and *Dhx9*^ΔIEC^ mice were subjected to immunofluorescence staining. Representative staining of Ki67 in colon (upper panel) and ileum (lower panel) sections (at least 30 crypt-villus axes were counted, colon of *Dhx9*^fl/fl^, *n* = 32; colon of *Dhx9*^ΔIEC^, *n* = 31; ileum of *Dhx9*^fl/fl^, *n* = 33; ileum of *Dhx9*^ΔIEC^, *n* = 31) (**g**); Lysozyme and E-cadherin in ileum sections (at least 30 crypt-villus axes were counted, *Dhx9*^fl/fl^, *n* = 30; *Dhx9*^ΔIEC^, *n* = 34) (**h**); Muc2 and E-cadherin in colon sections (at least 30 crypt axes were counted, *Dhx9*^fl/fl^, *n* = 35; *Dhx9*^ΔIEC^, *n* = 32) (**i**); Muc2 and E-cadherin in ileum sections (at least 30 crypt-villus axes were counted, *Dhx9*^fl/fl^, *n* = 31; *Dhx9*^ΔIEC^, *n* = 32) (**j**). Scale bars represent 100 μm. Each experiment was performed on *n* = 3 mice individually, with similar results. Error bars show means ± SEM. Statistical analyses were performed by two-tailed unpaired Student's *t*-test. Source data are provided as a Source Data file.

cells, tuft cells, and goblet cells, was conducted. Gene Ontology analysis indicated that the most significantly up-regulated pathways in $Dhx9^{\Delta IEC}$ IECs subpopulations were related to antigen presentation, response to interferon, and immune response. In contrast, the primary downregulated pathways were related to ATP metabolic processes, response to unfolded proteins, and mRNA processing (Supplementary Fig. 4). These patterns suggest that DHX9 deficiency disrupts cellular metabolism and essential cellular functions, which may account for the reduced proliferation of these cell populations.

Next, we performed immunofluorescence (IF) staining for markers of epithelial secretory cell lineages to determine whether DHX9 deficiency affected the number of specific cell groups. The proliferation marker Ki67, showed that ISC proliferation was significantly decreased in $Dhx9^{\Delta IEC}$ mice (Fig. 2g). The number of lysozyme-positive Paneth cells were reduced in the ileum of $Dhx9^{\Delta IEC}$ mice (Fig. 2h). We also observed a significant decrease in the number of mucin2-positive goblet cells in both the colon and ileum, which was accompanied by a noteworthy reduction in the thickness of the mucus layer within the colon (Fig. 2i, j). Consequently, DHX9 deficiency significantly impacts the epithelial secretory cell compartment. To determine whether the reduction in epithelial secretory cell lineages was associated with decreased cell survival, we performed a terminal deoxynucleotidyl transferase-mediated deoxyuridine triphosphate nick end labeling (TUNEL) assay. Our findings indicated a higher incidence of TUNEL-positive cells in the intestines of $Dhx9^{\Delta IEC}$ mice, especially within the crypt regions (Supplementary Fig. 5a). Additionally, the proportion of cleaved Caspase-3–positive cells was significantly higher in the $Dhx9^{\Delta IEC}$ group compared to their $Dhx9^{fl/fl}$ littermates, again predominantly in the crypt areas (Supplementary Fig. 5b). These results suggest that DHX9 plays a crucial role in maintaining IEC homeostasis.

Paneth cells secrete antimicrobial peptides, and goblet cells produce mucins, contributing to the control of commensal microbiota[23,24]. Therefore, we investigated whether the loss of DHX9, which led to an abnormal intestinal secretory cell lineage, would further result in defects in microbial control. We performed 16 S rRNA amplicon sequencing to assess the composition of the microbiota taxa in fecal samples obtained from adult single-housed $Dhx9^{\Delta IEC}$ mice and their $Dhx9^{fl/fl}$ littermates. Principal coordinates analysis (PCoA) revealed distinct differences in the composition of the bacterial community between fecal samples from $Dhx9^{\Delta IEC}$ mice and their $Dhx9^{fl/fl}$ littermates (Supplementary Fig. 6a). Detailed analysis at the family level demonstrated $Dhx9^{\Delta IEC}$ mice exhibit impaired control of the commensal microbiota (Supplementary Fig. 6b–e). Additionally, our cohousing experiments suggested that the compromised regulation of commensal microbiota, attributable to DHX9 deficiency, plays a contributory role in the development of the DSS-induced colitis (Supplementary Fig. 7a–d). However, this factor was not identified as the primary determinant. Taken together, our findings suggest that loss of DHX9 leads to a decrease in the number of ISCs and epithelial secretory cell types.

## DHX9 deficiency impairs intestinal crypt and organoid formation

The intestinal crypts are composed of Paneth cells and ISCs, and they are crucial for maintaining continuous renewal and differentiation of the intestinal epithelium[25,26]. The sequencing and staining results shown that $Dhx9^{\Delta IEC}$ mice displayed a significant reduction in the number of ISCs and Paneth cells. Therefore, we hypothesize that the morphology and structure of the intestinal crypts in $Dhx9^{\Delta IEC}$ mice would exhibit significant abnormalities. In line, transmission electron microscopy (TEM) showed a marked reduction in the number of Paneth cells with typical morphological features at the ileum crypt bases of $Dhx9^{\Delta IEC}$ mice. The remaining Paneth cells showed reduced secretory granules, intracytoplasmic vacuolations, and rough endoplasmic reticulum dilation (Fig. 3a). Of particular note, the ISCs,

located in close proximity to the Paneth cells, also exhibited morphological abnormalities, characterized by an increase in cytoplasmic vacuolation, as well as swelling, dissolution, and disruption of the mitochondrial cristae (Fig. 3a).

To analyze the functional relevance of DHX9 in ISCs, ex vivo culture of intestinal organoids was generated from the small intestinal crypts of $Dhx9^{\Delta IEC}$ and $Dhx9^{fl/fl}$ mice. Most of the crypts from $Dhx9^{fl/fl}$ intestine budded at day 2; however, $Dhx9$-deficient crypts exhibited increased cell death and produced fewer outgrowths (Fig. 3b). Propidium iodide (PI) staining conformed abundant cell death in $Dhx9^{\Delta IEC}$ organoids (Fig. 3c). These results suggesting that DHX9 is essential for the growth and development of crypts into organoids. Next, we compared the growth capacity of budding organoids by monitoring the volume changes of organoids in the same visual field between days 2 and 6. We found that $Dhx9$-deficient organoids grew at a significantly slower rate and differentiated significantly less than control organoids (Fig. 3d). Furthermore, to assess whether living $Dhx9^{\Delta IEC}$ organoids could maintain DHX9 deficiency, we serially passaged six-day-old organoids. Our findings revealed that as the passage number increased, both the budding rate and growth speed of the $Dhx9^{\Delta IEC}$ organoids gradually increased (Fig. 3e, f). Correspondingly, we observed a progressive recovery in $Dhx9$ mRNA levels (Fig. 3g). This pattern suggests that DHX9 is vital for the growth of organoids. Taken together, these data suggest that DHX9 deficiency in the epithelium impairs intestinal crypt and organoid formation.

## Paneth-cell-specific DHX9 deletion is dispensable for secretory cell differentiation and DSS-induced colitis

Paneth cells, situated at the base of the small intestinal crypts and neighboring Lgr5+ ISCs, play critical roles in maintaining the integrity of the intestinal epithelial barrier. The loss of Paneth cells and their antimicrobial peptides is a characteristic feature of CD[27]. Given the observation that DHX9 depletion resulted in a significant reduction in the number of ISCs, Paneth cells, and other secretory cells, we speculate that this effect may have arisen from two potential mechanisms. First, DHX9 deficiency directly regulates the function of these specific cell populations, leading to a decrease in their abundance. Second, the loss of DHX9 may directly affect ISC function, impeding the differentiation of these cells into Paneth cells and other secretory cell types. To elucidate whether DHX9 depletion directly affects the quantity of secretory cells, especially Paneth cells, we generated $Dhx9^{fl/fl}$ Defa6-Cre ($Dhx9^{\Delta Paneth}$) mice, allowing for specific deletion of DHX9 in Paneth cells. Our IF staining results showed that DHX9 was specifically knocked down in Paneth cells (Supplementary Fig. 8). We then compared whether cell subpopulations, as well as immune response, were significantly changed in $Dhx9^{\Delta Paneth}$ IECs. No discernible differences were observed in the expression of epithelial secretory cell lineages marker genes as well as interferon stimulates genes (ISGs) in these mice (Supplementary Fig. 9a). Additionally, we evaluated the susceptibility of $Dhx9^{\Delta Paneth}$ mice to DSS-induced colitis (Supplementary Fig. 9b). No significant differences were observed in terms of weight changes (Supplementary Fig. 9c), colon length shortening (Supplementary Fig. 9d), and intestinal histopathological scores (Supplementary Fig. 9e). Collectively, these findings indicate that Paneth cell-specific DHX9 deletion has no significant impact on IEC lineage commitment or DSS-induced colitis. Therefore, we speculate that the effect of DHX9 on intestinal function is mainly mediated by its influence on the functionality of ISCs.

## DHX9 deficiency in Lgr5+ stem cells decreases the number of ISCs and enhances DSS-induced colitis

To investigate whether DHX9 deficiency directly affects the quantity and functionality of ISCs, thereby influencing secretory cell lineage commitment and intestinal homeostasis, we generated tamoxifen-inducible DHX9 conditional-knockout mice (hereafter named

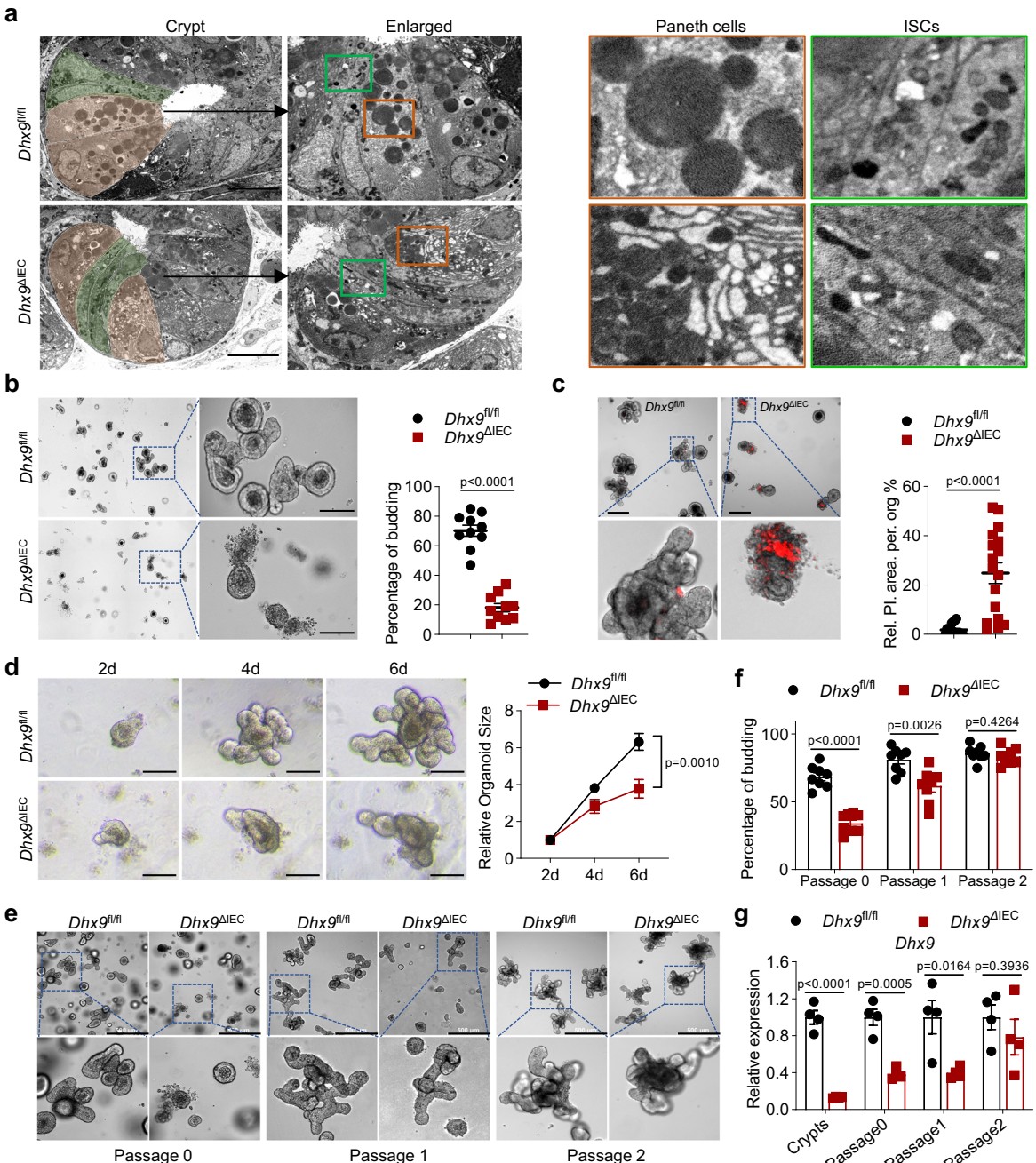

**Fig. 3 | DHX9 deficiency impairs intestinal crypt and organoid formation.**
**a** Representative transmission electron microscopy (TEM) images of ileal crypt bases from $Dhx9^{fl/fl}$ and $Dhx9^{\Delta IEC}$ mice. ISCs are highlighted in green, and Paneth cells are indicated in brown. Scale bar, 10 μm. Experiment was performed on $n = 3$ mice individually, with similar results. **b** Representative brightfield images of $Dhx9^{fl/fl}$ and $Dhx9^{\Delta IEC}$ ileum organoids after 2 days of culture ($n = 10$ per group). Scale bar, 100 μm. Quantifications provided on the right. Three individual experiments were performed, with similar results. **c** Propidium iodide (PI) staining of organoids derived from $Dhx9^{fl/fl}$ and $Dhx9^{\Delta IEC}$ mice. Scale bar, 200 μm. The panel to the right of the images provides a statistical analysis, quantifying the relative area of PI staining per organoid ($Dhx9^{fl/fl}$, $n = 16$; $Dhx9^{\Delta IEC}$, $n = 17$). Three individual experiments were performed, with similar results. **d** Comparative analysis of the growth status of intestinal organoids from $Dhx9^{fl/fl}$ and $Dhx9^{\Delta IEC}$ mice at different time

points (2 days, 4 days, and 6 days). Scale bar, 100 μm. The growth curves of intestinal organoids provided on the right ($n = 11$ per group). Three individual experiments were performed, with similar results. **e–g** $Dhx9^{fl/fl}$ and $Dhx9^{\Delta IEC}$ organoids were cultured for 6 days and underwent two serial passages, with images captured for each generation to illustrate morphological changes, scale bar, 500 μm (**e**) analysis of the budding rate in each generation of the organoids, $n = 8$ per group (**f**) and quantitative analysis of $Dhx9$ mRNA levels in the original crypts and across each generation of organoids, $n = 4$ per group (**g**). Data are representative of three independent experiments. Error bars show means ± SEM. Statistical analyses of organoid size changes were performed using two-way ANOVA analysis with Tukey's multiple comparisons (**d**). Two-tailed unpaired Student's $t$-test were used for other analyses. Source data are provided as a Source Data file.

$Dhx9^{i\Delta ISC}$) with a specifically targeted deletion of DHX9 in Lgr5[+] stem cells, by crossing $Lgr5$-$EGFP$-$IRES$-$cre^{ERT2}$ mice with $Dhx9^{fl/fl}$ mice. The transgenic mice carried an EGFP marker, enabling the monitoring of Lgr5[+] stem cells. IF staining in $Dhx9^{i\Delta ISC}$ mice post-tamoxifen treatment

revealed a notable knockdown of DHX9 in EGFP-positive ISCs (Supplementary Fig. 10). However, not all crypts displayed EGFP positivity. To assess the knockout efficiency in $Lgr5$-$EGFP$-$IRES$-$cre^{ERT2}$ mice, we crossed them with Ai14 reporter mice ($Rosa26^{lsl\text{-}tdTomato}$). In $Lgr5$-$EGFP$-

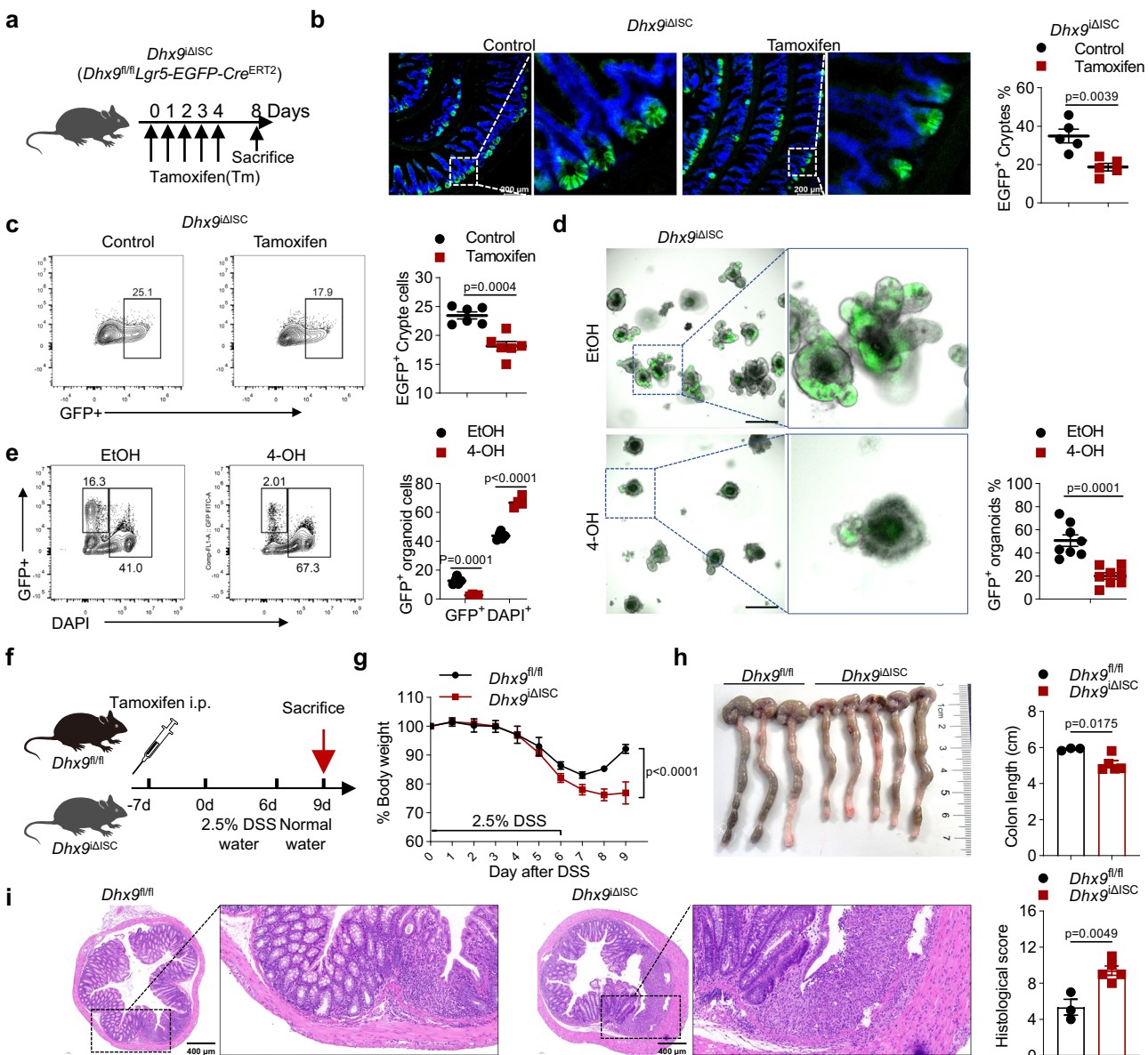

**Fig. 4 | DHX9 deficiency in Lgr5+ stem cells decreases the number of ISCs and enhances DSS-induced colitis. a** Schematic diagram illustrating the strategy of tamoxifen administration. *Dhx9*iΔISC (*Dhx9*fl/fl*Lgr5-EGFP-IRES-cre*ERT2) mice received tamoxifen injections for 5 consecutive days. **b** Representative images of ileums from corn oil-injected (control) and tamoxifen-treated *Dhx9*iΔISC mice. (*n* = 5 mice per group). Scale bar, 200 μm. Quantification of Lgr5-EGFP+ crypts in the small intestine of mice is provided on the right. **c** Representative flow cytometry plots showing the percentages of EGFP+ crypt cells from corn oil-injected (control) and tamoxifen-treated *Dhx9*iΔISC mice (*n* = 6 mice per group). Quantification of EGFP+ crypt cells is provided on the right. **d** Representative images of *Dhx9*iΔISC ileum organoids after 2 days of culture treated with EtOH (control) or 4-OH. Scale bar, 200 μm. Quantification of EGFP+ organoids were provided on the right (*n* = 8 per group). **e** Representative flow cytometry plots showing the percentages of EGFP+

organoid cells, obtained from organoids cultured for 2 days that treated with EtOH (*n* = 5) or 4-OH (*n* = 4). Quantification of EGFP+ and DAPI+ organoid cells is provided on the right. **f**–**i** Experimental design depicting the induction of colitis using DSS. *Dhx9*iΔISC mice (*n* = 5) and their *Dhx9*fl/fl littermates (*n* = 3) were injected with tamoxifen from 7 days before DSS treatment (5 consecutive injections), followed by exposure to 2.5% DSS for 6 days (**f**) monitoring of body weight changes throughout the experimental period (**g**); measurement of colon length upon sacrifice on day 9 (**h**); and representative H&E staining of colonic tissue, Scale bar, 400 μm (**i**). Data are representative of three independent experiments; error bars show means ± SEM. Statistical analyses of body weight changes were performed using two-way ANOVA analysis with Tukey's multiple comparisons (**g**). Two-tailed unpaired Student's *t*-test were used for other analyses. Source data are provided as a Source Data file.

*Cre*ERT2: Rosa26lsl-tdTomato mice, once Cre is expressed, the fluorescence of tdTomato is activated (Supplementary Fig. 11a). We observed that almost all EGFP-positive crypts exhibited red fluorescence throughout the villi or crypt after tamoxifen treatment, indicating that the *Lgr5*-driven gene knockout in ISCs was maintained as these cells differentiated into various IEC subpopulations (Supplementary Fig. 11b). Furthermore, imaging of the entire intestine in *Lgr5-EGFP-Cre*ERT2: Rosa26lsl-tdTomato mice six days after tamoxifen treatment revealed that

~40% of the villi or crypt were red (Supplementary Fig. 11c). This finding suggests that in *Dhx9*iΔISC mice, about 40% of the epithelial cells exhibited DHX9 knockout.

Upon tamoxifen administration, we observed a significant restriction of EGFP-tagged Lgr5+ ISCs in the crypts of *Dhx9*iΔISC mice (Fig. 4a, b), indicating that DHX9 deletion in Lgr5+ stem cells directly affects the number of ISCs. Consistently, we isolated the intestinal crypts and performed flow cytometry analysis to evaluate the

proportion of EGFP⁺ crypt cells, and similarly found a reduced proportion of ISCs in the crypt cells in tamoxifen-treated *Dhx9*[iΔISC] mice (Fig. 4c). To further investigate the specific impact of DHX9 on ISCs, we generated ex vivo intestinal organoids from *Dhx9*[iΔISC] mice and induced efficient deletion of DHX9 in ISCs by treating them with 4-hydroxytamoxifen (4-OH) or vehicle control (ethanol, EtOH). *Dhx9*[iΔISC] organoids exposed to 4-OH exhibited significantly impaired growth and increased cell death (as indicated by PI positivity) (Supplementary Fig. 12), while 4-OH treatment did not affect the growth of *Dhx9*[fl/fl] organoids (Supplementary Fig. 12), suggesting that the observed growth impairment in the organoids is specifically attributable to DHX9 deficiency. Furthermore, there was a substantial reduction in the number of EGFP⁺ organoids (Fig. 4d). Flow cytometry analysis of the organoid cells revealed a significant decrease in EGFP⁺ cells and an increase in cell death in the 4-OH-treated group (Fig. 4e). Thus, DHX9 is required for ISC proliferation. Since ISC damage affects the differentiation and migration of this cell type into epithelial secretory cell lineages, which in turn causes epithelial system disruption. Therefore, we performed IF staining for lysozyme and mucin2 in the intestines of *Dhx9*[fl/fl] and *Dhx9*[iΔISC] mice after treatment with tamoxifen to assess the impact of DHX9 deficiency on the number of Paneth cells and goblet cells. *Dhx9*[iΔISC] mice exhibited a significant reduction in the number of Paneth cells and goblet cells (Supplementary Fig. 13a, b). Moreover, the intestines of tamoxifen-treated *Dhx9*[iΔISC] mice exhibited a significant increase in both TUNEL-positive and caspase-3–positive cells compared to corn oil-injected *Dhx9*[iΔISC] mice (Supplementary Fig. 14a, b), particularly in the crypt areas. This suggests a crucial role for DHX9 in the survival and maintenance of ISCs.

Lgr5⁺ ISCs drive intestinal maintenance in terms of homeostasis and regeneration in response to injury[28]. To understand the role of DHX9 in adult ISC maintenance, we challenged *Dhx9*[iΔISC] mice and their *Dhx9*[fl/fl] littermates with 2.5% DSS to induce colitis (Fig. 4f). Consistent with the phenotype of *Dhx9*[ΔIEC], *Dhx9*[iΔISC] mice displayed significant weight loss (Fig. 4g) and colon length shortening (Fig. 4h). Histological analysis revealed obvious increases in inflammatory infiltrates, edema, and disrupted epithelial structure in the colon of *Dhx9*[iΔISC] mice when compared with their *Dhx9*[fl/fl] littermates (Fig. 4i). Collectively, these results demonstrate that DHX9 is critical for maintaining ISC function, and its absence directly leads to a reduction in the number of ISCs, thereby influencing secretory cell lineage commitment and intestinal homeostasis.

## DHX9 is essential for maintaining R-loop homeostasis and preventing genomic instability

Next, we aimed to elucidate the mechanisms underlying the functional impairment in ISCs caused by DHX9 deficiency. ISCs undergo rapid proliferation and differentiation to sustain continuous IEC renewal, making genomic stability crucial during extensive genome replication. Considering the role of DHX9 in maintaining genome stability[29,30], we postulated that inactivation of DHX9 results in genome damage and subsequent ISC death. First, we examined the protein expression of γH2AX, which is a marker of DNA double-stranded break foci used to assess genome instability, in *Dhx9*[ΔIEC] IECs. Western blot revealed a significant increase in γH2AX expression in DHX9 deficiency IECs (Fig. 5a). Consistently, IF staining also revealed elevated levels of γH2AX in the intestine of *Dhx9*[ΔIEC] mice, specifically in the intestinal crypts where ISCs reside (Fig. 5b). Moreover, we employed the neutral comet assay to detect genome instability in IECs. DHX9 depletion resulted in a significant increase in double-stranded breaks (Fig. 5c). These findings underscore the essential role of DHX9 in suppressing genomic instability.

Notably, DHX9 has been experimentally verified to unwind DNA and RNA displacement loops[13,14], and accumulating evidence

implicates abnormal R-loop accumulation as a key driver of genomic instability[31]. Therefore, we hypothesized that the observed genome instability following DHX9 deletion was attributed to aberrant R-loop accumulation in *Dhx9*-deficient cells. To test this hypothesis, we isolated IECs from *Dhx9*[ΔIEC] mice and their littermates, and we quantified global R-loops levels using dot blot analysis with a monoclonal antibody (S9.6) that specifically recognizes RNA:DNA hybrids in a sequence-independent manner. Indeed, the absence of DHX9 resulted in a marked increase in the overall abundance of R-loops, which were sensitive to RNase H1 treatment (Fig. 5d, e). To further confirm this, we adopted an alternative approach to detect the accumulation of nuclear R-loops. HeLa cells stably expressing both ZsGreen and RNase H1 on the same plasmid were constructed, with RNase H1 expression controlled by a doxycycline-inducible promoter. We then employed small interfering RNA (siRNA) transfection to efficiently silence DHX9 in these HeLa cells and assessed R-loops formation using indirect IF microscopy. Notably, the nuclei of cells transfected with DHX9 siRNA displayed significantly stronger RNase H1-sensitive fluorescent signals compared to the control group, indicating an accumulation of R-loops in *Dhx9*-deficient cells (Fig. 5f, g). Furthermore, we extracted DNA from the intestinal crypts of *Dhx9*[iΔISC] and *Dhx9*[fl/fl] mice induced with tamoxifen to assess the levels of R-loops. Dot blot analysis revealed a significant increase in the abundance of RNase H1-sensitive R-loops in *Dhx9*[iΔISC] crypts (Supplementary Fig. 15). These results indicate that loss of DHX9 causes R-loops accumulation, which probably triggered the observed genomic instability.

To further explore the impact of DHX9 deletion on R-loops, we utilized the R-loop CUT&Tag method[32,33] to analyze the distribution and characteristics of R-loops in *Dhx9*-deficient IECs. A comparison of R-loop signals across entire transcripts revealed that *Dhx9*[ΔIEC] IECs exhibited higher signal intensity compared to *Dhx9*[fl/fl], and *Dhx9*[ΔIEC] treated with the RNase H group (Supplementary Fig. 16a). A total of 5,524 R-loop peaks were identified in *Dhx9*[ΔIEC] IECs, significantly exceeding the 1,481 peaks found in the *Dhx9*[fl/fl] group across various regions including promoters, exons, introns, and intergenic areas, particularly in the promoter regions (Supplementary Fig. 16b). Genome browser representations of specific genes showed increased R-loop signals in different gene regions of *Dhx9*[ΔIEC] IECs, which notably decreased following RNase H treatment (Supplementary Fig. 16c). Subsequently, we performed an intersection analysis between genes associated with R-loop peaks and those significantly upregulated or downregulated in the RNA-seq analysis of *Dhx9*-deficient IECs. Our findings reveal that about one-tenth of the genes with significant changes in the RNA-seq analysis also displayed R-loop signals. This ratio is consistent with the general percentage of genes in the genome exhibiting R-loop signals (Supplementary Fig. 16d). Therefore, we speculate that the genes showing significant alterations in *Dhx9*[ΔIEC] IECs may not be directly regulated by R-loops, but could be indirectly influenced by R-loop-mediated genomic DNA damage. Next, we examined the expression of γH2AX and R-loops in intestinal biopsy samples obtained from patients with active IBD (including patients with UC and CD). Consistent with our hypothesis, both γH2AX and R-loop levels were significantly elevated in samples in both UC and CD patients when compared with healthy controls (Supplementary Fig. 17a, b). It is important to acknowledge that when employing staining methods to detect R-Loops, a significant number of positive signals are also observed in the cytoplasm. This occurrence may potentially impact the precision of R-loop detection results.

Furthermore, chronic intestinal inflammation and genomic instability are closely associated with the development of intestinal cancer. Therefore, we investigated the role of DHX9 in tumorigenesis using a model of intestinal adenomatosis, employing the well-established *Apc*[min/+] mouse tumor model. *Apc*[min/+] mice carry a dominant mutation in the adenomatous polyposis coli gene (*Apc*) and

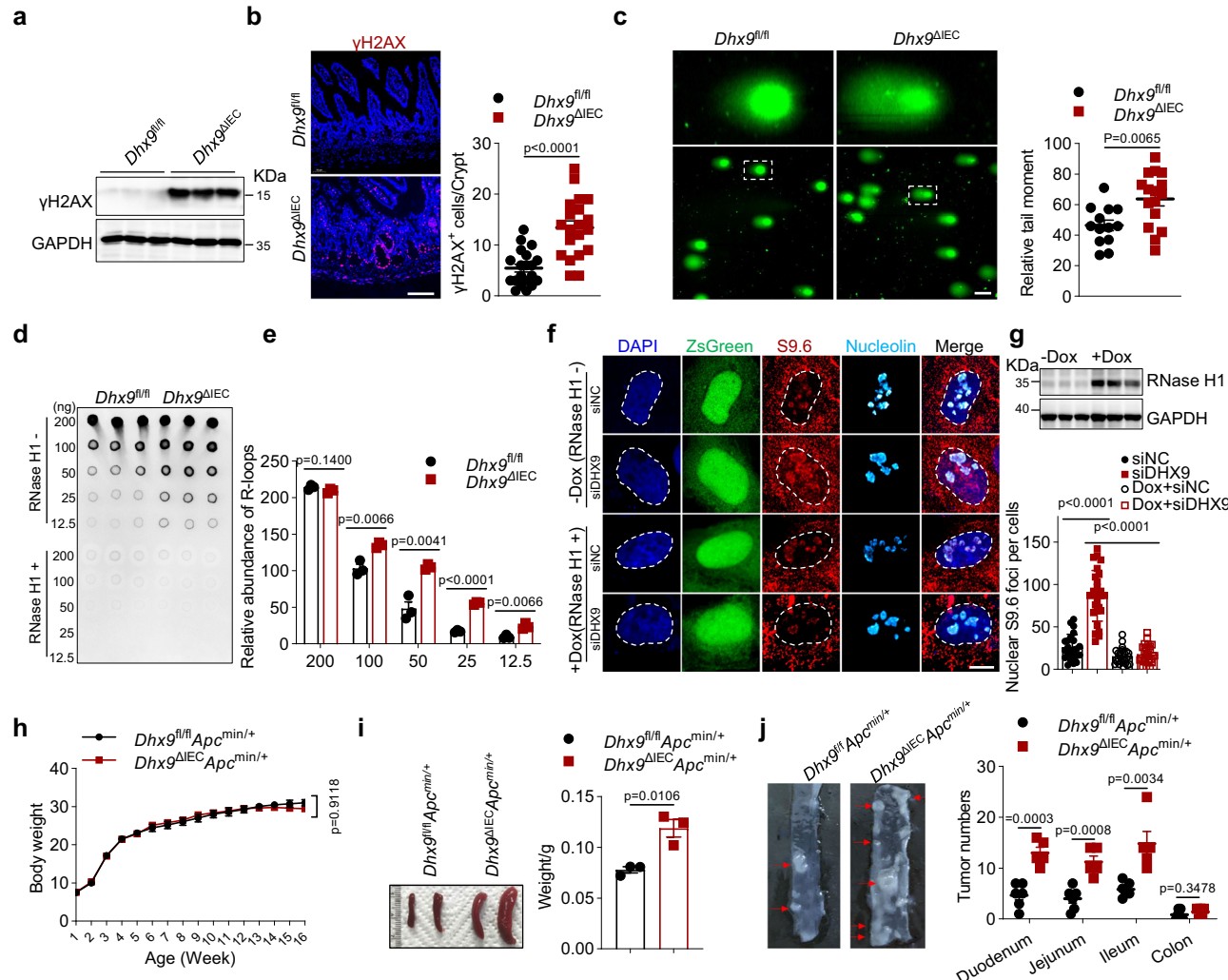

**Fig. 5 | DHX9 deficiency leads to R-loop accumulation and genomic instability.**
**a** Western Blotting analysis of γH2AX and GAPDH (loading control) in lysates from *Dhx9*^fl/fl^ and *Dhx9*^ΔIEC^ IECs. Three individual experiments were performed, with similar results. **b** Expression pattern of γH2AX in the ileum of *Dhx9*^fl/fl^ and *Dhx9*^ΔIEC^ mice. γH2AX is shown in red, and nuclei are stained with DAPI (blue). *n* = 20 villus axes were counted. Scale bar, 100 µm. Experiment was performed on *n* = 3 mice individually, with similar results. **c** Neutral comet assay of *Dhx9*^fl/fl^ and *Dhx9*^ΔIEC^ IECs. Representative images of comet tails in IECs stained with SYBR green. Scale bar, 10 µm. Quantifications are provided on the right (*Dhx9*^fl/fl^, *n* = 13; *Dhx9*^ΔIEC^, *n* = 16). Three individual experiments were performed, with similar results. **d**, **e** Dot blot analysis to quantify R-loops in IECs from *Dhx9*^ΔIEC^ and *Dhx9*^fl/fl^ mice (*n* = 3 per group). RNase H1 treatment was included as a negative control (**d**). Quantitative analysis of the dot blot results (**e**). **f** IF staining of HeLa cells stably transfected with a doxycycline (Dox)-inducible RNase H1 expression plasmid, which concurrently expresses zsGreen. These cells underwent treatment with specified siRNAs for 48 h

and were either induced with Dox for 24 h or left uninduced. Staining includes the S9.6 antibody (red) to detect R-loops and the nucleolin antibody (cyan) for nucleolar identification. Scale bar, 10 µm. **g** Upper panel: WB analysis of Flag-RNase H1 and Actin in lysates from Dox+ and Dox- HeLa cells from (**f**). Lower panel: Quantitative analysis of the dot blot results from panel (**f**) (SiNC, *n* = 23; SiDHX9, *n* = 27; Dox + SiNC, *n* = 25; Dox + SiDHX9, *n* = 27). Three individual experiments were performed, with similar results. **h**, **i** Monitoring of body weight changes over 16 weeks (**h**), and weight of spleen at 20 weeks (**i**) in *Dhx9*^fl/fl^*Apc*^min/+^ and *Dhx9*^ΔIEC^*Apc*^min/+^ mice (*n* = 3 mice per group). **j** The number of tumors in the intestine was calculated for *Dhx9*^fl/fl^*Apc*^min/+^ (*n* = 6) and *Dhx9*^ΔIEC^*Apc*^min/+^ mice (*n* = 5). Arrows indicate the tumors. Error bars show means ± SEM. Statistical analyses for (**h**) were conducted using two-way ANOVA analysis with Tukey's multiple comparisons. Two-tailed unpaired Student's *t*-test were used for other analyses. Source data are provided as a Source Data file.

develop multiple adenomas throughout the intestinal tract, primarily in the small intestine[34]. We generated mouse lines carrying the *Apc* mutation with or without specific intestinal deletion of DHX9, referred to as *Dhx9*^fl/fl^*Apc*^min/+^ and *Dhx9*^ΔIEC^*Apc*^min/+^, respectively. Compared with control *Dhx9*^fl/fl^*Apc*^min/+^ mice, the body weight of *Dhx9*^ΔIEC^*Apc*^min/+^ mice did not exhibit significant difference (Fig. 5h), but their spleens were significantly enlarged at 20 weeks of age (Fig. 5i). Notably, *Dhx9*^ΔIEC^*Apc*^min/+^ mice developed an increased number of tumors than *Dhx9*^fl/fl^*Apc*^min/+^ mice (Fig. 5j). Thus, DHX9 deletion-induced genomic instability promoted the development of intestinal cancer. Collectively, these results indicate that DHX9 defects lead to abnormal R-loop accumulation, which consequently induces genomic instability.

## DHX9 deficiency promotes cGAS-STING dependent inflammation

Accumulating evidence highlights that the cGAS-STING signaling pathway can be activated by genomic instability, leading to interferon-driven inflammation and apoptosis[35]. Furthermore, a recent study elucidated that abnormal R-loop accumulation induces the RNA:DNA hybrids release into the cytoplasm. The cytoplasmic hybrids combine with cGAS and TLR3, activating IRF3 and inducing apoptosis[36]. Based on these findings, we postulated that the aberrant accumulation of R-loops and the resultant genomic instability promoted intestinal inflammation through the cGAS-STING signaling pathway. To confirm this hypothesis, we analyzed the RNA-seq results of IECs isolated from

*Dhx9*<sup>ΔIEC</sup> and *Dhx9*<sup>fl//fl</sup> mice. Gene Ontology analysis revealed that the up-regulated pathways in *Dhx9*<sup>ΔIEC</sup> IECs were primarily associated with interferon-related processes, such as "Defense response to virus", "Defense response to symbiont", and "Pattern recognition receptor signaling pathway" (Fig. 6a). Volcano plot analysis indicated significant enrichment of downstream interferon-related genes (e.g., *Oasl2, ifit2, Isg15*) and NF-κB-related genes (e.g., *Tnfaip2, Il18, Tnfsf23*) in *Dhx9*<sup>ΔIEC</sup> IECs (Fig. 6b). Pathway correlation analysis demonstrated prominent activation of pattern recognition receptors and subsequent aberrant innate immune responses in *Dhx9*<sup>ΔIEC</sup> IECs (Fig. 6c). These significantly upregulated genes and pathways are important downstream targets of cGAS-STING. To clarify whether cGAS-STING activation is a consequence of genomic instability induced by DHX9 deficiency, we executed a time-course study following tamoxifen treatment in *Dhx9*<sup>iΔISC</sup> and control Dhx9<sup>fl/fl</sup> mice. Our findings indicate a reduction in the levels of *Ang4* and *Lgr5* as early as day 2 post-treatment, suggesting an early impact on the proliferation of ISCs and Paneth cells. However, at this stage, there were no notable differences in the levels of cGAS-STING downstream targets *Isg15* and *Oasl2* (Supplementary Fig. 18). Consistently, in *Dhx9*<sup>ΔIEC</sup> mice, a notable decrease in the expression of *Defa22* and *Ang4* in IECs was observed at 3 days post-birth. However, at this early stage, ISGs did not exhibit significant changes (Supplementary Fig. 19). These results imply that disruptions in ISC and Paneth cell functions precede the activation of the cGAS-STING pathway. Collectively, we propose that the activation of the cGAS-STING pathway induced by genomic instability serves as a key driver of innate immune activation.

Moreover, it has been observed that necrotic DNA, or cGAMP (synthesized by cGAS), can spread to neighboring cells, leading to remote STING activation and subsequent antiviral immune responses[37,38]. Therefore, we performed RT-qPCR to evaluate changes downstream of STING (interferon and NF-κB-related genes) in the intestine following IEC removal, revealing a significant upregulation of these genes in *Dhx9*<sup>ΔIEC</sup> mice (Supplementary Fig. 20). Therefore, *Dhx9*-deficient IECs are also capable of activating the cGAS-STING pathway in adjacent immune cells. Additionally, the activation of innate immunity would also induce an increase in myeloid cells within the lamina propria layer (LPL). We therefore examined the proportion of myeloid cells in the LPL isolated from *Dhx9*<sup>ΔIEC</sup> and *Dhx9*<sup>fl/fl</sup> mice. We observed a notable increase in myeloid cells in *Dhx9*<sup>ΔIEC</sup> mice after 5 days of DSS induction, including CD11b⁺, CD11b⁺Ly6c⁺, and CD11b⁺Ly6g⁺ cells (Fig. 6d). Thus, DHX9 deficiency in IECs amplifies cGAS-dependent inflammatory responses in both the IECs and myeloid cell populations.

Based on the activation of the genomic instability-induced cGAS-STING pathway as a contributing factor to colitis, we questioned whether blocking the cGAS-STING pathway in *Dhx9*<sup>ΔIEC</sup> mice would partially alleviate the colitis phenotype. To investigate this, we used a knockout strategy to eliminate *Sting* expression specifically in *Dhx9*<sup>ΔIEC</sup> mice, thereby inhibiting this pathway in vivo. *Dhx9*<sup>ΔIEC</sup>*Sting*⁻/⁻ mice exhibited a significant reduction in intestinal inflammation compared with *Dhx9*<sup>ΔIEC</sup> mice (Fig. 6e). As *Dhx9*<sup>ΔIEC</sup>*Sting*⁻/⁻ mice can mitigate the inflammatory response caused by DNA damage, we investigated whether this could also reverse the reduction in ISCs and epithelial secretory cell types. Our RT-qPCR analysis indicated that, compared to *Dhx9*<sup>ΔIEC</sup> mice, the *Dhx9*<sup>ΔIEC</sup>*Sting*⁻/⁻ mice showed a partial restoration in the proportions of ISCs and goblet cells in the intestine. However, there was no change observed in the populations of Paneth cells and EECs (Supplementary Fig. 21a). These findings suggest that additional knockout of *Sting* in DHX9 deficiency IECs can alleviate intestinal epithelial abnormalities, but its effectiveness is still limited. Additionally, we assessed the R-loops levels in *Dhx9*<sup>ΔIEC</sup>*Sting*⁻/⁻ IECs, and found that the *Dhx9*<sup>ΔIEC</sup>*Sting*⁻/⁻ IECs did not reverse the aberrant accumulation of R-loops (Supplementary Fig. 21b). Subsequently, we challenged *Dhx9*<sup>fl//fl</sup> mice, *Dhx9*<sup>ΔIEC</sup> mice, *Dhx9*<sup>ΔIEC</sup>*Sting*⁻/⁻ mice, and *Sting*⁻/⁻ mice with DSS to induce colitis (Fig. 6f). Compared with *Dhx9*<sup>ΔIEC</sup> littermates,

DSS-treated *Dhx9*<sup>ΔIEC</sup>*Sting*⁻/⁻ mice demonstrated colitis attenuation, as indicated by reduced weight loss (Fig. 6g), less colon length shortening (Fig. 6h), and less inflammatory cell infiltration and crypt damage (Fig. 6i). These results suggest that blocking the cGAS-STING pathway partially ameliorates the colitis phenotype in *Dhx9*<sup>ΔIEC</sup> mice. It is noteworthy that even in a *Sting* knockout background, the additional deletion of DHX9 still led to a markedly more severe DSS colitis, as evidenced by the comparison between *Dhx9*<sup>ΔIEC</sup>*Sting*⁻/⁻ mice and *Sting*⁻/⁻ mice (Fig. 6f to i). These results suggest that the cellular damage caused by DHX9 deficiency is a critical factor in the dysfunction of ISCs, and the cGAS-STING pathway mediated inflammatory response accelerate this process. Take together, our results suggest that DHX9 depletion-induced intestinal inflammation is partially dependent on the amplification of inflammation through the cGAS-STING pathway. Hence, R-loop-mediated genomic instability, along with triggered activation of the cGAS-STING pathway, impairs ISC function and contributes to the pathogenesis of IBD (Fig. 7).

## Discussion

RNA helicases constitute a large family of proteins that function in a variety of cellular processes, such as transcription, splicing, translation, and RNA decay[39]. Emerging evidence suggests that RNA helicases also play a crucial role in the maintenance of intestinal homeostasis. For instance, loss-of-function variants of MDA5, an important member of the RNA helicase family, have been implicated in very early onset IBD[40]. DDX5 has been shown to regulate the function of epithelial tuft cells and intestinal regulatory T cells, which control the microbial repertoire and inhibits intestinal inflammation[41,42]. However, the roles of most RNA helicases in regulating intestinal function remain unclear. In our previous studies, we demonstrated that the RNA helicase DHX15 can interact with NLRP6 in IECs to facilitate viral RNA recognition, leading to downstream interferon pathways activation or promoting phase separation and activation of the NLRP6 inflammasome, thus exerting antiviral activity[43,44]. Furthermore, DHX9 pairs with Nlrp9b to sense short dsRNA, forming inflammasome complexes that promote the maturation of interleukin-18 and GSDMD-induced pyroptosis, thus providing resistance against rotavirus infection in IECs[18]. Moreover, DHX15 regulates Wnt-induced α-defensins in Paneth cells, contributing to the antibacterial response in the intestine[45]. In this study, we investigated the role of DHX9 in intestine by using DHX9 conditional knockout mice in IECs, Paneth cells, and ISCs. Our findings indicate that DHX9 regulates the transcriptional stability of R-loops, thus maintaining ISC function and preserving the integrity of the intestinal epithelium.

IBD is a complex disorder with an unclear etiology. One fundamental question in IBD pathogenesis is identifying the factors that initiate inflammation. IBD susceptibility loci indicated that not only immune cell-related genes (e.g., *IL10*, *IL23R*, *TNFSF15*), epithelial-specific genes (e.g., *ATG16L1*, *XIAP*, *MUC19*) may also determine susceptibility to IBD[46]. Extensive research has implicated immune cell dysregulation as a driving force of IBD[47,48]. However, recent studies have shed light on the impaired function of epithelial cells, particularly Paneth cells, as a site of origin for intestinal inflammation[27,49,50]. Our study suggesting that dysregulated ISCs play a pivotal role in initiating intestinal inflammation. Importantly, our findings demonstrate that the loss of DHX9 in ISCs significantly impairs the secretory lineage of the epithelium, particularly affecting Paneth cells. However, direct knockout of DHX9 in Paneth cells did not affect the susceptibility of mice to enteritis. These findings highlight the critical role of DHX9 derived from ISCs as a driving force behind disruptions in the epithelial system. Consistent with our observations, mitochondrial dysfunction in ISC by Hsp60 loss results in ISC exhaustion, driving ISC transition towards dysfunctional Paneth cells in CD patients[51]. The critical role of ISCs in the pathogenesis of IBD is gaining increasing attention. Reduced ISC proliferation and function contribute to breakdown of

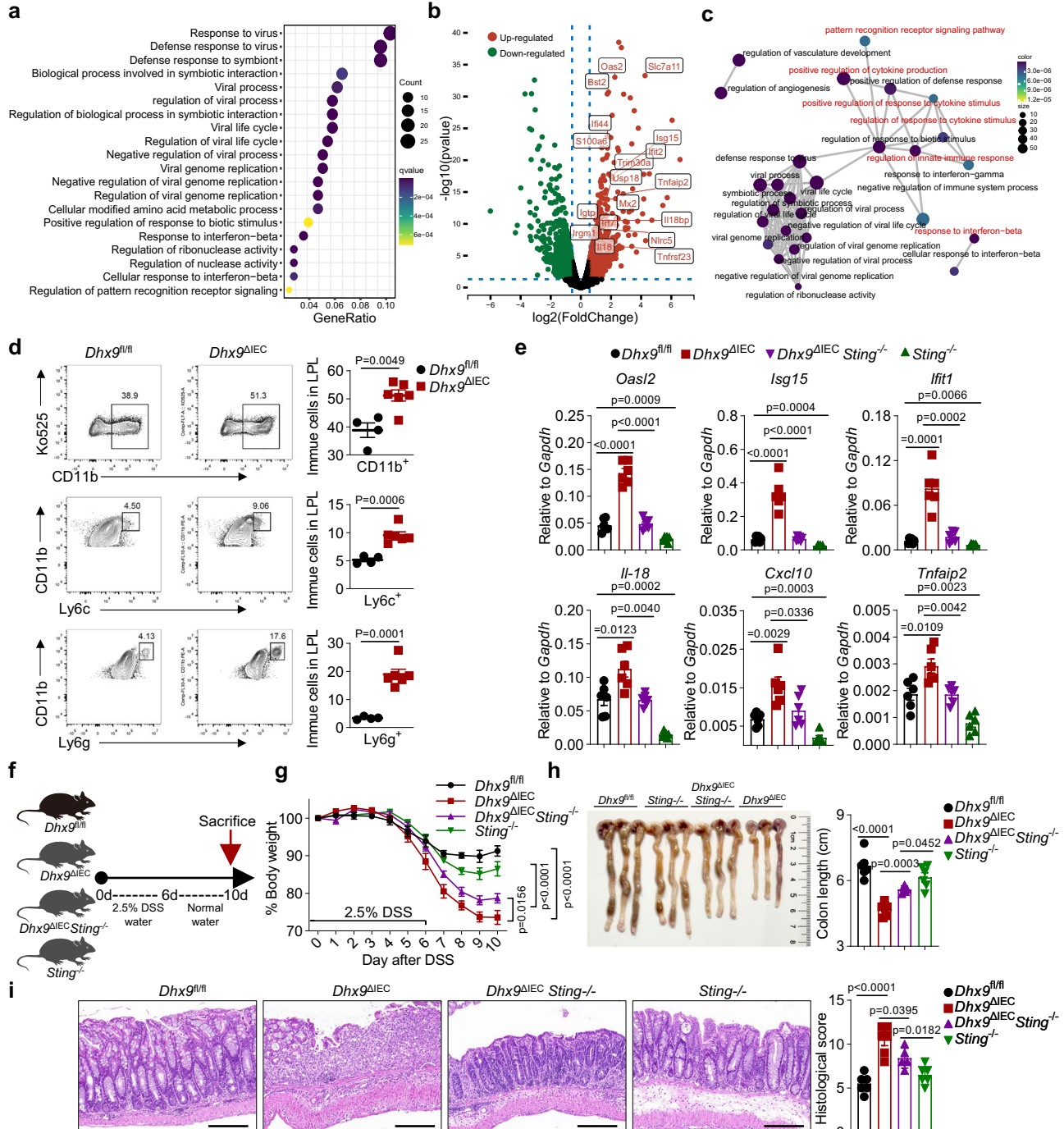

**Fig. 6 | DHX9 deficiency promotes cGAS-STING dependent inflammation.**
**a** Gene ontology analysis revealing the up-regulated expressed genes in IECs of $Dhx9^{\Delta IEC}$ mice compared to $Dhx9^{fl/fl}$ mice, based on RNA-seq data ($P < 0.05$; right-tailed Fischer's exact $t$-test). **b** Volcano plot analysis of the RNA-seq dataset illustrating the top upregulated and downregulated genes in IECs of $Dhx9^{\Delta IEC}$ mice compared to $Dhx9^{fl/fl}$ mice (The black dots represent genes with no statistically significant differences; the green dots and the red dots, $P < 0.05$; right-tailed Fischer's exact $t$-test). **c** Interaction network of up-regulated expressed genes in IECs of $Dhx9^{\Delta IEC}$ compared to $Dhx9^{fl/fl}$, based on RNA-seq data. **d** Representative flow cytometry plots showing the colonic lamina propria-infiltrated myeloid cells of $Dhx9^{fl/fl}$ ($n = 4$) and $Dhx9^{\Delta IEC}$ mice ($n = 6$) on day 5 after DSS treatment, including CD11b⁺, CD11b⁺Ly6c⁺, and CD11b⁺Ly6g⁺ myeloid cells. Quantification is provided on the right. **e** RT-qPCR analysis was performed to assess the expression of

downstream genes of the cGAS-STING pathway, including $Oasl2$, $Isg15$, $Ifit1$, $Il-18$, $Cxcl10$, and $Tnfaip2$, in the ileum of $Dhx9^{fl/fl}$, $Dhx9^{\Delta IEC}$, $Dhx9^{\Delta IEC}Sting^{-/-}$, and $Sting^{-/-}$ mice at 8 weeks of age ($n = 6$ per group). **f–i** Experimental design illustrating the induction of colitis using DSS. $Dhx9^{fl/fl}$ ($n = 6$), $Dhx9^{\Delta IEC}$ ($n = 6$), $Dhx9^{\Delta IEC}Sting^{-/-}$ ($n = 5$), and $Sting^{-/-}$ ($n = 6$) mice were treated with 2.5% DSS for 6 days (**f**), with monitoring of body weight changes throughout the experimental period (**g**); measurement of colon length upon sacrifice on day 9 (**h**); and representative H&E staining of colonic tissue, Scale bar, 200 μm (**i**). Data are representative of three independent experiments, and error bars show means ± SEM. Statistical analyses of body weight changes were performed using two-way ANOVA analysis with Tukey's multiple comparisons (**f**). Two-tailed unpaired Student's $t$-test were used for other analyses (**d**, **e**, **h**, and **i**). Source data are provided as a Source Data file.

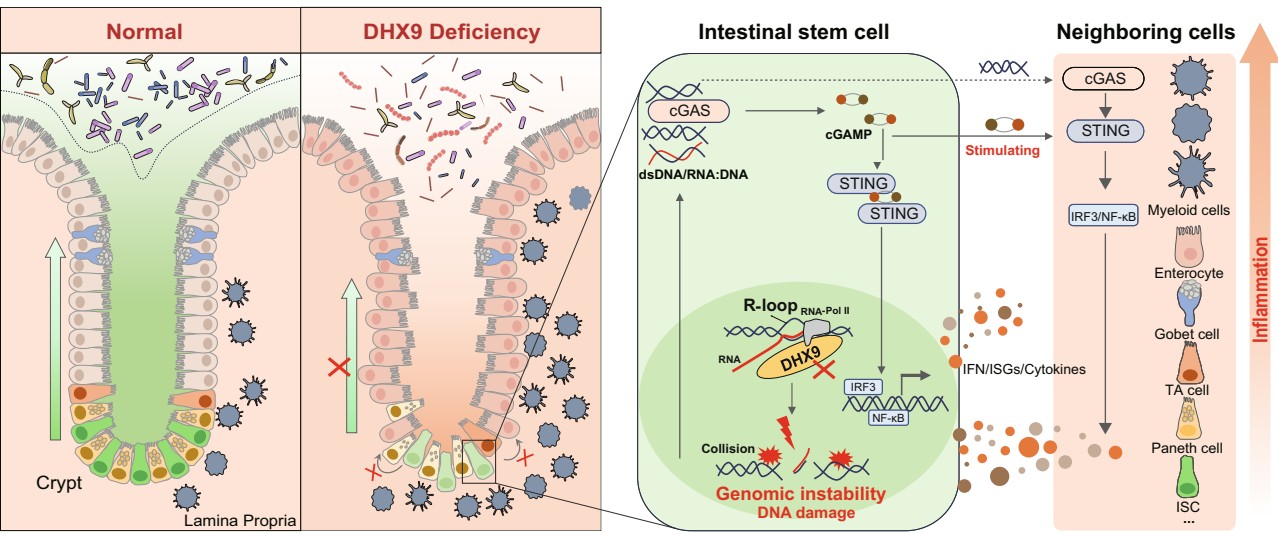

**Fig. 7 | Proposed model showing the mechanism underlying DHX9 maintains epithelial homeostasis and prevents IBD by restraining R-loop-mediated genomic instability in ISCs.** DHX9 deficiency in ISCs leads to the abnormal accumulation of R-loops, subsequently inducing genomic instability. This directly results in a reduction in the number of ISCs and influences secretory cell lineage commitment, thereby affecting intestinal homeostasis. DNA damage amplifies cGAS-dependent inflammatory responses in both IECs and myeloid cell populations. R-loop-mediated genomic instability, in conjunction with the triggered activation of the cGAS-STING pathway, impairs ISC function and contributes to the pathogenesis of IBD.

the intestinal epithelial barrier, leading to increased bacterial translocation and chronic inflammation[51,52]. Several genes have been identified as critical regulators of ISC function. For instance, ISC-specific knockout of SETDB1 in mice triggers the reactivation of endogenous retroviruses in the intestine, leading to ISC necroptosis and enteritis[12]. Deletion of Znhit1 in intestinal epithelium depletes Lgr5+ stem cells, thereby disrupting postnatal establishment and maintenance of intestinal homeostasis[53]. This raises the question of whether loss-of-function mutations in important IBD susceptibility genes, such as *NOD2*, *XIAP*, and *ATG16L*, could also lead to ISC dysfunction[54]. Therefore, our research emphasizes the critical role of ISC dysfunction as a primary driver of epithelial disruptions, thus implicating ISCs as a significant origin of intestinal inflammation.

Rapid ISC proliferation and differentiation are dependent on the stability of the transcriptional machinery, which is partially affected by R-loops homeostasis during transcription. Abnormal R-loop accumulation associated with the development of various diseases, including cancers[55], neurological diseases[56], and immune system disorders[57]. R-loop interaction proteomics have identified several RNA helicases that are thought to play important roles in maintaining R-loop stability, such as DDX5, DHX9, and DDX4[58-60]. The present study suggests that targeting R-loops, particularly by modulating DHX9 activity, may provide a potential therapeutic approach for treating these diseases. While we demonstrated that DHX9 deficiency in IECs leads to abnormal R-loops accumulation and genomic instability, the intricacies of this process merit further exploration. The relationship between R-loop accumulation and genomic instability is well-established[31]. Current theories about how DHX9 influences R-loop accumulation include its prolonged association with RNA Polymerase II in cells with impaired RNA splicing, leading to DNA-RNA hybrid formation that impacts DNA replication[61]. DHX9 also collaborates with TDRD3 and TOP3B in resolving R-loops at gene promoters[62]. ATR phosphorylation of DHX9 at serine 321 suppress R-loop accumulation under genotoxic stress[63]. Moreover, DHX9 deficiency affects genomic stability through various mechanisms beyond R-loop metabolism, including impacts on DNA replication and repair[29], impairment of ATR-mediated damage signaling[30], and interactions with γH2AX[64]. This complexity underscores the multifaceted roles of DHX9 in R-loops and genomic stability.

Genomic instability is associated with a wide range of pathological conditions, including cancer, aging, developmental disorders, and inflammatory diseases[65,66]. Research has established that the cGAS-STING pathway plays a key role in triggering cytokine production in response to DNA damage. Our rescue experiments demonstrated that blocking the cGAS-STING pathway partially rescues the colitis phenotype in *Dhx9*-deficient mice, supporting the idea that cGAS-STING activation contributes to colitis pathogenesis. These findings are consistent with previous studies demonstrating the role of cGAS-STING in inflammatory diseases, such as lupus, psoriasis, and neurodegenerative disorders[35]. Notably, DHX9 plays a pivotal role in both maintaining genomic stability and acting as a transcription co-factor. In steady-state conditions, DHX9 knockout precipitates R-loop-mediated genomic instability, triggering the activation of the cGAS-STING-interferon pathway and the consequent induction of ISGs. Intriguingly, under conditions of viral infection or direct interferon stimulation, DHX9 uniquely associates with the ISG promoter region, facilitating STAT1-mediated transcription of a of ISGs[67]. We think this difference is related to the intensity of the stimulus. This differential response may illuminate the observed nuances in ISG expression levels in macrophages, where unstimulated DHX9 knockout cells show marginally elevated ISG levels, in contrast to a markedly restrained ISG production when exposed to substantial interferon stimuli.

In summary, this study revealed a critical connection between DHX9, R-loops, and IBD. Through *Dhx9*^ΔIEC, *Dhx9*^ΔPaneth, and *Dhx9*^iΔISC mice and organoids, we demonstrated that DHX9 is crucial for maintaining intestinal homeostasis. Mechanistically, the loss of DHX9 leads to the abnormal R-loop accumulation, which induces genomic instability. DNA damage and the subsequent cGAS-STING-mediated inflammatory response contribute to ISC functional impairment and IBD development. Importantly, we identified a clinical correlation between DHX9 and R-loop levels in IBD, suggesting their potential as targets for clinical diagnosis and precision treatment. Furthermore, our research highlighting the dysfunction of ISCs as a key driver of epithelial disruptions and intestinal inflammation. These findings advance our understanding of IBD pathogenesis and provide potential avenues for targeted therapy. Further investigations are warranted to fully elucidate the complex interactions between R-loops, DNA

damage, inflammation, and stem cell dysfunction in disease development.

## Methods

### Ethics statement

All animal experiments were approved by the Ethics Committee of the University of Science and Technology of China. All mice were bred and maintained at accredited animal facilities under specific pathogen-free conditions on a strict 12-h light/12-h dark cycle with a regular chow diet. The collection and use of human paraffin-embedded specimens and intestinal biopsy specimens for this study were approved by the Ethical Committee of the First Affiliated Hospital of Anhui Medical University and the First Affiliated Hospital of University of Science and Technology of China. Informed consent was obtained from all human subjects enrolled in the study for biopsy sampling.

### Mice

*Dhx9*-floxed mice were generated as previously described[67]. *Villin-Cre* and *Lgr5-EGFP-IRES-Cre*[ERT2] mice were obtained from the Jackson Laboratory. The *Defa6-Cre* mice were kindly provided by Richard Blumberg (Harvard Medical School, Boston). The *Sting*[−/−] mice were kindly provided by Daxing Gao (University of Science and Technology of China, Hefei). *Apc*[min/+] mice, and Ai14 reporter mice (*Rosa26*[lsl-tdTomato]) were provided by Richard A. Flavell from Yale University. *Dhx9*[fl/fl] mice were crossed with *Villin-Cre* mice, *Lgr5-EGFP-IRES-cre*[ERT2] mice, or *Defa6-Cre* mice to generate *Dhx9* depletion in IECs (*Dhx9*[ΔIEC]), Paneth cells (*Dhx9*[ΔPaneth]) and Lgr5+ stem cells (*Dhx9*[ΔISC]) respectively. *Lgr5-EGFP-IRES-cre*[ERT2] mice were crossed them with *Rosa26*[lsl-tdTomato] to generate *Lgr5-EGFP-Cre*[ERT2]: Rosa26[lsl-tdTomato] mice. *Dhx9*[fl/fl] Villin-Cre mice (*Dhx9*[ΔIEC]) were crossed with *Sting*[−/−] mice to generate double-knockout mice. Primers sequence for mice genotype identification were provided in the supplementary materials (Supplementary Table 2). 6–12 week-old and sex-matched mice in C57BL/6 background were used in all assays.

### Human subjects

For western blot, the specimens from patients with IBD and health control samples were collected from the First Affiliated Hospital of University of Science and Technology of China (Hefei, China). Paraffin-embedded specimens for IHC from patients with IBD, as well as healthy controls, were sourced from the First Affiliated Hospital of Anhui Medical University, Hefei, China. Additionally, biopsy specimens of UC for RT-qPCR analysis were obtained from the same institution. The CD biopsy specimens used for RT-qPCR analysis were derived from our previous study[68]. Detailed information of patients was show included in Supplementary Table 1.

### DSS and TNBS-induced colitis model

TNBS-induce colitis was administered as described previously[22]. Briefly, 8–12 week-old male C57BL/6 mice were anesthetized using 1.5% isoflurane-mixed gas, and a 1.5 × 1.5-cm area of skin on the back of the mouse was shaved. Next, 150 μl of 1% TNBS (mix 4 volume of acetone/olive (4:1) oil with 1 volume of 5% (wt/vol) TNBS solution) pre-sensitization solution was applied to the shaved skin area. On day 8, the mice were anesthetized again, and 100 μl of 2.5% TNBS (mix 1 volume of 5% (wt/vol) TNBS solution in H2O with 1 volume of absolute ethanol) solution was slowly injected into the lumen of the colon. The control group was treated only with 50% ethanol. Mouse weight was monitored daily, and the mice were sacrificed 72 h after colitis induction. Colon tissues were collected for histological analysis.

DSS-induce colitis was established as previously, 8–12 week-old male C57BL/6 mice were fed with 2.5 % DSS (MP Biomedicals, molecular mass 36–50 kDa, 160110) in drinking water for 6 d, followed by normal drinking water until the end of the experiment. Mice were weighed daily and the fecal status was observed. After sacrifice of the mice, colon length was measured and colon tissue samples were collected for histology staining.

### Apc[Min/+] mouse model

All animal experimentation involving the *Apc*[Min/+] mouse model was conducted in accordance with the guidelines approved by the Institutional Animal Care and Use Committee of the University of Science and Technology of China, which mandated that tumors in the mice not exceed a volume of 2000 mm³. In this experiment, none of the tumor volumes exceed the maximal tumor size/burden.

*Dhx9*[fl/fl] Villin-Cre mice (*Dhx9*[ΔIEC]) were crossed with *Apc*[min/+] mice to generate mouse lines carrying the *Apc* mutation with or without specific deletion of *Dhx9* in IECs, referred to as *Dhx9*[fl/fl]*Apc*[min/+] and *Dhx9*[ΔIEC]*Apc*[min/+], respectively. Gender-matched *Dhx9*[fl/fl]*Apc*[min/+] mice and their *Dhx9*[ΔIEC]*Apc*[min/+] littermates were regularly monitored for signs of illness or distress and euthanized based on veterinarian recommendation or when they reached ~20 weeks of age. Intestinal tissues were collected post-euthanasia for tumor analysis.

### Histopathology and immunohistochemistry

For histopathology, intestinal samples were fixed in 10% formalin, embedded in paraffin, and sectioned into 4 μm thick slices. Hematoxylin and eosin (H&E) staining was performed to visualize tissue structures and identify pathological changes. Slides were examined under a light microscope (BX53, Olympus), and images were captured for subsequent analysis. The histopathological score for each section was referenced to previous studies[68,69]. Briefly, each sample was blindly scored for four criteria: amount of inflammation (0 = none, 1 = slight, 2 = moderate, 3 = severe), extent of inflammation (0 = none, 1 = mucosa, 2 = mucosa and submucosa, 3 = transmural), regeneration (0 = complete regeneration or normal tissue, 1 = almost complete regeneration, 2 = regeneration with crypt depletion, 3 = surface epithelium not intact, 4 = no tissue repair), crypt damage (0 = none, 1 = basal 1/3 damaged, 2 = Basal 2/3 damaged, 3 = only surface epithelium intact, 4 = entire crypt and epithelium lost). The overall pathology score was calculated as the sum of the individual scores for each criterion.

For immunohistochemistry (IHC), sections of the intestine were deparaffinized, rehydrated, and subjected to heat-induced epitope retrieval for antigen retrieval. Endogenous peroxidase activity was blocked using hydrogen peroxide, and nonspecific binding was prevented by blocking with serum or protein blocking reagents. The sections were then incubated overnight at 4 °C with specific antibodies targeting R-loops (ENH001, Kerafast, 1:200), γH2AX (GB111841, Servicebio, 1:200), or DHX9 (ab26271, Abcam, 1:200), Subsequently, HRP-conjugated secondary antibodies were applied, and the positive signals were detected using a DAB kit (DAB, Vector, Burlingame, CA). Finally, the sections were counterstained with hematoxylin for visualization.

### Immunofluorescent (IF) staining

The small intestine and colon tissues were isolated, fixed in 4% paraformaldehyde (PFA) overnight, and washed with PBS. Cryoprotection was performed by incubating the tissues in a 30% sucrose solution overnight. Afterwards, the tissues were embedded in optimal cutting temperature (OCT) compound and sectioned into 10 μm-thin slices. For IF staining, primary antibodies against Ki67 (GB111141, Servicebio, 1:500), Lysozyme (GB11345, Servicebio, 1:500), MUC2 (GB11344, Servicebio, 1:500), E-cadherin (GB12082, Servicebio, 1:500), DHX9 (ab26271, Abcam, 1:200), DHX9 (67153, Proteintech, 1:200), Cleaved Caspase-3 (9661, Cell Signaling Technology, 1:200) were used. The primary antibodies were diluted in 1% blocking buffer and incubated with the tissue sections overnight at 4 °C. Following PBS washes, the sections were incubated with fluorochrome-conjugated secondary antibodies for 1 h at room temperature. To visualize the nuclei, the

sections were counterstained with DAPI for 5 min. After additional washing steps, the sections were mounted with a mounting medium and coverslips. IF staining was examined using Zeiss LSM880. For the IF staining of *Lgr5-EGFP-Cre*[ERT2] mice, detailed methods are provided in the Supplementary methods section.

For R-loops staining in HeLa cells, the cells were plated on coverslips in 24-well plates and incubated overnight at 37 °C in a 5% CO2 incubator. Post the designated treatments, cells were fixed with 4% PFA for 30 min and blocked with 1% BSA in PBS for 30 min at room temperature. For permeabilization, cells were treated with 0.5% Triton X-100 and 1% BSA in PBS for 1 h at room temperature. The fixed cells were then incubated with the S9.6 (ENH001, Kerafast, 1:200) and Nucleolin (10556-1-AP, Proteintech, 1:200) primary antibodies diluted 1% in blocking buffer overnight at 4 °C. Following this, the cells were incubated with fluorochrome-conjugated secondary antibodies for 1 h at room temperature. Nuclei were visualized with DAPI. All fluorescence images were acquired and analyzed using confocal imaging, specifically Zeiss LSM880.

## Transmission electron microscopy (TEM)
For Transmission Electron Microscopy (TEM) analysis, small intestine samples were fixed overnight at 4 °C in 2.5% glutaraldehyde in 0.1 M sodium cacodylate buffer (pH 7.4). Subsequently, they were postfixed in 2% aqueous osmium tetraoxide, dehydrated through a graded series of ethanol (30-100%) and propylene oxide, and embedded in Epon resin. Ultrathin sections were then obtained from the embedded samples and counterstained with uranyl acetate and lead citrate. TEM images were acquired using a Tecnai G2 Spirit transmission electron microscope operating at an acceleration voltage of 120 kV.

## R-loop detection using Dot-Blot
Genomic DNA was extracted from ileum IECs of mice using the QIAmp DNA mini kit (QIAZEN, 51304) following the manufacturer's protocol. The concentration of DNA was determined using a spectro-photometer. The extracted DNA was then diluted to a standard concentration of 200 ng/µl and subjected to further serial dilutions for detailed analysis. As a control, an aliquot of the DNA was also diluted to 200 ng/µl in a reaction mixture containing 5 units of RNase H (M0297, New England Biolabs) and a reaction buffer with a water base, to a final volume of 20 µl, and then incubated in 37° for 20 min. This control setup was similarly serially diluted. Subsequently, 2 µl samples from each of these diluted DNA concentrations were carefully spotted onto a nitrocellulose membrane for further experimental processes. The membrane was air-dried for 1 h at room temperature and cross-linked using a UV cross-linker (1200 µJ x 100). To block nonspecific binding, the membrane was incubated with 5% nonfat dry milk in Tris-buffered saline with 0.1% Tween 20 (TBST) for 1 h at room temperature. Next, the membrane was incubated overnight at 4 °C with gentle shaking with an antibody specific to R-loops, S9.6 (ENH001, Kerafast, 1:200) antibody. After washing the membrane three times, it was incubated with a secondary antibody, goat anti-mouse IgG-HRP (Beyotime, A0216). R-loop signals were visualized using an enhanced chemilumi-nescence (ECL) substrate.

## Isolation of intestinal epithelial cell (IEC) and lamina propria cell (LPL)
The small intestine was carefully dissected and thoroughly flushed with ice-cold PBS to remove any luminal contents. The intestine was then turned inside out and cut into ~1 cm sections. To dissociate the epithelial cells, the tissue pieces were incubated in RPMI medium containing 2 mM EDTA at 37 °C for 20 min with shaking. Following the incubation, the supernatant containing the detached intestinal epi-thelial cells (IECs) was collected and passed through a 100 µm cell strainer to remove larger tissue fragments. The collected cells con-stituted the IEC fraction, which comprised ~90% epithelial cells and

10% lymphocytes (IEL). Single-cell suspensions were incubated at 4 °C with fluorescence-activated cell sorting (FACS) antibodies in PBS containing 0.5% fetal bovine serum (FBS) and 2 mM EDTA for 20 min. The cells were then sorted using a BD FACS Aria flow cytometer. CD45.2-EpCAM+ cells were specifically sorted as IECs.

To isolate lamina propria cells (LPLs), the bowel was subjected to three rounds of shaking, following described above method for IEC isolation. Subsequently, the supernatants were discarded, and tissue sections were collected and rinsed with RPMI (EDTA-free) for an additional 5 min. The intestinal sections were then transferred into a digestion buffer consisting of RPMI supplemented with 0.5 mg/mL collagenases II (Sigma C6885), 0.5 mg/mL DNase I (Sigma), and 5% FBS. The sections were subjected to shaking for 30 min at 37 °C. After the incubation, the supernatants containing the released lamina propria lymphocytes (LPLs) were collected by passing through a 70 µm cell strainer, resulting in the generation of single-cell suspesions.

## Isolation of crypts and culture of murine intestinal organoids
Murine intestinal crypts were isolated following a previously described protocol[70], with slight modifications. Briefly, 10-cm small intestines were dissected and flushed with ice-cold phosphate-buffered saline (PBS) to remove any remaining fecal matter. The tissue was then opened longitudinally and cut into small pieces. The tissue pieces were washed with ice-cold PBS multiple times to remove excess debris. Next, the tissue pieces were incubated in 10 mM EDTA in PBS at 4 °C for 30 min with gentle shaking. Afterward, the tissues were vigorously shaken to dislodge the cells, and the supernatant containing detached crypts was collected and passed through a 70 µm cell strainer to remove larger tissue fragments. The crypts were pelleted by centrifugation at 300 g for 2 min. Isolated crypts were resuspended in PBS and the crypts were counted. An appro-priate number of crypts was collected and centrifuged at 300 g for 2 min. The pellet was then resuspended in an appropriate volume of Organoid Growth Medium to achieve a concentration of 80 crypts/25 µl. Subsequently, a 25 µl mixture of crypts and Matrigel (356231, Corning) was distributed into each well of a 24-well plate. After the Matrigel polymerized at 37 °C and 5% CO2 for 30 min, the matrix was overlaid with 500 µl of Organoid Growth Medium (06005, STEM-CELL) containing 1 x penicillin-streptomycin (Invitrogen). The Orga-noid Growth Medium was changed every 2 days. The images of organoids were captured, and the number of buds was counted under an Olympus CKX53 microscope. For the flow cytometry experiment, Organoids were resuspended in 500 µl of TrypLE Express (Gibco), and the resuspended cells were incubated at 37 °C for 5 min. Cells were then pelleted after the addition of 500 µl of 1 x PBS containing 5% FBS and processed for flow cytometry.

## Induction of gene knockouts
To induce DHX9 knockout in *Dhx9*[fl//fl] *Lgr5-EGFP-Cre*[ERT2] mice, intra-peritoneal administration of tamoxifen (1 mg/mouse) (T5648, Sigma-Aldrich) or corn oil (control) was performed for consecutive 5 days. IECs were harvested for RT-qPCR analysis at 5 days and 7 days after the last tamoxifen injection. For DHX9 knockout induction in *Dhx9*[fl//fl] *Lgr5-EGFP-Cre*[ERT2] organoids, primary crypts were cultured for 24 h and then treated with 200 nM 4-OH (HY-16950, MedChemExpress) or EtOH (control) for 24 h before being removed.

## Flow cytometry analysis
For surface staining, single-cell suspensions were incubated with FACS antibodies in FACS buffer, which consisted of PBS with 2% BSA and 5 mM EDTA, for 20 min at 4 °C. The antibodies used for staining included APC/Cy7 anti-CD45.2 (Biolegend, clone 104), PE anti-Ep-CAM (Biolegend, clone G8.8), PE anti-CD11b (Biolegend, clone M1/70), FITC anti-Ly6G (Biolegend, clone 1A8), PerCP/Cy5.5 anti-Ly6C (Biolegend, clone HK1.4). Flow cytometric analyses were performed using a

cytometer (CytoFlex s, Beckman Coulter) and the acquired data were analyzed using FlowJo v.10.0.7 software.

## Western blot
Biopsy tissues or cells were collected and lysed using a lysis buffer containing 50 mM Tris-HCl (pH 7.4), 2 mM EDTA, 150 mM NaCl, 0.5% NP-40, phenylmethylsulfonyl fluoride (50 µg/ml), and complete protease inhibitor mixtures (Sigma-Aldrich). The lysates were incubated on ice for 30 min with intermittent vortexing to ensure complete cell lysis. Cellular debris was removed by centrifugation at 15,000 rpm for 10 min at 4 °C. The protein concentrations were measured using the BCA assay. Equal amounts of protein were loaded onto 12% SDS-PAGE gels and separated by electrophoresis. The proteins were then transferred from the gel to a PVDF membrane (Millipore). After blocking, primary antibody incubation, and secondary antibody incubation, the protein bands were visualized using an enhanced chemiluminescence reagent (Millipore). The band intensities were quantified using ImageJ. Antibodies specific for DHX9 (ab26271, Abcam, 1:2000) and γH2AX (GB111841, Servicebio, 1:200) were used.

## Neutral comet assay
The neutral comet assay was performed using the CometAssay Single Cell Gel Electrophoresis Assay kit (4250-050-K, R&D) following the manufacturer's instructions. In brief, cells were prepared at a concentration of $1 \times 10^5$ cells/ml and embedded in molten low-melting-point agarose (at 37 °C) at a ratio of 1:10 (v/v). Subsequently, 50 µl of the cell-agarose mixture was pipetted onto CometSlides. After solidification at 4 °C for 10 min, the slides were immersed in a lysis solution at 4 °C in the dark for 1 h. Electrophoresis was performed using a horizontal electrophoresis unit (Bio-Rad) with freshly prepared electrophoresis buffer at 4 °C for 45 min and 21 volts. Following electrophoresis, DNA staining was conducted using SYBR Green dye (Bio-Rad) for 15 min in the dark. Images of comets were captured using a Zeiss confocal microscope. The extent of DNA damage was assessed by measuring parameters such as comet tail length and the percentage of DNA in the tail.

## RNA isolation and RT-qPCR
Total RNA was isolated from cells or mouse tissues using the TRIzol reagent (Invitrogen) according to the manufacturer's instructions. For cDNA synthesis, 500 ng of total RNA was reverse transcribed using the HiScript III RT SuperMix kit (Vazyme, China). Quantitative reverse transcription polymerase chain reaction (RT-qPCR) was performed to analyze the expression levels of specific genes. The RT-qPCR reactions were conducted using the Bio-Rad CFX384 real-time PCR system with a 10 µl reaction volume containing SYBR Green qPCR Master Mix (Vazyme, China). The relative expression levels of the target genes were determined using the comparative CT (cycle threshold) method. The expression data were normalized to the internal control *Gapdh*. qPCR primers used in this study are provided in the Supplementary Materials (Supplementary Table 3).

## RNA-mediated interference
The siRNA targeting human DHX9 and the negative control siRNA (nontargeting siRNA) were sourced from GenePharma Company. HeLa cells were transfected with the siRNAs using Lipofectamine RNAiMAX Transfection Reagent (Invitrogen, 13778075). After 48 h, the cells were subjected to immunofluorescence analysis. siRNA sequences of DHX9: DHX9-homo-598 (GAGCCAACUU GAAGGAUUAT T, UAAUCCUUCA AGUUGGCUCT T), and DHX9-homo-2944 (CCUGGGAUGA UGCUUA-GAAUT T, AUUCUAGCAU CAUCCCAGGT T).

## RNA sequencing
RNA-seq libraries were prepared from IECs and sequenced using the Illumina 2500 platform. The sequencing procedure followed the manufacturer's protocols, generating raw sequencing data in FASTQ format for further analysis. The raw sequencing data underwent pre-processing to remove adapter sequences, low-quality reads, and potential contaminants. Trimmomatic (v0.39) was utilized for this purpose. Subsequently, the processed reads were aligned to the mouse reference genome (mm10) using STAR (v2.5.3a), and gene expression quantification was performed with HTSeq (0.11.0). The resulting read counts were normalized using the DESeq2 package. Differential expression analysis was conducted using edgeR. To gain insights into the biological implications of the differentially expressed genes, gene ontology enrichment analyses were performed utilizing the R package clusterProfiler (v4.0.5).

## Statistical analysis
Statistical analysis was conducted using GraphPad Prism 8.0 software. All data are expressed as means ± standard error of the mean (SEM). Group comparisons were analyzed using unpaired two-tailed Student's *t*-test or one-way/two-way analysis of variance (ANOVA). A significance level of $P < 0.05$ was considered statistically significant.

## Reporting summary
Further information on research design is available in the Nature Portfolio Reporting Summary linked to this article.

## Data availability
The RNA-seq, scRNA-seq, 16 S rRNA sequencing, and R-loop Cut&Tag data have been deposited in the NCBI Sequence Read Archive (SRA) BioProject (PRJNA989744, PRJNA1077865). The remaining data are available within the Article, Supplementary Information or Source Data file. Source data are provided with this paper.

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

## Acknowledgements

We thank Dr. Richard Blumberg for providing Defa6-Cre mice. We thank Dr. Daxing Gao for providing *Sting*[-/-] mice. We thank Dr. Xiong Ji (Peking University) for help with the CUT&Tag analysis. This work was supported by grants from the Key Research and Development Program of the Ministry of Science and Technology (MOST) (2023YFC2306203) (WP); the National Natural Science Foundation of China (82325025, 92374204, 82341121, 81821001) (SZ), (32170912, 82371828) (WP), (82370529, 82070565) (M. Li), (82200572) (XR); the CAS Project for Young Scientists in Basic Research (YSBR-074) (SZ and WP); and the China Postdoctoral Science Foundation (2021M700948) (XR). We also thank Long Li and SciDraw for the mouse illustrations.

## Author contributions

X.R., Q.L. and P.Z. performed most of the experiments. Q.M. provided clinical IBD samples. D.W. helped with animal experiments and cellular experiments. T.Z. analyzed the sequencing data. R.A.F. generated the *Dhx9* conditional KO mice. Z.L., M.L. and W.P. provided critical comments and suggestions. X.R. wrote the manuscript. S.Z., W.P. and M.L. supervised the project.

## Competing interests

S.Z. is a cofounder of Ibiome which studies microbial regulation of immune responses. All other authors declare that they have no competing interests.
