## [Peer Review File · Nature Communications]

DHX9 maintains epithelial homeostasis by restraining R-loop-mediated genomic instability in intestinal stem cellsEditorial Note: Parts of this Peer Review File have been redacted as indicated to remove third-party material where no permission to publish could be obtained.

REVIEWER COMMENTS

Reviewer #1 (Remarks to the Author):

In this manuscript, Ren et al. investigated the biological function of DHX9 in the context of intestinal epithelial homeostasis and inflammatory bowel disease. They showed that DHX9 loss in epithelial cells enhances DSS- and TNBS-induced colitis. Subsequent RNA-seq analysis indicated that down-regulated genes upon DHX9 loss are associated with those markers of ISCs and Paneth cells. The authors then generated DHX9 conditional knockout (cKO) in Paneth cells and in ISCs, which enables them to pinpoint the essential function of DHX9 in ISCs, but not in Paneth cells, which is more relevant to the observed phenotypes. Mechanistically, they showed that DHX9 loss in epithelial cells is associated with increased R-loops and DNA damage, as well as the activation of the cGAS-STING pathway and innate immune response.

Overall, this manuscript is well-written and easy to follow. The strength of this study is that the authors used various cKO mouse models to figure out the relevant cell population/type that is responsible for the phenotype. The weakness is that there is limited new knowledge revealed from this study. DHX9 is a multi-functional protein that is essential for the viability of many proliferating cells. It is not unexpected that DHX9 loss in ISCs will decrease its number because of increased cell death. The enthusiasm is further decreased due to the lack of mechanistic studies and the missing of important proper controls. The detailed critiques are listed below:

1. In Fig. 1a, & 1b, the level of DHX9 expression is much lower in UC and CD patients' intestinal tissues. IHC staining should be included to confirm these results. More importantly, what is the cause of this reduced DHX9 expression? Is it at the mRNA level or protein level? Understating this might help the future development of therapeutics.
2. In Fig. 2, bulk RNA-seq was performed in control and IEC-specific DHX9 cKO cells. The differential expressed genes identified here are indistinguishable among various cell types within the IECs. The authors should perform single-cell RNA-seq to identify DEGs in different epithelial subpopulations. This would allow the unbiased analysis of the impact of DHX9 loss on gene expression in different IEC cell types.
3. In Fig. 2f – 2i, ISC proliferation and differentiation was reduced in IEC-specific DHX9 KO. They should also test if there is more apoptosis of ISCs upon DHX9 KO. This question also applies to Fig. 3b and Fig. 4, in which ISC-specific DHX9 KO was generated. Basically, does DHX9 KO cause more apoptosis in ISCs?
4. In Fig. 4c, the quantification results are opposite to the flow cytometry plots.
5. In Fig. 5, the authors attribute the observed ISC defects caused by DHX9 KO to the increased R-loops and DNA damage. Although there seems to be an overall increase of genomic R-loops detected by dot-blot (Fig. 5d), the role of DHX9 in R-loop regulation is highly context-dependent. Knockdown DHX9 in HeLa cells using siRNA has been shown to reduce global R-loops (as detected by dot-blot), which is in contrast to the results that the authors showed in Fig. 5e (same HeLa cells, reference 57). How to explain this discrepancy? Others reported that DHX9 knockdown increases R-loops at gene promoters (Yuan et al. *Nucleic Acids Res.* 2021 Sep 7;49(15):8573-8591). To further link DHX9's function in ISCs with R-loops, the authors should compare the genome-wide R-loop

localization in control and DHX9 KO in ISCs or IECs to clearly define DHX9-regulated R-loops. This would also allow them to test if there are any connections between DEGs and R-loops caused by DHX9 KO.

6. One of the standard controls for R-loop detection is RNase H. This control is missing in both Fig. 5d and 5e. Also, in these two panels, the quantification results are opposite to the dot-blot (5d) and IF (5e). Additionally, the legend for 5e says this is in HeLa cells, whereas in the Figure, it says IEC. There are also some issues with S9.6 detection of R-loops in HeLa cells using IF staining (Fig. 5e). Nucleolus R-loops formed by rRNA transcription were not detected here. This should be a positive control for the R-loops detected by IF.

7. Fig. 5F, there is no data from this study to support how DHX9 loss leads to R-loop increase and DNA damage. This model is misleading because it implies that R-loops are the cause of increased DNA damage. Here, they only show that there are more R-loops and more H2AX phosphorylation.

8. Whether R-loop antibody S9.6 can be used for IHC has not been evaluated. The signal detected in Fig. 5g might not reflect the level of R-loops.

9. In Fig. 6e – 6i, the significance of the cGAS-STING pathway and innate immune response in ISC homeostasis is not clear. Is it responsible for the decreased ISC numbers? Or it is just a result of increased DNA damage? Sting KO should be included in these experiments for comparison. Although Sting KO in DHX9 KO background reduces IBD severity, does this have anything to do with ISC homeostasis? Will Sting KO rescue the ISC numbers and ability to form organoids?

Reviewer #2 (Remarks to the Author):

The manuscript submitted by Li, Pan, Zhu and team presents interesting functions attributed to DHX9 in the intestinal epithelium. Loss of DHX9 leads to activation of the cGAS-STING pathway and disruption in intestinal epithelial lineage counts. The knockout animals appear susceptible to DSS, TNBS, and tumorigenesis in the APC^{min} background. The author's use of the STING KO model to indicate the importance of the cGAS-STING pathway in the DHX9 KO IFN gene activation phenotype is a particularly nice experiment. This work presents interesting new findings, potentially linking R loop formation to loss of DHX9 and a potential trigger for IFN signaling. There are some conflicting gaps in the work, for example, the authors recently published in Science Advances that DHX9 is required for transcription of IFN-stimulated genes, yet in this paper they see increases in IFN-stimulated genes upon loss of DHX9. Still the intestine-specific knockout is quite interesting and should be published. The manuscript has significant areas that could be strengthened as detailed below.

What appears first, activation of the IFN pathway or the stem cell/Paneth cell disruptions? It is important to the model presented by the authors to know the order of these events. A timecourse after tamoxifen treatment might help resolve the order of events.

Do the organoids survive passage and retain DHX9 KO? Do they still activate IFN signaling target genes in organoid cultures in the presumed absence of the microbiome in organoid culture conditions? This experiment is important because the authors implicate a shift in the microbiome in their phenotype but it is uncertain whether the microbiome is important for the phenotypes observed. Doing these experiments in GF mice would be even better, but probably not practical.

Lgr5-Cre-ERT2+, DHX9 WT organoids should be used as controls when treating with 4-OH tamoxifen, as this hormone can be toxic to organoids and induce the cell death effects that are observed.

The result in figure 4 is surprising. There are not many Lgr5-CreERT2 marked cells in the colon. Therefore, I wouldn't expect a strong DSS-induced phenotype, as I wouldn't expect DHX9 to be substantially reduced in this model. For this result to be considered rigorously, the authors should confirm loss of DHX9 in situ, either using RNA FISH or IHC.

Similarly, the authors should confirm that DHX9 is actually lost in the Paneth-specific KO. Are any of the IFN-pathway genes activated in the Paneth-specific KO?

Are other aspects of the phenotype rescued in the STING;DHX9 double knockout model? For example are the Paneth cells still affected? Are other cell lineages still altered? Do R loops still form? Understanding which aspects of the phenotype can be attributed to STING activation would be an important mechanistic insight for the model proposed.

The model in S6 shows microbes penetrating the DHX9 KO, however there is no evidence to support this idea.

The authors should comment on their recent Science Advances paper that suggests DHX9 is important for transcription of ISGs and the alternative explanation that is now presented in this work. How do they think these mechanisms are both at play in the intestine?

Statistics and quantification of results should be based upon biological replicates rather than technical counts, such as in figure 2F-I, 3C, S4, and elsewhere.

The quantitation graphs in Fig. 5d-e don't match the data shown in the same figure panels.

Reviewer #3 (Remarks to the Author):

Ren et al. described the role of the RNA helicase DHX9 in the maintenance of intestinal epithelium. They showed that loss of DHX9 is associated with increased inflammation, reduced proliferation and differentiation of intestinal stem cells. Loss of DHX9 also promotes tumorigenesis in the mouse intestine. The findings are interesting and relevant to the clinics giving that reduced DHX9 is also found in human ulcerative colitis and Crohn's disease samples. That said, the authors may need to address a few issues as following,

1. Although the authors used Lgr5-CreER to conditionally delete DHX9 in the adult mice and showed that ISCs are affected, the majority of the experiments are done with the deletion mice using villin-cre (e.g. Fig5d). Use of Vil-cre to delete DHX9 can lead to developmental phenotypes which may have little relevance to ISCs. Therefore, the authors are suggested to analyze phenotypes using the Lgr5-CreER mouse line, for example, examination of R-Loop levels, cGAS-STING activation in the ISCs.

2. Regarding the phenotypes seen in villin-cre;Dhx9flox/flox, the authors are suggested to provide a comprehensive analysis of intestinal phenotypes at different developmental

stages. The goal is to provide information whether the intestinal epithelium is affected during early stage of development. When does the phenotypes start?

3. The authors may want to compare the intestinal phenotypes seen in vill-cre;dhx9flox/flox and Lgr5-creER;Dhx9fl/fl.

Minor issue:

The authors are suggested to provide high magnification images of Fig5g where rH2ax and R-loop IHC was shown. It appears the staining is not specific, especially Rloop staining.

Response to reviewers

We thank the reviewers for their constructive comments and valuable suggestions. During last 3 months, we have made substantial changes in our manuscript in accordance with the reviewers' suggestions. Our point-by-point responses to the reviewers' comments are listed below. Text changes are highlighted with **yellow background** in our revised manuscript.

REVIEWER COMMENTS

Reviewer #1 (Remarks to the Author):

In this manuscript, Ren et al. investigated the biological function of DHX9 in the context of intestinal epithelial homeostasis and inflammatory bowel disease. They showed that DHX9 loss in epithelial cells enhances DSS- and TNBS-induced colitis. Subsequent RNA-seq analysis indicated that down-regulated genes upon DHX9 loss are associated with those markers of ISCs and Paneth cells. The authors then generated DHX9 conditional knockout (cKO) in Paneth cells and in ISCs, which enables them to pinpoint the essential function of DHX9 in ISCs, but not in Paneth cells, which is more relevant to the observed phenotypes. Mechanistically, they showed that DHX9 loss in epithelial cells is associated with increased R-loops and DNA damage, as well as the activation of the cGAS-STING pathway and innate immune response.

Overall, this manuscript is well-written and easy to follow. The strength of this study is that the authors used various cKO mouse models to figure out the relevant cell population/type that is responsible for the phenotype. The weakness is that there is limited new knowledge revealed from this study. DHX9 is a multi-functional protein that is essential for the viability of many proliferating cells. It is not unexpected that DHX9 loss in ISCs will decrease its number because of increased cell death. The enthusiasm is further decreased due to the lack of mechanistic studies and the missing of important proper controls. The detailed critiques are listed below:

Response: First and foremost, we would like to express our sincere gratitude for your diligent work in reviewing our manuscript. Your professional insights and constructive suggestions have been invaluable in guiding our research to a higher standard of scientific inquiry and clarity. We are particularly thankful for your positive remarks on the strengths of our manuscript. Addressing the concerns you raised regarding the novelty, mechanistic depth, and control experiments in our study, we have conducted a thorough re-evaluation and additional experiments to enhance our manuscript significantly:

We summarize the following three points regarding the novelty issue. (1) DDX9 as a Guardian of Intestinal Epithelial Homeostasis: Our study elucidated the crucial role of DDX9 in intestinal epithelial homeostasis, especially in the context of IBD. This study is the first to unveil the role of DDX9 in IBD, supported by detailed evidence spanning from clinical IBD samples to mouse models, thereby enriching our understanding of the link between DDX9 and IBD; (2) Regulatory Role of DDX9 within ISCs and Its Crucial Impact on the Progression of IBD: Our study highlights the direct influence of DDX9 on ISC function, emphasizing its pivotal role in maintaining the health and stability of these cells. Furthermore, it underscores that dysregulated ISCs could be a central element in epithelial disruptions and a significant causative factor of intestinal inflammation in IBD; (3) R-Loop-Mediated Genomic Instability in ISCs and IBD: While there has been extensive research on R-loops, their association with autoimmune diseases, particularly IBD, remains largely unexplored. Our findings reveal the critical contribution of R-loop-mediated genomic instability in ISCs as a key pathogenic factor in IBD. This novel understanding could inspire further research into the pathological roles of R-loops and offer new perspectives for exploring therapeutic interventions in IBD.

Addressing the issue of Deepening Mechanistic Insight, we have included a series of additional experiments in our revised manuscript based on your insightful suggestions. These additions include single-cell RNA sequencing (scRNAseq) to analyze epithelial cellular heterogeneity, R-loop CUT&Tag to delineate the characteristics of R-loops in *Ddx9*^{ΔIEC} IECs, Western blot and RT-qPCR analysis to investigate the causes and mechanisms behind the reduction of DDX9, and IF staining to demonstrate the enhanced death of ISCs upon DDX9 loss, among other experimental enhancements. The results have significantly enhanced our understanding of the molecular mechanisms at play. Furthermore, in response to your point about Incorporating Appropriate Control Experiments, we have diligently incorporated additional controls in our study design. This includes the use of RNase H1 to verify the specificity of R-loop accumulation and Nucleolin as a comparative marker. Moreover, to solidify our findings in the cGAS-STING pathway analysis, we introduced *Sting* knockout controls. These amendments, we believe, comprehensively address your concerns and reinforce the validity of our study.

In response to your specific queries, we have carefully revised our manuscript to provide clearer explanations and justifications for our experimental design, data interpretation, and conclusions. We believe these revisions and additional experiments address your concerns comprehensively and enhance the scientific value of our work. We are confident that our manuscript now provides a more complete and nuanced understanding of the role of DDX9 in intestinal epithelium homeostasis and its implications in the pathogenesis of IBD.

Thank you again for your invaluable feedback and guidance.

1. In Fig. 1a, & 1b, the level of DHX9 expression is much lower in UC and CD patients' intestinal tissues. IHC staining should be included to confirm these results. More importantly, what is the cause of this reduced DHX9 expression? Is it at the mRNA level or protein level? Understating this might help the future development of therapeutics.

Response: Thank you for your constructive comments and suggestions. We have supplemented our study with DHX9 immunohistochemistry (IHC) staining in IBD patients. IHC results also revealed a significant reduction in DHX9 protein levels in the epithelial cells of UC and CD patients (**Supplementary Fig. 1a**), consistent with our WB results. Additionally, we investigated whether the reduction of DHX9 was at the mRNA or protein level. Analysis of *DHX9* mRNA levels in IBD patients showed no significant change in either UC or CD Patients (**Supplementary Fig. 1b**). Hence, we infer that the reduction of DHX9 is predominantly at the protein level.

More importantly, we explored the cause of this decreased protein level of DHX9. In our WB analysis of IBD samples, we observed distinct degradation bands for DHX9, particularly in UC samples (**Fig. 1a, b**), suggesting that protein cleavage might be involved in DHX9 degradation. Previous studies have highlighted increased Caspase-7 levels in IBD patients¹, and Caspase-7 is highly expressed in the IECs and plays unusual role in the extrusion of IEC². Our recent research also demonstrated that Caspase-3 and Caspase-7 were activated in IECs and have functions beyond apoptosis induction in the intestine³. Therefore, we investigated whether caspase family proteins mediate the cleavage and degradation of DHX9. Co-expression of Flag-tagged DHX9 with Caspase-1, Caspase-3, and Caspase-7 respectively in 293T cells revealed a noticeable cleavage band specifically when Caspase-7 was co-expressed with Flag-DHX9 (**Supplementary Fig. 1c**). This led us to hypothesize that Caspase-7 is responsible for DHX9 degradation. Subsequently, we incubated purified Caspase-3 and Caspase-7 proteins with Flag-DHX9, and found that Flag-DHX9 showed significant cleavage bands upon incubation with Caspase-7 (**Supplementary Fig. 1d**). Moreover, both mouse DHX9 and human DHX9 exhibited increased cleavage over time when incubated with Caspase-7 (**Supplementary Fig. 1e, f**). Therefore, we hypothesize that Caspase-7-mediated cleavage may be the major cause of DHX9 cleavage and subsequent degradation. Our future research will delve deeper into the specific mechanisms and implications of Caspase-7-mediated cleavage of DHX9, particularly in the context of IBD, potentially uncovering new insights into the pathogenesis and progression of this disease.

The new data can be found in **Supplementary Fig. 1**, and the description of the results are added in the text **line 104-122**.

Supplementary Fig. 1

Figure S1. Caspase-7-mediated specific cleavage of DHX9 protein.

a, Representative images DHX9 IHC in human IBD (UC and CD) and healthy control (HC) intestinal samples (n = 10 per group), Quantifications are provided on the right. **b**, Left panel: *DHX9* expression in patients with UC (n = 10) and HC (n = 8). Right panel: *DHX9* expression in patients with CD (n = 19) and HC (n = 10). **c**, Co-expression of empty vector (EV), Caspase-1, Caspase-3, and Caspase-7 plasmids with Flag-tagged DHX9 in 293T cells. At 24 hours post-transfection, immunoprecipitated with anti-Flag beads and Western blot analysis using anti-Flag antibodies. Black arrows denote the cleavage fragments. **d**, Flag-DHX9 was overexpressed in 293T cells, immunoprecipitated, and incubated with Caspase-3, Caspase-7, or negative control (NC), with subsequent detection by Western blot using anti-Flag antibodies. The black arrows highlight the cleavage fragments. **e**, Time-dependent cleavage of overexpressed Flag-DHX9 by Caspase-7, analyzed via Western blot with anti-Flag antibodies. Black arrows mark the cleaved products. **f**, Time-dependent cleavage of overexpressed human DHX9 (hDHX9) by Caspase-7, analyzed via Western blot with anti-Flag antibodies. Black arrows mark the cleaved products. Data represent mean \pm s.e.m. Statistical significance was assessed with a two-tailed unpaired Student's t-test. *P < 0.05, **P < 0.01; ns = not significant.

2. In Fig. 2, bulk RNA-seq was performed in control and IEC-specific DHX9 cKO cells. The differential expressed genes identified here are indistinguishable among various cell types within the IECs. The authors should perform single-cell RNA-seq to identify DEGs in different epithelial subpopulations. This would allow the unbiased analysis of the impact of DHX9 loss on gene expression in different IEC cell types.

Response: Thank you for your insightful suggestion. Following your recommendation, we conducted single-cell RNA sequencing (scRNA-seq) on upper small intestine cells isolated from two-month-old *Dhx9^{ΔIEC}* mice and their *Dhx9^{fl/fl}* littermates. This analysis encompassed 30252 cells, with 12108 from the *Dhx9^{fl/fl}* group and 18144 of *Dhx9^{ΔIEC}* group. Unsupervised clustering revealed 15 distinct clusters, each distinguished by unique gene expression patterns (Fig. 2e). A comparative analysis between the *Dhx9^{fl/fl}* and *Dhx9^{ΔIEC}* group cells highlighted a notable difference in cellular composition. Specifically, in the *Dhx9^{ΔIEC}* group, there was a significant decrease in the proportion of ISCs, Paneth cells, goblet cells, and EECs, along with an expansion of TA cells (Fig. 2f). These observations are consistent with our RNA-seq data and imply that DHX9 deficiency impacts epithelial secretory cell lineages.

We further performed differential expression analysis on these epithelial subpopulations, including ISCs, Paneth cells, Tuft cells, and goblet cells. Gene Ontology analysis indicated that the most significantly up-regulated pathways in *Dhx9^{ΔIEC}* IECs subpopulations were related to antigen presentation, response to interferon, and immune response. In contrast, the primary downregulated pathways were related to ATP metabolic processes, response to unfolded proteins, and mRNA processing (Supplementary Fig. 4). These patterns suggest that DHX9 deficiency disrupts cellular metabolism and essential cellular functions, which may account for the reduced proliferation of these cell populations. Our results thus provide a more nuanced understanding of how DHX9 loss impacts gene expression across different IEC cell types, offering valuable insights into the pathophysiological consequences of this alteration in the context of intestinal epithelium.

The new data can be found in Fig 2e, f, Supplementary Fig. 4, and the text line 165-182.

Fig. 2. Ablation of DHX9 results in aberrant ISCs and intestinal secretory cells.

e, Single-cell RNA sequencing (scRNA-seq) of IECs from *Dhx9^{ΔIEC}* and their *Dhx9^{fl/fl}* littermates. Visualized as a Uniform Manifold Approximation and Projection (UMAP) plot, categorizing cells from the upper small intestine into 15 distinct clusters based on their gene expression profiles. **f**, Graphical representation of the proportion of each identified cell cluster relative to the total population in the analyzed samples.

Supplementary Fig. 4

a

b

c

d

Figure S4. Gene Ontology analysis of downregulated and upregulated genes in IEC Subsets of $Dhx9^{AIEC}$ mice. Gene ontology enrichment analysis of genes showing downregulated and upregulated in ISC, Paneth cells, TA cells, Tuft cells, and goblet cells in $Dhx9^{AIEC}$ mice relative to $Dhx9^{fl/fl}$ controls. The analysis is based on single-cell RNA sequencing (scRNA-seq) data with statistical significance set at $P < 0.05$, determined by right-tailed Fisher's exact test.

3. In Fig. 2f – 2i, ISC proliferation and differentiation was reduced in IEC-specific DHX9 KO. They should also test if there is more apoptosis of ISCs upon DHX9 KO. This question also applies to Fig. 3b and Fig. 4, in which ISC-specific DHX9 KO was generated. Basically, does DHX9 KO cause more apoptosis in ISCs?

Response: Thank you for your insightful question. According to your suggestion, we initiated apoptosis detection in the intestines of DHX9-IEC-specific knockout mice ($Dhx9^{AIEC}$) using the terminal deoxynucleotidyl transferase-mediated deoxyuridine triphosphate nick end labeling (TUNEL) assay. Our findings indicated a higher incidence of TUNEL-positive cells in the intestines of $Dhx9^{AIEC}$ mice, especially within the crypt regions (**Supplementary Fig. 5a**). Additionally, the proportion of cleaved Caspase-3–positive cells was significantly higher in the $Dhx9^{AIEC}$ group compared to their $Dhx9^{fl/fl}$ littermates, again predominantly in the crypt areas (**Supplementary Fig. 5b**). These results suggest that DHX9 plays a crucial role in maintaining IEC homeostasis.

To further investigate if DHX9 KO leads to increased apoptosis in ISCs, we conducted cell death assays using organoids derived from $Dhx9^{AIEC}$ and their $Dhx9^{fl/fl}$ littermates. Propidium iodide (PI)

staining conformed abundant cell death in $Dhx9^{\Delta IEC}$ organoids (Fig. 3c). These results clearly indicates that DHX9 KO induces increased apoptosis in ISCs.

For a more precise exploration of the impact of DHX9 loss on ISCs apoptosis, we performed TUNEL assays on the intestines of DHX9-ISC-specific KO mice ($Dhx9^{i\Delta ISC}$). The intestines of tamoxifen-treated $Dhx9^{i\Delta ISC}$ mice exhibited a significant increase in both TUNEL-positive and caspase-3–positive cells compared to corn oil-injected $Dhx9^{i\Delta ISC}$ mice (Supplementary Fig. 14a, b), particularly in the crypt areas specifically lacking DHX9 (EGFP positive). This suggests a crucial role for DHX9 in the survival and maintenance of ISCs, and DHX9 KO leads to increased apoptosis in ISCs.

Additionally, we cultured $Dhx9^{\Delta ISC}$ organoids and treated them with 4-hydroxytamoxifen (4-OH) or vehicle control (ethanol, EtOH) to induce efficient deletion of DHX9 in ISCs. After 48 hours, apoptosis in these organoids was assessed using PI staining. Our findings showed a significant increase in apoptosis following the induced knockout of DHX9 in ISCs, while 4-OH had no impact on the growth of $Dhx9^{fl/fl}$ organoids. This suggests that the observed apoptosis in $Dhx9^{\Delta ISC}$ organoids is specifically attributable to the loss of DHX9, thereby supporting the conclusion that DHX9 KO leads to increased apoptosis in ISCs.

The new data can be found in Supplementary Fig. 5, Fig. 3c, Supplementary Fig. 14a, b, Supplementary Fig. 12, and in the text line 192-198, line 230-331, line 306-309, line 293-297.

Supplementary Fig. 5

Figure S5. Apoptotic markers in the intestine of $Dhx9^{\Delta IEC}$ mice.

a, TUNEL assay staining of ileum sections from $Dhx9^{fl/fl}$ and $Dhx9^{\Delta IEC}$ mice at 6 weeks of age to detect DNA fragmentation indicative of apoptosis. Representative images are shown with quantification on the right. $n = 8$ per genotype. TUNEL-

positive cells are indicated with arrows. **b**, IF staining for cleaved caspase-3 on ileum sections from *Dhx9^{fl/fl}* and *Dhx9^{ΔIEC}* mice at 6 weeks of age. Representative images are shown with quantification on the right. $n = 8$ per genotype. Cells positive for cleaved caspase-3 are marked with arrows. Data represent mean \pm s.e.m. Statistical analysis was performed using two-tailed unpaired Student's t test. * $P < 0.05$, ** $P < 0.01$, *** $P < 0.001$, and **** $P < 0.0001$.

Fig. 3c

Fig. 3. DHX9 deficiency impairs intestinal crypt and organoid formation.

c, Propidium iodide (PI) staining of organoids derived from *Dhx9^{fl/fl}* and *Dhx9^{ΔIEC}* mice. The panel to the right of the images provides a statistical analysis, quantifying the relative area of PI staining per organoid. At least 15 organoids were counted. Three individual experiments were performed, with similar results. The panel to the right of the images provides a statistical analysis, quantifying the relative area of PI staining per organoid.

Supplementary Fig. 14

Figure S14. Apoptotic markers in the intestine of *Dhx9^{iAISC}* mice.

a, TUNEL assay staining of ileum sections from corn oil-injected (control) and tamoxifen-treated $Dhx9^{i\Delta ISC}$ mice at 8 weeks of age to detect DNA fragmentation indicative of apoptosis. Representative images are shown with quantification on the right. $n = 8$ per genotype. TUNEL-positive cells are indicated with arrows. **b**, IF staining for cleaved caspase-3 on ileum sections from corn oil-injected (control) and tamoxifen-treated $Dhx9^{i\Delta ISC}$ mice at 8 weeks of age. Representative images are shown with quantification on the right. $n = 8$ per genotype. Cells positive for cleaved caspase-3 are marked with arrows. All data are presented as mean \pm s.e.m. Statistical analysis was performed using a two-tailed unpaired Student's t test. * $P < 0.05$, ** $P < 0.01$, *** $P < 0.001$.

Supplementary Fig. 12

Figure S12. 4-OH induced DHX9 knockout increased the mortality of $Dhx9^{i\Delta ISC}$ organoids, but did not affect the growth of $Dhx9^{fl/fl}$ organoids. Representative images of $Dhx9^{i\Delta ISC}$ and $Dhx9^{fl/fl}$ ileum organoids, cultured for 5 days. Organoids were treated with EtOH as a control or with 4-OH (200 nM) to induce DHX9 knockout for 24 hours. Propidium iodide (PI) staining traces cell mortality. The right panel provides statistical analysis of the relative PI area per organoid, indicating cell death. At least 15 organoids were counted. Three individual experiments were performed, with similar results. EGFP is derived from endogenous fluorescence. Scale bars represent 500 μ m. All data are presented as mean \pm s.e.m. Statistical significance was assessed with a two-tailed unpaired Student's t-test. * $P < 0.05$, ** $P < 0.01$, *** $P < 0.001$, and **** $P < 0.0001$.; ns = not significant.

4. In Fig. 4c, the quantification results are opposite to the flow cytometry plots.

Response: We apologize for the error. The figure has been revised accordingly to accurately reflect the correct data.

5. In Fig. 5, the authors attribute the observed ISC defects caused by DHX9 KO to the increased R-loops and DNA damage. Although there seems to be an overall increase of genomic R-loops detected by dot-blot (Fig. 5d), the role of DHX9 in R-loop regulation is highly context-dependent. Knockdown DHX9 in HeLa cells using siRNA has been shown to reduce global R-loops (as detected by dot-blot), which is in contrast to the results that the authors showed in Fig. 5e (same HeLa cells, reference 57). How to explain this discrepancy? Others reported that DHX9 knockdown increases R-loops at gene promoters (Yuan et al. *Nucleic Acids Res.* 2021 Sep 7;49(15):8573-8591).

To further link DHX9's function in ISCs with R-loops, the authors should compare the genome-wide R-loop localization in control and DHX9 KO in ISCs or IECs to clearly define DHX9-regulated R-loops. This would also allow them to test if there are any connections between DEGs and R-loops caused by DHX9 KO.

Response: Thank you for your insightful question, which highlights the current complexity and variability in the literature regarding the role of DHX9 in R-loop regulation. We have thoroughly reviewed the impact of DHX9 deficiency on R-loop levels under steady-state conditions (without stimulation) and found that the results are indeed inconsistent. For instance, *Liu et al.*'s study demonstrated that interfering with DHX9 in HeLa cells without any stimulation did not significantly change R-loop levels⁴. In contrast, *Cristini et al.* found a significant decrease in R-loop levels upon DHX9 interference in HeLa cells.⁵ However, *Huang et al.* reported a significant increase in R-loops under steady-state conditions after DHX9 interference⁶. In other cell lines, *Suzuki et al.* indicated a significant increase in R-loops in U2OS 2-6-3 cells upon DHX9 disruption⁷. *Patel et al.* observed that R-loop levels significantly increased in MDAMB-436 cells transfected with mutant DHX9⁸. Furthermore, *Yuan et al.* reported an increase in R-loops at gene promoters following DHX9 knockdown⁹.

We believe these inconsistent results could be attributed to several factors: (1) Inherent limitations of R-loop detection methods. We are aware that R-loop staining results in substantial nonspecific staining in the cytoplasm, an issue that current S9.6 antibodies cannot resolve. This nonspecific cytoplasmic staining could affect different detection methods to varying degrees. (2) Efficiency of siRNA interference. It is well-known that during interference, some DHX9 continues to be expressed normally. Whether this residual normal DHX9 expression is sufficient to maintain normal R-loop function remains unknown. (3) Baseline R-loop levels in cells. Detecting low-level R-loops can be challenging when the baseline R-loop levels in cells are already low. For example, although different studies show inconsistent changes in R-loop levels upon DHX9 interference under unstimulated conditions, these studies consistently demonstrate that R-loop levels significantly increase after DHX9 interference when cells are stimulated with CPT^{4,5}.

Our results, which are inconsistent with *Cristini et al.* but consistent with *Huang et al.*, might be due to differences in culture conditions (potentially leading to different baseline R-loop levels) and detection methods. Both our study and Huang et al.'s used immunofluorescence to detect R-loops in HeLa cells, whereas *Cristini et al.* employed the slot blot method. These are our speculations, and we believe that as R-loop detection methods improve, these issues can be more accurately addressed in the future.

[Redacted]

To further elucidate the link between DHX9's role in ISCs and R-loops, we examined the distribution and characteristics of R-loops in both *Dhx9^{fl/fl}* and *Dhx9^{ΔIEC}* IECs using the R-loop CUT&Tag method^{10,11}. Our analysis demonstrated a pronounced increase in R-loop reads in *Dhx9^{ΔIEC}* IECs. Genome-wide localization studies of R-loops revealed a more uniform distribution in *Dhx9^{fl/fl}* IECs, predominantly within the promoter regions (upstream 5 kb). In contrast, *Dhx9^{ΔIEC}* IECs exhibited a significant increase in R-loops in intronic and intergenic areas (**Supplementary Fig. 16b**). This shift in R-loop localization in the absence of DHX9 suggests potential extensive effects on cellular

functions. Additionally, we conducted a Gene Ontology analysis of genes associated with enriched peaks. In *Dhx9^{fl/fl}* IECs, enriched pathways were primarily related to metabolic processes. However, in *Dhx9^{ΔIEC}* IECs, enriched pathways were notably linked to system development, regulation of cellular component organization, and multicellular organismal development (**Supplementary Fig. 16c**). These findings imply that the absence of DHX9 significantly affects the genomic distribution of R-loops. However, it is important to note that the significantly changed genes in R-loops do not overlap with those in RNA-seq. Therefore, we think that the effect of DHX9 deficiency on IECs is primarily exerted through promoting the accumulation of global R-loops and genomic instability.

The new data can be found in **Supplementary Fig. 16**, and the text **line 354-365**.

Supplementary Fig. 16

Figure S16. R-loop CUT&Tag analysis in *Dhx9^{ΔIEC}* IECs.

a, Heatmaps depicting the CUT&Tag-seq signal distribution around the transcription start sites (TSS) within a 2.5 kb range, both upstream and downstream. Signals are shown for S.96 antibody in *Dhx9^{ΔIEC}* IECs, S.96 antibody in *Dhx9^{fl/fl}* IECs,

and IgG control in *Dhx9^{fl/fl}* IECs. **b**, Genomic localization of S9.6-binding sites in *Dhx9^{fl/fl}* and *Dhx9^{ΔIEC}* IECs, as revealed by CUT&Tag-Seq data. The annotations highlight the distribution of these sites across different genomic regions. **c**, Gene ontology enrichment analysis of genes associated with R-loop peaks identified in both *Dhx9^{fl/fl}* and *Dhx9^{ΔIEC}* IECs, providing insights into the functional implications of these genomic alterations.

6. One of the standard controls for R-loop detection is RNase H1. This control is missing in both Fig. 5d and 5e. Also, in these two panels, the quantification results are opposite to the dot-blot (5d) and IF (5e). Additionally, the legend for 5e says this is in HeLa cells, whereas in the Figure, it says IEC. There are also some issues with S9.6 detection of R-loops in HeLa cells using IF staining (Fig. 5e). Nucleolus R-loops formed by rRNA transcription were not detected here. This should be a positive control for the R-loops detected by IF.

Response: Thank you for pointing out these critical aspects. We have thoroughly revised the experiments concerning R-loop detection to include the RNase H1 control and Nucleolin staining for more rigorous validation.

We isolated IECs from *Dhx9^{ΔIEC}* mice and their littermates, and we quantified global R-loop levels using dot blot analysis with S9.6 antibody that specifically recognizes RNA:DNA hybrids in a sequence-independent manner^{12,13}. Indeed, the absence of DHX9 resulted in a marked increase in the overall abundance of R-loops, which were sensitive to RNase H1 treatment (**Fig. 5d, e**). To further confirm this, we adopted an alternative approach to detect the accumulation of nuclear R-loops. HeLa cells stably expressing both ZsGreen and RNase H1 on the same plasmid were constructed, with RNase H1 expression controlled by a doxycycline-inducible promoter. We then employed siRNA transfection to efficiently silence DHX9 in these HeLa cells and assessed R-loop formation using indirect IF microscopy. Notably, the nuclei of cells transfected with DHX9 siRNA displayed significantly stronger RNase H1-sensitive fluorescent signals compared to the control group, indicating an accumulation of R-loops in DHX9-deficient cells (**Fig. 5f, g**). These results indicate that loss of DHX9 causes R-loops accumulation.

The new data can be found in **Fig. 5 d, e, f**, and the text **line 340-348**.

Fig. 5

Fig. 5. DHX9 deficiency leads to R-loop accumulation and genomic instability.

d, Dot blot analysis to quantify R-loops in IECs from 8-week-old *Dhx9^{ΔIEC}* mice and their *Dhx9^{fl/fl}* littermates. Equal

amounts of DNA from each group were spotted onto a nitrocellulose membrane for analysis. Detection of R-loops was performed using the S9.6 antibody. RNase H1 treatment was included as a negative control to confirm the specificity of the R-loop signal. **e**, Quantitative analysis of the dot blot results from panel **d**. **f**, IF staining of HeLa cells stably transfected with a doxycycline (Dox)-inducible RNase H1 expression plasmid, which concurrently expresses zsGreen for identification purposes. These cells underwent treatment with specified siRNAs for 48 hours and were either induced with Dox for 24 hours or left uninduced. Staining includes the S9.6 antibody (red) to detect R-loops and the nucleolin antibody (cyan) for nucleolar identification. Cell nuclei are counterstained with DAPI (blue). The scale bar is set at 10 μ m. At least 20 cells were counted. Three individual experiments were performed, with similar results. **g**, Upper panel: WB analysis of Flag-RNase H1 and Actin in lysates from Dox+ and Dox- HeLa cells from panel **f**. Lower panel: Quantitative analysis of the dot blot results from panel **f**.

7. Fig. 5F, there is no data from this study to support how DHX9 loss leads to R-loop increase and DNA damage. This model is misleading because it implies that R-loops are the cause of increased DNA damage. Here, they only show that there are more R-loops and more H2AX phosphorylation.

Response: Thank you for your insightful observations regarding Fig. 5F. we acknowledge that our initial results might not have comprehensively demonstrated the causal relationship between DHX9 loss, R-loop accumulation, and subsequent DNA damage. However, integrating current research with our findings offers a more detailed perspective. We have now supplemented our response with additional insights and references to address this gap. Considering these insights, we have removed Fig. 5F from the current submission. We have also discussed the relationship between DHX9 loss, R-loop accumulation, and DNA damage in the discussion section, incorporating relevant literature to provide a clearer and more comprehensive understanding of these mechanisms (**line 509-520**).

- 1. DHX9 Deficiency Leading to R-loop Accumulation:** DHX9 plays a critical role in both the formation and resolution of R-loops¹⁴. Its strand-annealing activity facilitates the hybridization of RNA to DNA, contributing to R-loop formation, while its unwinding activity helps dissociate RNA-DNA hybrids, resolving R-loops¹⁵. In cells lacking sufficient splicing factors, prolonged association of DHX9 with RNA Polymerase II leads to DNA-RNA hybrids formation, impacting DNA replication¹⁶. This is notably significant in diseases characterized by deregulated RNA processing. DHX9 helicase also promotes R-loop suppression and transcriptional termination. DHX9 interacts with PARP1, and both proteins prevent R-loop-associated DNA damage⁵. Furthermore, DHX9, in a complex with TDRD3 and TOP3B, plays a role in resolving R-loops at gene promoters, crucial for maintaining genomic stability⁹. Recently study found that ATR phosphorylates DHX9 at serine 321 to suppress R-loop accumulation upon genotoxic stress. Inhibition of ATR or expression of the non-phosphorylatable DHX9S321A prevents DHX9 from interacting with RPA and R-loops, leading to the accumulation of stress-induced R-loops⁴.
- 2. R-loop Accumulation Inducing DNA Damage:** The accumulation of R-loops can lead to DNA damage and genomic instability through several mechanisms. R-loops can create transcription-replication conflicts, resulting in replication fork stalling and collapse, which

cause DNA double-strand breaks and genomic instability¹⁷. The formation of R-loops exposes single-stranded DNA (ssDNA), which is more susceptible to damage, mutations, or forming DSBs¹⁸. R-loops also hinder the normal functioning of DNA repair processes, leading to an accumulation of unrepaired DNA damage. Persistent R-loops can activate DNA damage response pathways, potentially causing cell cycle arrest, senescence, or apoptosis¹⁹. Knockdown of DHX9 inactivates the PI3K-AKT and ATR-Chk1 signaling pathways, promotes R-loop accumulation, and leads to R-loop-mediated DNA damage⁶. Furthermore, R-loops can induce chromosomal rearrangements and aneuploidy, particularly in rapidly dividing cells. The ssDNA regions in R-loops are prone to mutations, leading to increased genomic instability²⁰.

- 3. Alternative Pathways through which DHX9 Loss Impacts DNA Damage:** The absence of DHX9 affects DNA integrity through various mechanisms. Affects DNA replication and repair: DHX9 binds to single-stranded DNA and RNA, unwinding abnormal nucleic acid structures during DNA replication and repair. This function is vital for maintaining DNA integrity^{21,22}. Impairment of ATR-mediated damage signalling: DHX9 defects lead to impaired ATR-mediated damage signaling and an inability to restart DNA replication at camptothecin-induced DSBs²³. Further evidence supporting a role in DNA repair and maintenance of genomic stability is demonstrated by evidence that DHX9 is phosphorylated by DNA-PK, a major player in NHEJ-mediated DNA repair²⁴. DHX9 was found to interact with γ H2AX *via* its helicase core domain, and this association is significantly increased upon actinomycin D treatment, where DHX9 accumulates in RNA-containing nuclear bodies adjacent to γ H2AX foci²⁵. Upon DNA damage, DHX9 also localizes to promyelocytic leukaemia (PML) nuclear bodies, which are involved in the DNA damage response²⁶. In summary, DHX9 loss leads to DNA damage primarily through its role in unwinding abnormal DNA and RNA structures, maintaining the stability of R-loops, and regulating crucial signaling pathways involved in DNA repair and cell survival.

8. Whether R-loop antibody S9.6 can be used for IHC has not been evaluated. The signal detected in Fig. 5g might not reflect the level of R-loops.

Response: Thank you for your insightful observation regarding the use of the S9.6 antibody in IHC for R-loop detection. The use of the S9.6 antibody in IHC for R-loop detection is indeed a point of debate. While S9.6 is widely used in various assays to detect RNA-DNA hybrids, its specificity and effectiveness in IHC settings require careful consideration, even though there have been many studies using IHC to evaluate the level of R-loops²⁷⁻²⁹. Recognizing this, we have moved the results of Fig. 5g to supplementary material and emphasized "It is important to acknowledge that when employing staining methods to detect R-Loops, a significant number of positive signals are also observed in the cytoplasm. This occurrence may potentially impact the precision of R-loop detection results." in the results.

9. In Fig. 6e – 6i, the significance of the cGAS-STING pathway and innate immune response in ISC homeostasis is not clear. Is it responsible for the decreased ISC numbers? Or it is just a result of

increased DNA damage? Sting KO should be included in these experiments for comparison. Although Sting KO in DHX9 KO background reduces IBD severity, does this have anything to do with ISC homeostasis? Will Sting KO rescue the ISC numbers and ability to form organoids?

Response: Thank you for your insightful questions regarding the cGAS-STING pathway's significance in ISC homeostasis. Our results indicated that the deficiency of DHX9 in epithelial cells, particularly in ISCs, leads to DNA damage, which is a primary cause of intestinal ISC dysfunction. Concurrently, this DNA damage activates the cGAS-STING pathway in epithelial and myeloid cells, inducing inflammatory responses like interferon and NF- κ B pathways. These inflammatory factors from epithelial and myeloid cells further exacerbate ISC damage and death. Thus, DHX9 deficiency leads to abnormal R-loops accumulation and genomic instability, resulting in cGAS-STING-mediated inflammatory responses that impair ISC homeostasis. DNA damage caused by DHX9 deficiency is a critical factor in the dysfunction of ISCs, and the cGAS-STING pathway mediated inflammatory response accelerate this process.

The role of the cGAS-STING-IFN pathway in intestinal epithelial homeostasis has been extensively studied, revealing several key aspects:

- (1) Activation of cGAS-STING in IBD:** Both UC and CD patients with active disease exhibit activation of the cGAS-STING pathway. Increased levels of cGAS, p-STING, p-TBK1, and p-IRF3 proteins were observed in the colon of UC patients during active disease compared to those in remission and healthy controls³⁰. STING expression was higher in inflamed regions of UC patients' colons compared to healthy controls and non-inflamed areas³¹. A significant increase in STING, TBK1, IRF3, IFNB1 gene levels in active UC, indicating activation of the STING-TBK1-IRF3 axis³². In CD patients, elevated levels of p-STING, p-IRF3, and p-p65 proteins were detected in colon tissues during active disease phases³³.
- (2) cGAS-STING Activation Accelerates Intestinal Epithelium Damage:** Mice with constitutively active STING (*Sting*^{+/^{N153S}}) showed symptoms like reduced body weight, colon shortening, diarrhea, rectal bleeding, crypt abscesses, erosion, hyperplasia, ulceration, and loss of goblet cells³¹. Hence, activation of cGAS-STING accelerates intestinal epithelial injury. In contrast, mice lacking *Atg16ll* IECs developed spontaneous ileitis, resembling ileal CD. In these mice, cell death was enhanced in *Atg16ll* ^{Δ IEC} organoids treated with IFN- β or IL-22, which activate STING signaling. Blocking the IFN receptor improved disease severity in these models³⁴. *Sting* deficiency in Il-10^{-/-} mice, a model for chronic enterocolitis, reduced inflammation severity and pro-inflammatory cytokine expression³⁵.
- (3) cGAS-STING-IFN Pathway Accelerates Genomic Instability and Cell Death: Apoptosis Activation:** IFN induces apoptosis through extrinsic signaling, intrinsic mitochondrial pathways, and stress kinase signaling³⁶⁻³⁸. **Upregulation of Pro-Cell Death Molecules:** Type I IFNs increase pro-apoptotic molecules like BAK and TRAIL, enhancing apoptosis^{39,40,41}. **Induction of ISGs:** IFNs stimulate ISG transcription, encoding proteins that interfere with viral replication, modulate immune responses, or induce apoptosis⁴². **Production of ROS:** IFNs can induce ROS, leading to oxidative DNA damage like single-strand breaks and base modifications^{43,44}.

Influence on DNA Repair: IFNs may affect DNA repair protein expression/activity, potentially contributing to genomic instability⁴⁵. **Cell Cycle Modulation:** IFNs can disrupt the cell cycle by altering the expression of cell cycle-related proteins, potentially leading to DNA damage or genomic instability⁴⁶.

Overall, aberrant activation of the cGAS-Sting-IFN pathway aggravated intestinal inflammation and promoted IECs damage, which is consistent with our results.

Additionally, our experiments in Fig. 6e-6i supplemented *Sting*^{-/-} mice for comparison. Compared with *Dhx9*^{ΔIEC} littermates, DSS-treated *Dhx9*^{ΔIEC}*Sting*^{-/-} mice demonstrated colitis attenuation, as indicated by reduced weight loss (**Fig. 6g**), less colon length shortening (**Fig. 6h**), and less inflammatory cell infiltration and crypt damage (**Fig. 6i**). These results suggest that blocking the cGAS-STING pathway partially ameliorates the colitis phenotype in *Dhx9*^{ΔIEC} mice. It is noteworthy that even in a *Sting* knockout background, the additional deletion of DHX9 still led to a markedly more severe DSS colitis, as evidenced by the comparison between *Dhx9*^{ΔIEC} *Sting*^{-/-} mice and *Sting*^{-/-} mice (**Fig. 6f to i**). These results suggest that the cellular damage caused by DHX9 deficiency is a critical factor in the dysfunction of ISCs, and the cGAS-STING pathway mediated inflammatory response accelerate this process.

Furthermore, As *Dhx9*^{ΔIEC} *Sting*^{-/-} mice can mitigate the inflammatory response caused by DNA damage, we investigated whether this could also reverse the reduction in ISCs and epithelial secretory cell types. Our RT-qPCR analysis indicated that, compared to *Dhx9*^{ΔIEC} mice, the *Dhx9*^{ΔIEC} *Sting*^{-/-} mice showed a partial restoration in the proportions of ISCs and goblet cells in the intestine. However, there was no change observed in the populations of Paneth cells and EECs (**Supplementary Fig. 21a**).

Furthermore, *Sting*^{-/-} could not rescue the ability to form organoids suggesting that genome stability plays a key role in nascent organoids formation (**Supplementary Fig. 21c**). These findings suggest that while the additional knockout of *Sting* in DHX9-deficient IECs has some impact, it is not sufficient to completely restore normal ISC numbers and function.

DHX9 deficiency led to abnormal R-loop accumulation, resulting in genomic instability and the cGAS-STING-mediated inflammatory response, which together impaired ISC homeostasis.

The new data can be found in **Fig. 6e to i, Supplementary Fig. 21**, and the description of the results are added in the text **line 430-450**.

Fig. 6. DHX9 deficiency promotes cGAS-STING dependent inflammation.

e, RT-qPCR analysis of downstream genes of cGAS-STING pathway (including *Oasl2*, *Isg15*, *Ifit1*, *Il-18*, *Cxcl10*, and *Tnfaip2*) expression in the ileum of *Dhx9^{fl/fl}*, *Dhx9^{ΔIEC}*, *Dhx9^{ΔIEC} Sting^{-/-}*, and *Sting^{-/-}* mice (n = 6 per group) at 8 weeks of age. **f**, Experimental design depicting the induction of colitis using DSS. *Dhx9^{fl/fl}*, *Dhx9^{ΔIEC}*, *Dhx9^{ΔIEC} Sting^{-/-}* and *Sting^{-/-}* mice were treated with 2.5% DSS for 6 days. **g**, Monitoring of body weight changes throughout the experimental period. **h**, Measurement of colon length upon sacrifice on day 10, with quantifications provided on the right. **i**, Representative H&E staining of colon sections and corresponding histological scores (right). All data are presented as mean ± s.e.m. Statistical analysis were performed using two-way ANOVA (g) and two-tailed unpaired Student's t test. *P < 0.05, **P < 0.01, ***P < 0.001, and ****P < 0.0001.

Supplementary Fig. 21

Figure S21. IEC subpopulation dynamics analysis in *Dhx9^{ΔIEC} Sting^{-/-}* mice.

a, RT-qPCR analysis the expression of *Lgr5*, *Defa22*, *Muc2*, and *Chga* in IECs from *Dhx9^{ΔIEC}* mice (n = 3), *Dhx9^{fl/fl}* mice (n = 3), and *Dhx9^{ΔIEC} Sting^{-/-}* mice (n = 3) at 8 weeks of age. **b**, Dot blot analysis to quantify R-loops in *Dhx9^{ΔIEC}*, *Dhx9^{fl/fl}*, and *Dhx9^{ΔIEC} Sting^{-/-}* IECs. Equal amounts of DNA were spotted onto a nitrocellulose membrane, and R-loops were detected using the S9.6 antibody. Methylene blue (MB) staining served as the loading control. Quantifications are provided on the right. All data are presented as mean ± s.e.m. **c**, PI staining of organoids derived from *Dhx9^{fl/fl}*, *Dhx9^{ΔIEC}*, and *Dhx9^{ΔIEC} Sting^{-/-}* mice. At least 15 organoids were counted. Three individual experiments were performed, with similar results. The panel to the right of the images provides a statistical analysis, quantifying the relative area of PI staining per organoid. Statistical analysis was performed using a two-tailed unpaired Student's t test. *P < 0.05, **P < 0.01, ***P < 0.001, ****P < 0.0001. ns, not significant.

Reviewer #2 (Remarks to the Author):

The manuscript submitted by Li, Pan, Zhu and team presents interesting functions attributed to DHX9 in the intestinal epithelium. Loss of DHX9 leads to activation of the cGAS-STING pathway and disruption in intestinal epithelial lineage counts. The knockout animals appear susceptible to DSS, TNBS, and tumorigenesis in the APC^{min} background. The author's use of the STING KO model to indicate the importance of the cGAS-STING pathway in the DHX9 KO IFN gene activation phenotype is a particularly nice experiment. This work presents interesting new findings, potentially linking R loop formation to loss of DHX9 and a potential trigger for IFN signaling. There are some conflicting gaps in the work, for example, the authors recently published in Science Advances that DHX9 is required for transcription of IFN-stimulated genes, yet in this paper they see increases in IFN-stimulated genes upon loss of DHX9. Still the intestine-specific knockout is quite interesting and should be published. The manuscript has significant areas that could be strengthened as detailed below.

Response: Thank you for your thoughtful and positive assessment of our research. We greatly appreciate your constructive comments and feedback. Your insights have been instrumental in guiding us to improve the manuscript. We have taken your suggestions to heart and have made extensive revisions accordingly. Your invaluable contributions have greatly enhanced the overall quality of our work.

1. What appears first, activation of the IFN pathway or the stem cell/Paneth cell disruptions? It is important to the model presented by the authors to know the order of these events. A timecourse after tamoxifen treatment might help resolve the order of events.

Response: Thank you for your insightful question. Our data indicate that the loss of DHX9 leads to aberrant accumulation of R-loops and consequent genomic instability, accompanied by an aberrant activation of the intestinal inflammatory response. We posit that the trigger for the abnormal activation of the interferon pathway is this genomic instability. DHX9 plays a crucial role in maintaining genomic stability, and the accumulation of extensive DNA breaks and R-loops leads to cellular damage. This damage activates the cGAS-STING pathway, which in turn impairs ISC function and contributes to the pathogenesis of IBD.

According to your suggestion, we executed a time-course study following tamoxifen treatment in *Dhx9*^{ΔISC} and control *Dhx9*^{fl/fl} mice. Our findings indicate a reduction in the levels of *Ang4* and *Lgr5* as early as day 2 post-treatment, suggesting an early impact on the proliferation of ISCs and Paneth cells. However, at this stage, there were no notable differences in the levels of cGAS-STING downstream targets *Isg15* and *Oasl2* (**Supplementary Fig. 18**). On day 3, we observed a marked elevation in the interferon-stimulated genes *Isg15* and *Oasl2* in *Dhx9*^{ΔISC} mice.

These findings suggest that the disruptions in ISC and Paneth cell functions precede the activation of the IFN pathway, aligning with our model that genomic instability triggers inflammatory responses. We believe this additional data provides a clearer understanding of the sequence of events and further

supports our proposed model.

The new data can be found in **Supplementary Fig. 18**, and the description of the results are added in the text **line 403-409**.

Supplementary Fig. 18

Figure S18. time-course study post-tamoxifen treatment in *Dhx9^{ΔISC}* mice.

a, Schematic diagram illustrating the strategy of tamoxifen administration. *Dhx9^{ΔISC}* mice were injected with tamoxifen and subsequently sacrificed at 1, 2, and 3 days post-injection. **b**, RT-qPCR analysis the expression of *Lgr5*, *Ang4*, *Isg15*, and *Oas12* in IECs from *Dhx9^{ΔISC}* mice (n = 3) and their *Dhx9^{fl/fl}* littermates (n = 3) at 8 weeks of age. All data are presented as mean ± s.e.m. Statistical analysis was performed using a two-tailed unpaired Student's t test. *P < 0.05, **P < 0.01, ***P < 0.001. ns, not significant.

2. Do the organoids survive passage and retain DHX9 KO? Do they still activate IFN signaling target genes in organoid cultures in the presumed absence of the microbiome in organoid culture conditions? This experiment is important because the authors implicate a shift in the microbiome in their phenotype but it is uncertain whether the microbiome is important for the phenotypes observed. Doing these experiments in GF mice would be even better, but probably not practical.

Response: Thank you for your question. To address whether the organoids survive passage and retain DHX9 KO, we isolated crypts from the intestine and cultured them into organoids. On the 6th day, an equal number of organoids were passaged, and this process was repeated for two generations. At each generation, we sampled and analyzed the changes in DHX9 mRNA levels. Our findings revealed that as the passage number increased, both the budding rate and growth speed of the *Dhx9^{ΔIEC}* organoids gradually increased (**Fig. 3e, f**). Correspondingly, we observed a progressive recovery in DHX9 mRNA levels (**Fig. 3g**). This pattern suggests that DHX9 is vital for the growth of organoids.

Next, we examined whether IFN signaling is activated in organoid systems where microbes are absent. Our results showed that the expression of interferon-inducible genes was still stronger in *Dhx9^{ΔIEC}* than in *Dhx9^{fl/fl}* organoids, albeit lower, indicating that DHX9 deficiency-driven DNA

damage can independently activate interferon pathway in IECs.

Furthermore, as you rightly pointed out, conducting experiments in germ-free (GF) mice would ideally eliminate microbial interference and clarify DHX9's function. However, under our current conditions, this is challenging to implement. To elucidate the role of microbiota in our model, we employed a co-housing approach for enteritis studies in mice. We observed that the DSS colitis phenotype was significantly mitigated in *Dhx9*^{ΔIEC} mice co-housed with *Dhx9*^{fl/fl} mice, compared to singly housed *Dhx9*^{ΔIEC} mice. These results suggested that the compromised regulation of commensal microbiota, attributable to DHX9 deficiency, plays a contributory role in the development of the DSS-induced colitis (**Supplementary Fig. 7a to d**). Moreover, when comparing co-housed *Dhx9*^{ΔIEC} mice with co-housed *Dhx9*^{fl/fl} mice, the colitis was more severe in *Dhx9*^{ΔIEC} mice group. This suggests that, even when microbial influences are excluded, the accumulation of R-loops and genomic instability due to DHX9 deficiency remains a primary cause of exacerbated colitis. The resultant dysbiosis and inflammatory response play a secondary role.

The new data can be found in **Fig3 e, f, g, Supplementary Fig. 7a to d**, and the description of the results are added in the text **line 235-240, line 209-212**.

Fig. 3. DHX9 deficiency impairs intestinal crypt and organoid formation.

e, The organoids were cultured for 6 days and underwent two serial passages, with images captured for each generation to illustrate morphological changes. **f**, Analysis of the budding rate in each generation of the organoids described in panel **e**. **g**, Quantitative analysis of *Dhx9* mRNA levels in the original crypts and across each generation of organoids as described in panel **e**. All data are presented as mean \pm s.e.m. Statistical analysis for (**f**) and (**g**) were performed using two-tailed unpaired Student's *t* test **P* < 0.05, ***P* < 0.01, ****P* < 0.001, and *****P* < 0.0001; ns, not significant.

Figure. The expression of ISGs in DHX9-deficient organoids.

Supplementary Fig. 7

Figure S7. Co-house treatment partially alleviates colitis in *Dhx9*^{ΔIEC} mice.

a, Experimental design depicting the induction of colitis using DSS. Co-house *Dhx9*^{fl/fl} (n = 5), Co-house *Dhx9*^{ΔIEC} (n = 5), and Single-house *Dhx9*^{ΔIEC} (n = 5) mice were treated with 2.5% DSS for 6 days. b, Monitoring of body weight changes throughout the experimental period. c, Measurement of colon length upon sacrifice on day 10, with quantifications provided on the right. d, Representative H&E staining of colon sections and corresponding histological scores (right). All data are presented as mean ± s.e.m. Statistical analysis were performed using two-way ANOVA (b) and two-tailed unpaired Student's t test. *P < 0.05, **P < 0.01, ***P < 0.001, and ****P < 0.0001.

3. *Lgr5*-Cre-ERT2+, DHX9 WT organoids should be used as controls when treating with 4-OH tamoxifen, as this hormone can be toxic to organoids and induce the cell death effects that are observed.

Response: Thanks for your suggestions. To ensure that the observed organoid cell death in our experiments was a result of DHX9 knockout induced by 4-OHT and not due to the inherent toxicity of

4-OHT itself, we conducted additional control experiments using $Dhx9^{fl/fl}$ organoids. Our findings indicate that a concentration of 200 nM 4-OHT does not affect the growth of $Dhx9^{fl/fl}$ organoids, as evidenced by the lack of increased cell death (PI positive) in these organoids. However, cell death of $Dhx9^{i\Delta ISC}$ organoids was significantly increased. This observation confirms that 200 nM 4-OHT does not adversely impact the normal growth of organoids. In our study, this concentration was selected based on extensive literature review and is consistent with the concentrations used in similar studies for inducing Cre expression in organoids⁴⁷.

Furthermore, *in vitro* cytotoxicity studies suggest that 4-OH concentrations inducing cell death typically exceed $10 \mu\text{M}$ ^{48,49}, which is 50 times higher than the concentration we used for our organoid cultures. This significant difference in concentration provides a reasonable explanation for why 200 nM 4-OHT does not negatively affect our organoid cultures. These additional experiments and considerations strongly support our conclusion that the cell death observed in $Dhx9^{i\Delta ISC}$ organoids is specifically due to the knockout of DHX9 and not a result of 4-OHT toxicity at the concentration used in our study. The new data can be found in **Supplementary Fig. 12**, and text line 293-298.

Supplementary Fig. 12

Figure S12. 4-OH induced DHX9 knockout increased the mortality of *Dhx9*^{ΔISC} organoids, but did not affect the growth of *Dhx9*^{fl/fl} organoids. Representative images of *Dhx9*^{ΔISC} and *Dhx9*^{fl/fl} ileum organoids, cultured for 5 days. Organoids were treated with EtOH as a control or with 4-OH (200 nM) to induce DHX9 knockout for 24 hours. Propidium iodide (PI) staining traces cell mortality. The right panel provides statistical analysis of the relative PI area per organoid, indicating cell death. At least 15 organoids were counted. Three individual experiments were performed, with similar results. EGFP is derived from endogenous fluorescence. Scale bars represent 500 μm. All data are presented as mean ± s.e.m. Statistical significance was assessed with a two-tailed unpaired Student's t-test. *P < 0.05, **P < 0.01, ***P < 0.001, and ****P < 0.0001.; ns = not significant.

4. The result in figure 4 is surprising. There are not many *Lgr5*-CreERT2 marked cells in the colon. Therefore, I wouldn't expect a strong DSS-induced phenotype, as I wouldn't expect DHX9 to be substantially reduced in this model. For this result to be considered rigorously, the authors should confirm loss of DHX9 in situ, either using RNA FISH or IHC.

Response: Thank you for your insightful question regarding the knockout efficiency in our *Lgr5-EGFP-Cre*^{ERT2} mice and the surprising result in Figure 4. We understand your concern about the extent of DHX9 knockout in these mice and its potential impact on the DSS-induced phenotype.

In our study, we observed that approximately 40% of the crypts in normal *Lgr5-EGFP-Cre*^{ERT2} mice expressed EGFP fluorescence, indicating that at least 40% of the crypts' ISCs could undergo DHX9 knockout. It's important to note that some crypts not visibly expressing EGFP might still express Cre, as the fluorescence intensity of EGFP can be relatively weak after intestinal processing and may fall below the detection threshold. We have updated **Figure 4b** to include a larger field of view, allowing for better visualization of more EGFP-positive crypts.

To assess the knockout efficiency in *Lgr5-EGFP-IRES-cre*^{ERT2} mice, we crossed them with Ai14 reporter mice (*Rosa26*^{lsl-tdTomato}). In *Lgr5-EGFP-Cre*^{ERT2}:*Rosa26*^{lsl-tdTomato} mice, once Cre is expressed, the fluorescence of tdTomato is activated (**Supplementary Fig. 11a**). We observed that almost all EGFP-positive crypts exhibited red fluorescence throughout the villi after tamoxifen treatment, indicating that the *Lgr5*-driven gene knockout in ISCs was maintained as these cells differentiated into various IEC subpopulations (**Supplementary Fig. 11b**). Furthermore, imaging of the entire intestine in *Lgr5-EGFP-Cre*^{ERT2}:*Rosa26*^{lsl-tdTomato} mice six days after tamoxifen treatment revealed that approximately 40% of the villi were red (**Supplementary Fig. 11c**). This finding suggests that in *Dhx9*^{ΔISC} mice, about 40% of the epithelial cells exhibited DHX9 knockout.

Finally, we employed IHC to detect the knockout of DHX9 in the intestines of *Dhx9*^{fl/fl} *Lgr5-EGFP-Cre*^{ERT2} mice, particularly within ISCs. We observed a significant reduction or absence of DHX9 expression in most ISCs (**Supplementary Fig. 10**). This confirms that DHX9 is effectively knocked out in ISCs. Therefore, these additional experiments substantiate that the DSS colitis phenotype observed in *Dhx9*^{fl/fl} *Lgr5-EGFP-Cre*^{ERT2} mice is plausible.

The new data can be found in **Fig4 b, Supplementary Fig. 11, Supplementary Fig. 10**, and the description of the results are added in the text **line 247-285**.

b**Supplementary Fig. 11****a****b****c**
Figure S11. The knockout efficiency of *Lgr5-EGFP-Cre^{ERT2}* was tracked by tdTomato fluorescence.

a, Diagram of the *loxP*-flanked STOP cassette upstream of tdTomato with and without Cre recombination in *Rosa26^{lsl}-tdTomato* mice. **b**, Representative image of EGFP and tdTomato in ileum and colon sections from *Lgr5-EGFP-Cre^{ERT2}; Rosa26^{lsl}-tdTomato* mice, which were treated with tamoxifen for 5 and 6 days at 8 weeks of age. EGFP and tdTomato are derived from endogenous fluorescence. Nuclei are stained with DAPI (blue). Scale bars represent 50 μ m. **c**, Representative image of whole intestine from *Lgr5-EGFP-Cre^{ERT2}; Rosa26^{lsl}-tdTomato* mice, which were treated with tamoxifen for 6 days at 8 weeks of age. tdTomato are derived from endogenous fluorescence. Nuclei are stained with DAPI (blue). $n = 4$ per group. Data represent mean \pm s.e.m. Statistical analysis was performed using two-tailed unpaired Student's t test. **** $P < 0.0001$.

Supplementary Fig. 10

Figure S10. Specific knockdown of DHX9 in ISCs of the intestines in *Dhx9^{ΔISC}* mice.

Representative immunostaining of Epcam (white), DHX9 (red) in ileum sections from *Dhx9^{fl/fl}* and *Dhx9^{ΔISC}* mice, which were treated with tamoxifen for 6 days at 8 weeks of age to induce specific gene knockdown. EGFP is derived from endogenous fluorescence. Nuclei are stained with DAPI (blue). Scale bars represent 50 μ m. At least of 18 ISCs were counted. Experiment was performed on $n = 3$ mice individually, with similar results. Quantifications are provided on the right. Data represent mean \pm s.e.m. Statistical analysis were performed using two-tailed unpaired Student's t test. **** $P < 0.0001$.

5. Similarly, the authors should confirm that DHX9 is actually lost in the Paneth-specific KO. Are any of the IFN-pathway genes activated in the Paneth-specific KO?

Response: Thank you for your questions regarding the specificity and effects of DHX9 knockout in Paneth cells, and its impact on the IFN-pathway.

To validate the specific knockout of DHX9 in Paneth cells, we employed IF staining. We used Lysozyme to mark Paneth cells (green fluorescence) and an antibody against DHX9 to detect its expression (red fluorescence). Our results showed a noticeable reduction or absence of red fluorescence in lysozyme-positive cells in *Dhx9*^{ΔPaneth} mice (**Supplementary Fig. 8**), while other cells remained unaffected. This confirms that DHX9 can be specifically knocked out in Paneth cells of *Dhx9*^{ΔPaneth} mice.

Regarding the activation of IFN-pathway genes in the *Dhx9*^{ΔPaneth} mice, we examined the expression of interferon downstream genes in both *Dhx9*^{ΔPaneth} mice and their *Dhx9*^{fl/fl} littermate controls. We found no significant differences, indicating that the specific knockout of DHX9 in Paneth cells does not affect the overall activation of the IFN-pathway in the epithelium (**Supplementary Fig. 9**). We hypothesize that this could be due to the relatively small number of Paneth cells compared to the entire IEC population. The deficiency caused by the absence of DHX9 in Paneth cells under steady-state conditions may not be sufficient to impact the entire epithelial system. In contrast, abnormal ISCs rapidly differentiate into various epithelial cell groups, leading to anomalies in the entire IEC population. This could explain why DHX9 knockout in ISCs presents a more pronounced phenotype.

The new data can be found in **Supplementary Fig. 8**, **Supplementary Fig. 9**, and the description of the results are added in the text **line 254-258**.

Supplementary Fig. 8

Representative immunostaining of Lysozyme (green) and DHX9 (red) in ileum sections from *Dhx9^{fl/fl}* and *Dhx9^{ΔPaneth}* mice at 8 weeks of age. Nuclei are stained with DAPI (blue). Experiment was performed on n = 3 mice individually, with similar results. Mean Fluorescence Intensity (MFI). Scale bars represent 50 μm. Quantifications are provided on the right. Data represent mean ± s.e.m. Statistical analysis were performed using two-tailed unpaired Student's t test. ****P < 0.0001.

Supplementary Fig. 9

Figure S9. Paneth-cell-specific DHX9 signaling is dispensable for DSS-induced colitis.

a, RT-qPCR analysis of marker gene expression for epithelial cells and interferon stimulating genes in IECs from *Dhx9^{ΔPaneth}* mice (n = 6) and their *Dhx9^{fl/fl}* littermates (n = 6) at 8 weeks of age.

6. Are other aspects of the phenotype rescued in the STING;DHX9 double knockout model? For example are the Paneth cells still affected? Are other cell lineages still altered? Do R loops still form? Understanding which aspects of the phenotype can be attributed to STING activation would be an important mechanistic insight for the model proposed.

Response: Thank you very much for your question. Our study posits that the loss of DHX9 in epithelial cells, particularly in ISCs, leads to aberrant R-loop formation, which in turn causes DNA damage. This is a primary cause of the anomalies observed in intestinal ISCs. Concurrently, this DNA damage activates the cGAS-STING pathway in epithelial and mononuclear cells, triggering an aberrant inflammatory response through interferon and NF-κB pathways. The inflammatory cytokines secreted by these cells further stimulate ISCs, exacerbating cell damage and death. Hence, DHX9 deficiency leads to abnormal R-loop accumulation, resulting in genomic instability and a cGAS-STING-mediated inflammatory response, which together impair ISC homeostasis.

The *Dhx9^{ΔIEC}Sting^{-/-}* double knockout model primarily mitigates the inflammatory response caused by DNA damage and can only partially rescue the phenotype resulting from DHX9 knockout. Our DSS colitis results have already substantiated this. To specifically address whether the Paneth cells are still affected and if other cell lineages remain altered, we examined various IEC lineages. Our RT-qPCR analysis indicated that, compared to *Dhx9^{ΔIEC}* mice, the *Dhx9^{ΔIEC}Sting^{-/-}* mice showed a partial restoration in the proportions of ISCs and goblet cells in the intestine. However, there was no change

observed in the populations of Paneth cells and IECs (**Supplementary Fig. 21a**). These findings suggest that additional knockout of *Sting* in DHX9 deficiency IECs can alleviate intestinal epithelial abnormalities, but its effectiveness is still limited.

Additionally, we assessed the presence of R-loops and found that the *Dhx9*^{ΔIEC}*Sting*^{-/-} double knockout did not reverse the aberrant accumulation of R-loops (**Supplementary Fig. 21b**). This aligns with our hypothesis that the double knockout cannot fundamentally resolve the R-loop accumulation and genomic instability caused by DHX9 deficiency. These findings provide important mechanistic insights into the role of STING activation in the phenotype induced by DHX9 loss and highlight the complex interplay between genomic instability, inflammatory pathways, and cellular homeostasis in the intestine.

The new data can be found in **Supplementary Fig. 21**, and the description of the results are added in the text **line 436-441**.

Supplementary Fig. 21

Figure S21. IEC subpopulation dynamics and R-loop analysis in *Dhx9*^{ΔIEC} *Sting*^{-/-} mice.

a, RT-qPCR analysis the expression of *Lgr5*, *Defa22*, *Muc2*, and *Chga* in IECs from *Dhx9*^{ΔIEC} mice (n = 3), *Dhx9*^{fl/fl} mice (n = 3), and *Dhx9*^{ΔIEC}*Sting*^{-/-} mice (n = 3) at 8 weeks of age. **b**, Dot blot analysis to quantify R-loops in *Dhx9*^{ΔIEC}, *Dhx9*^{fl/fl}, and *Dhx9*^{ΔIEC}*Sting*^{-/-} IECs. Equal amounts of DNA were spotted onto a nitrocellulose membrane, and R-loops were detected using the S9.6 antibody. Methylene blue (MB) staining served as the loading control. Quantifications are provided on the right. All data are presented as mean ± s.e.m. Statistical analysis was performed using a two-tailed unpaired Student's t test. *P < 0.05, **P < 0.01, ***P < 0.001, ****P < 0.0001. ns, not significant.

7. The model in S6 shows microbes penetrating the DHX9 KO, however there is no evidence to support this idea.

Response: We apologize for the error. The figure has been revised.

8. The authors should comment on their recent Science Advances paper that suggests DHX9 is important for transcription of ISGs and the alternative explanation that is now presented in this work. How do they think these mechanisms are both at play in the intestine?

Response: Thank you for your question. As we understand, DHX9 plays a crucial role in various physiological processes such as RNA processing, transcription, and translation. Its role in transcriptional mechanisms has been widely reported. For example, DHX9 links RNA Polymerase II to BRCA1 to initiate transcription and directly links p65 and RNA Polymerase II, enhancing NF- κ B-dependent transcriptional activation. Our previous research demonstrated that in macrophages, under viral infection and interferon stimulation, DHX9 directly binds to the ISG promoter region to participate in STAT1-mediated transcription of ISGs. However, it is noteworthy that in the absence of interferon or viral stimulation, the ISG levels in DHX9 KO cells do not decrease but are slightly higher than in DHX9 WT cells. This observation is consistent with what we have seen in IECs.

We think the reason for this is that in immune cells under infection and significant interferon stimulation, there is a need for rapid and substantial expression of ISGs (increases by nearly a hundredfold). In such scenarios, many transcriptional co-factors, including DHX9, are required for ISG transcription, thereby playing a role in positively regulating ISGs. However, under steady-state or low-level stimulation conditions, ISG expression levels are relatively lower compared to the antiviral response, and DHX9 may not be necessary for transcriptional co-activation. Therefore, knocking out DHX9 under steady-state conditions can lead to the accumulation of R-loops and genomic instability, both of which activate the cGAS-STING-dependent interferon pathway, resulting in the observed increase in ISG production. Hence, these findings are not contradictory.

Furthermore, the activation of the cGAS-STING-dependent interferon pathway in IECs following DHX9 knockout is partially due to the specific loss of DHX9 in IECs leading to cell damage and the generation of free DNA, cGAMP, etc., which activate neighboring myeloid cells. The DHX9 in these immune cells remains unaffected.

We have addressed this issue in the discussion section of our paper (**line 528-537**).

9. Statistics and quantification of results should be based upon biological replicates rather than technical counts, such as in figure 2F-I, 3C, S4, and elsewhere.

Response: Thank you for your reminder about the importance of basing statistics and quantification of results on biological replicates. We have clearly indicated the number of biological replicates in the legends.

10. The quantitation graphs in Fig. 5d-e don't match the data shown in the same figure panels.

Response: We apologize for the error. The figure has been revised.

Reviewer #3 (Remarks to the Author):

Ren et al. described the role of the RNA helicase DHX9 in the maintenance of intestinal epithelium. They showed that loss of DHX9 is associated with increased inflammation, reduced proliferation and differentiation of intestinal stem cells. Loss of DHX9 also promotes tumorigenesis in the mouse intestine. The findings are interesting and relevant to the clinics giving that reduced DHX9 is also found in human ulcerative colitis and Crohn's disease samples. That said, the authors may need to address a few issues as following.

Response: We want to express our sincere gratitude to you for providing such a thorough and positive summary of our work. Your thoughtful review has been incredibly insightful, and we deeply appreciate the time and effort you've dedicated to evaluating our manuscript. We have carefully considered the points and revised the manuscript accordingly.

1. Although the authors used Lgr5-CreER to conditionally delete DHX9 in the adult mice and showed that ISCs are affected, the majority of the experiments are done with the deletion mice using villin-cre (e.g. Fig5d). Use of Vil-cre to delete DHX9 can lead to developmental phenotypes which may have little relevance to ISCs. Therefore, the authors are suggested to analyze phenotypes using the Lgr5-CreER mouse line, for example, examination of R-Loop levels, cGAS-STING activation in the ISCs.

Response: Thank you for your valuable suggestion. In response to your advice, we have conducted further experiments in *Dhx9^{iΔISC}* mice to examine R-loop levels. we extracted DNA from the intestinal crypts of *c* and *Dhx9^{fl/fl}* mice induced with tamoxifen to assess the levels of R-loops. Dot blot analysis revealed a significant increase in the abundance of RNase H1-sensitive R-loops in *Dhx9^{iΔISC}* crypts (**Supplementary Fig. 15**). This is consistent with our findings in *Dhx9^{ΔIEC}* mice.

Additionally, we investigated the activation of the cGAS-STING pathway in *Dhx9^{iΔISC}* mice, a point also raised by Reviewer #2. We performed a time-course study post-tamoxifen treatment in *Dhx9^{iΔISC}* mice to detect changes in cGAS-Sting dependent IFN pathway activation, and alterations in ISCs and Paneth cells. Our results indicated that tamoxifen injection led to aberrant activation of the IFN pathway by day three, accompanied by a significant decrease in the numbers of Paneth cells and ISCs, consistent with observations in *Dhx9^{ΔIEC}* mice (**Supplementary Fig. 18**).

It is important to note that the *Dhx9*-deficient Lgr5+ ISCs are dynamically changing. In *Dhx9^{iΔISC}* mice, most of the IEC subpopulations turn into *Dhx9* knockouts within five days post-tamoxifen injection. This is due to the 3-5 day renewal cycle of IECs. As the effect of tamoxifen progresses, DHX9-deficient ISCs gradually differentiate into other IEC subgroups (**Supplementary Fig. 11**). Thus, the tamoxifen-treated *Dhx9^{iΔISC}* mice in later stages become quite similar with *Dhx9^{ΔIEC}* mice. Furthermore, not all crypts in *Dhx9^{iΔISC}* mice express Cre (marked by EGFP, approximately 40%), which is a limitation of this mouse model. Therefore, using Vil-cre for epithelial system knockout is more comprehensive, including the ISC population. This is why most experiments were conducted

with *Dhx9^{ΔIEC}* mice.

The new data can be found in **Supplementary Fig. 15**, **Supplementary Fig. 18**, and **Supplementary Fig. 11**, and the description of the results are added in the text **line 349-351**, **line 403-409**.

Supplementary Fig. 15

Figure S15. R-Loop quantification in ileum crypts following DHX9 knockdown.

Dot blot analysis conducted to quantify R-loops in ileum crypts of *Dhx9^{ΔISC}* and *Dhx9^{fl/fl}* mice. These mice were treated with tamoxifen for 6 days at 8 weeks of age to induce specific DHX9 knockdown. Consistent amounts of DNA from each sample were applied onto a nitrocellulose membrane. R-loops were detected using the S9.6 antibody, with RNase H treatment serving as a negative control. Quantitative results are presented on the right. All data are presented as mean \pm s.e.m. Statistical analysis was performed using a two-tailed unpaired Student's t test. * $P < 0.05$, ** $P < 0.01$. ns, not significant.

Supplementary Fig. 18

Figure S18. time-course study post-tamoxifen treatment in *Dhx9^{ΔISC}* mice.

a, Schematic diagram illustrating the strategy of tamoxifen administration. *Dhx9^{ΔISC}* mice were injected with tamoxifen and subsequently sacrificed at 1, 2, and 3 days post-injection. **b**, RT-qPCR analysis the expression of *Lgr5*, *Ang4*, *Isg15*, and *Oas2* in IECs from *Dhx9^{ΔISC}* mice ($n = 3$) and their *Dhx9^{fl/fl}* littermates ($n = 3$) at 8 weeks of age. All data are presented as mean \pm s.e.m. Statistical analysis was performed using a two-tailed unpaired Student's t test. * $P < 0.05$, ** $P < 0.01$, *** $P < 0.001$. ns, not significant.

2. Regarding the phenotypes seen in villin-cre;*Dhx9*^{flox/flox}, the authors are suggested to provide a comprehensive analysis of intestinal phenotypes at different developmental stages. The goal is to provide information whether the intestinal epithelium is affected during early stage of development. When does the phenotypes start?

Response: Thank you for your suggestion regarding a comprehensive analysis of intestinal phenotypes at different developmental stages in *Dhx9*^{ΔIEC} mice. To ascertain when the phenotypes start, we selected mice at three distinct developmental stages: 3 days, 7 days, and 1 month after birth, along with their *Dhx9*^{fl/fl} controls. We then conducted RT-qPCR analysis on ileum tissue samples, focusing on genes that showed significant alterations in our previous RNAseq studies of *Dhx9*^{ΔIEC} mice, including *Defa22* and *Ang4* (which were notably decreased), and *Isg15* and *Oasl2* (which were significantly increased). In *Dhx9*^{ΔIEC} mice, a notable decrease in the expression of *Defa22* and *Ang4* in IECs was observed at three days post-birth. However, at this early stage, *Isg15* and *Oasl2* did not exhibit significant changes (**Supplementary Fig. 19**). By 30 days, there were substantial changes in *Defa22*, *Ang4*, and *Isg15*, suggesting that the loss of DHX9 impacts the mouse IECs from an early stage and that the activation of the IFN-Pathway progressively intensifies over time. Therefore, the effect of DHX9 deficiency on IECs is persistent. However, based on indicators like body weight, the absence of DHX9 in the intestines does not seem to affect the overall development of the mice under steady-state conditions. Additionally, in our subsequent studies, we utilized *Dhx9*^{ΔISC} mice, where the deletion of DHX9 in ISC is induced only in the presence of tamoxifen. This approach eliminates the potential impact of DHX9 deficiency on early mouse development.

The new data can be found in **Supplementary Fig. 119**, and the description of the results are added in the text **line 409-411**.

Supplementary Fig. 19

Figure S19. Age-related changes in IECs of *Dhx9*^{ΔIEC} mice.

RT-qPCR analysis assessing the expression of *Defa22*, *Ang4*, *Oasl2*, and *Isg15* in IECs from *Dhx9*^{ΔIEC} mice at different ages. The study included analysis of IECs at 3 days, 7 days, and 30 days after birth. Mice in the experimental group

(*Dhx9*^{ΔIEC}) had n = 3 for each time point, while their *Dhx9*^{fl/fl} littermates (control group) had n = 4 for each age group. All data are presented as mean ± s.e.m. Statistical analysis was performed using a two-tailed unpaired Student's t test. *P < 0.05, **P < 0.01, ***P < 0.001. ns, not significant.

3. The authors may want to compare the intestinal phenotypes seen in vill-cre;dhx9flox/flox and Lgr5-creER;Dhx9fl/fl.

Response: Thank you for your valuable suggestion. We have compared the differences between *Dhx9*^{iΔISC} mice and *Dhx9*^{ΔIEC} mice, from the principles of knockout to their phenotypic manifestations, and included these comparisons in our article.

We believe there are several key differences between the two:

Knockout Mechanism and Efficiency: The knockout in *Dhx9*^{ΔIEC} is mediated by Vill1, which is expressed in all epithelial cell subgroups, including ISCs. This leads to a more comprehensive and thorough knockout. In contrast, in *Dhx9*^{iΔISC} mice, the Cre-mediated knockout is driven by Lgr5, which is expressed only in ISCs and TA.G2 IEC subgroups, making the knockout more specific. However, it's important to note that not all crypts in *Lgr5-EGFP-IRES-cre*^{ERT2} mice express Lgr5-Cre (as demonstrated by our Tdtomato mice), leading to variability in knockout efficiency. Additionally, while Cre expression in *Dhx9*^{ΔIEC} mice is constitutive and may impact intestinal development, it is inducible in *Lgr5-EGFP-IRES-cre*^{ERT2} mice, avoiding this issue. Hence, using both mouse models in our study helps mitigate these issues and makes our results more comprehensive and reliable.

DSS-Induced Colitis Phenotype: When comparing *Dhx9*^{iΔISC} mice with *Dhx9*^{ΔIEC} mice, we observed that the latter exhibited a more pronounced weight loss trend and severe colitis phenotype. We attribute this to the higher and more comprehensive efficiency of DHX9 knockout in *Dhx9*^{ΔIEC} mice.

Impact on IEC Types: Similarly, the effect on IEC types was more pronounced in *Dhx9*^{ΔIEC} mice compared to *Dhx9*^{iΔISC} mice. ISC cell markers like *Lgr5* and Paneth cell markers like *Ang4* showed a more significant decrease in the IECs of *Dhx9*^{ΔIEC} mice.

These comparative analyses enhance our understanding of the specific roles and effects of DHX9 knockout in different intestinal cell types and contribute significantly to the robustness of our findings.

Figure. Comparison of phenotypes of $Dhx9^{\Delta IEC}$ and $Dhx9^{iAISC}$ mice.

a, Expression pattern of Lgr5 and Vil1 in IECs subpopulations. b, Phenotypes of DSS-induced colitis in $Dhx9^{\Delta IEC}$ and $Dhx9^{iAISC}$ mice. c, Impact on IEC types, demonstrating more pronounced changes in $Dhx9^{\Delta IEC}$ mice compared to $Dhx9^{iAISC}$.

Minor issue:

The authors are suggested to provide high magnification images of Fig5g where rH2ax and R-loop IHC was shown. It appears the staining is not specific, especially Rloop staining.

Response: Thanks for your suggestions. We have replaced the picture with a higher definition.

References

- 1 Huang, B. *et al.* Mucosal Profiling of Pediatric-Onset Colitis and IBD Reveals Common Pathogenics and Therapeutic Pathways. *Cell* **179**, 1160-1176 e1124, doi:10.1016/j.cell.2019.10.027 (2019).
- 2 Nozaki, K. *et al.* Caspase-7 activates ASM to repair gasdermin and perforin pores. *Nature* **606**, 960-967, doi:10.1038/s41586-022-04825-8 (2022).
- 3 He, K. *et al.* Gasdermin D licenses MHCII induction to maintain food tolerance in small intestine. *Cell* **186**, 3033-3048 e3020, doi:10.1016/j.cell.2023.05.027 (2023).
- 4 Liu, M. Y. *et al.* ATR phosphorylates DHX9 at serine 321 to suppress R-loop accumulation upon genotoxic stress. *Nucleic Acids Research*, doi:10.1093/nar/gkad973 (2023).
- 5 Cristini, A., Groh, M., Kristiansen, M. S. & Gromak, N. RNA/DNA Hybrid Interactome Identifies DXH9 as a Molecular Player in Transcriptional Termination and R-Loop-Associated DNA Damage. *Cell Rep* **23**, 1891-1905, doi:10.1016/j.celrep.2018.04.025 (2018).
- 6 Huang, N. *et al.* DHX9-mediated pathway contributes to the malignant phenotype of myelodysplastic syndromes. *iScience* **26**, 106962, doi:10.1016/j.isci.2023.106962 (2023).
- 7 Suzuki, M. M. *et al.* TUG1-mediated R-loop resolution at microsatellite loci as a prerequisite for cancer cell proliferation. *Nature Communications* **14**, doi:10.1038/s41467-023-40243-8 (2023).
- 8 Patel, P. S. *et al.* RNF168 regulates R-loop resolution and genomic stability in BRCA1/2-deficient tumors. *J Clin Invest* **131**, doi:10.1172/JCI140105 (2021).
- 9 Yuan, W. *et al.* TDRD3 promotes DHX9 chromatin recruitment and R-loop resolution. *Nucleic Acids Res* **49**, 8573-8591, doi:10.1093/nar/gkab642 (2021).
- 10 Wang, K. *et al.* Genomic profiling of native R loops with a DNA-RNA hybrid recognition sensor. *Sci Adv* **7**, doi:10.1126/sciadv.abe3516 (2021).
- 11 Wang, H., Li, C. & Liang, K. Genome-Wide Native R-Loop Profiling by R-Loop Cleavage Under Targets and Tagmentation (R-Loop CUT&Tag). *Methods Mol Biol* **2528**, 345-357, doi:10.1007/978-1-0716-2477-7_23 (2022).
- 12 Boguslawski, S. J. *et al.* Characterization of monoclonal antibody to DNA.RNA and its application to immunodetection of hybrids. *J Immunol Methods* **89**, 123-130, doi:10.1016/0022-1759(86)90040-2 (1986).
- 13 Phillips, D. D. *et al.* The sub-nanomolar binding of DNA-RNA hybrids by the single-chain Fv fragment of antibody S9.6. *J Mol Recognit* **26**, 376-381, doi:10.1002/jmr.2284 (2013).
- 14 Yang, S., Winstone, L., Mondal, S. & Wu, Y. Helicases in R-loop Formation and Resolution. *J Biol Chem* **299**, 105307, doi:10.1016/j.jbc.2023.105307 (2023).
- 15 Chakraborty, P. & Grosse, F. Human DHX9 helicase preferentially unwinds RNA-containing displacement loops (R-loops) and G-quadruplexes. *DNA Repair (Amst)* **10**, 654-665, doi:10.1016/j.dnarep.2011.04.013 (2011).
- 16 Chakraborty, P., Huang, J. T. J. & Hiom, K. DHX9 helicase promotes R-loop formation in cells with impaired RNA splicing. *Nat Commun* **9**, 4346, doi:10.1038/s41467-018-06677-1 (2018).
- 17 Hamperl, S., Bocek, M. J., Saldivar, J. C., Swigut, T. & Cimprich, K. A. Transcription-Replication Conflict Orientation Modulates R-Loop Levels and Activates Distinct DNA Damage Responses. *Cell* **170**, 774-786 e719, doi:10.1016/j.cell.2017.07.043 (2017).
- 18 Ngo, G. H. P., Grimstead, J. W. & Baird, D. M. UPF1 promotes the formation of R loops to stimulate DNA double-strand break repair. *Nat Commun* **12**, 3849, doi:10.1038/s41467-021-24201-w (2021).
- 19 Marnef, A. & Legube, G. R-loops as Janus-faced modulators of DNA repair. *Nat Cell Biol* **23**, 305-313, doi:10.1038/s41556-021-00663-4 (2021).

- 20 Skourti-Stathaki, K. & Proudfoot, N. J. A double-edged sword: R loops as threats to genome integrity and powerful regulators of gene expression. *Genes Dev* **28**, 1384-1396, doi:10.1101/gad.242990.114 (2014).
- 21 Jain, A. *et al.* DHX9 helicase is involved in preventing genomic instability induced by alternatively structured DNA in human cells. *Nucleic Acids Research* **41**, 10345-10357, doi:10.1093/nar/gkt804 (2013).
- 22 Lee, T. *et al.* Suppression of the DHX9 helicase induces premature senescence in human diploid fibroblasts in a p53-dependent manner. *J Biol Chem* **289**, 22798-22814, doi:10.1074/jbc.M114.568535 (2014).
- 23 Chakraborty, P. & Hiom, K. DHX9-dependent recruitment of BRCA1 to RNA promotes DNA end resection in homologous recombination. *Nature Communications* **12**, doi:ARTN 4126 10.1038/s41467-021-24341-z (2021).
- 24 Zhang, S., Schlott, B., Gorch, M. & Grosse, F. DNA-dependent protein kinase (DNA-PK) phosphorylates nuclear DNA helicase II/RNA helicase A and hnRNP proteins in an RNA-dependent manner. *Nucleic Acids Res* **32**, 1-10, doi:10.1093/nar/gkg933 (2004).
- 25 Mischo, H. E., Hemmerich, P., Grosse, F. & Zhang, S. Actinomycin D induces histone gamma-H2AX foci and complex formation of gamma-H2AX with Ku70 and nuclear DNA helicase II. *J Biol Chem* **280**, 9586-9594, doi:10.1074/jbc.M411444200 (2005).
- 26 Liu, J. *et al.* Functional proteomic analysis of promyelocytic leukaemia nuclear bodies in irradiation-induced MCF-7 cells. *J Biochem* **148**, 659-667, doi:10.1093/jb/mvq105 (2010).
- 27 Gorthi, A. *et al.* EWS-FLI1 increases transcription to cause R-loops and block BRCA1 repair in Ewing sarcoma. *Nature* **555**, 387-391, doi:10.1038/nature25748 (2018).
- 28 Karyka, E. *et al.* SMN-deficient cells exhibit increased ribosomal DNA damage. *Life Sci Alliance* **5**, doi:10.26508/lsa.202101145 (2022).
- 29 Grunseich, C. *et al.* Senataxin Mutation Reveals How R-Loops Promote Transcription by Blocking DNA Methylation at Gene Promoters. *Mol Cell* **69**, 426-437.e427, doi:10.1016/j.molcel.2017.12.030 (2018).
- 30 Chen, C. *et al.* Atrial Natriuretic Peptide Attenuates Colitis via Inhibition of the cGAS-STING Pathway in Colonic Epithelial Cells. *Int J Biol Sci* **18**, 1737-1754, doi:10.7150/ijbs.67356 (2022).
- 31 Shmuel-Galia, L. *et al.* Dysbiosis exacerbates colitis by promoting ubiquitination and accumulation of the innate immune adaptor STING in myeloid cells. *Immunity* **54**, 1137-1153 e1138, doi:10.1016/j.immuni.2021.05.008 (2021).
- 32 Flood, P. *et al.* DNA sensor-associated type I interferon signaling is increased in ulcerative colitis and induces JAK-dependent inflammatory cell death in colonic organoids. *American journal of physiology. Gastrointestinal and liver physiology* **323**, G439-g460, doi:10.1152/ajpgi.00104.2022 (2022).
- 33 Zhao, F. *et al.* Extracellular vesicles package dsDNA to aggravate Crohn's disease by activating the STING pathway. *Cell Death Dis* **12**, 815, doi:10.1038/s41419-021-04101-z (2021).
- 34 Aden, K. *et al.* ATG16L1 orchestrates interleukin-22 signaling in the intestinal epithelium via cGAS-STING. *J Exp Med* **215**, 2868-2886, doi:10.1084/jem.20171029 (2018).
- 35 Ahn, J., Son, S., Oliveira, S. C. & Barber, G. N. STING-Dependent Signaling Underlies IL-10 Controlled Inflammatory Colitis. *Cell Rep* **21**, 3873-3884, doi:10.1016/j.celrep.2017.11.101 (2017).
- 36 Malireddi, R. K. & Kanneganti, T. D. Role of type I interferons in inflammasome activation, cell death, and disease during microbial infection. *Front Cell Infect Microbiol* **3**, 77, doi:10.3389/fcimb.2013.00077 (2013).

- 37 Terawaki, S. *et al.* IFN-alpha directly promotes programmed cell death-1 transcription and limits the duration of T cell-mediated immunity. *J Immunol* **186**, 2772-2779, doi:10.4049/jimmunol.1003208 (2011).
- 38 Hu, Q. *et al.* The emerging role of stimulator of interferons genes signaling in sepsis: Inflammation, autophagy, and cell death. *Acta Physiol (Oxf)* **225**, e13194, doi:10.1111/apha.13194 (2019).
- 39 Fuertes Marraco, S. A. *et al.* Type I interferon drives dendritic cell apoptosis via multiple BH3-only proteins following activation by PolyIC in vivo. *PLoS One* **6**, e20189, doi:10.1371/journal.pone.0020189 (2011).
- 40 Cohen, T. S. & Prince, A. S. Activation of inflammasome signaling mediates pathology of acute P. aeruginosa pneumonia. *J Clin Invest* **123**, 1630-1637, doi:10.1172/JCI66142 (2013).
- 41 Papageorgiou, A., Dinney, C. P. & McConkey, D. J. Interferon-alpha induces TRAIL expression and cell death via an IRF-1-dependent mechanism in human bladder cancer cells. *Cancer Biol Ther* **6**, 872-879, doi:10.4161/cbt.6.6.4088 (2007).
- 42 Chawla-Sarkar, M. *et al.* Apoptosis and interferons: role of interferon-stimulated genes as mediators of apoptosis. *Apoptosis : an international journal on programmed cell death* **8**, 237-249, doi:10.1023/a:1023668705040 (2003).
- 43 Tomic, J., Lichty, B. & Spaner, D. E. Aberrant interferon-signaling is associated with aggressive chronic lymphocytic leukemia. *Blood* **117**, 2668-2680, doi:10.1182/blood-2010-05-285999 (2011).
- 44 Glennon-Alty, L., Moots, R. J., Edwards, S. W. & Wright, H. L. Type I interferon regulates cytokine-delayed neutrophil apoptosis, reactive oxygen species production and chemokine expression. *Clinical and experimental immunology* **203**, 151-159, doi:10.1111/cei.13525 (2021).
- 45 Klein, B. & Gunther, C. Type I Interferon Induction in Cutaneous DNA Damage Syndromes. *Front Immunol* **12**, 715723, doi:10.3389/fimmu.2021.715723 (2021).
- 46 Sangfelt, O., Erickson, S. & Grander, D. Mechanisms of interferon-induced cell cycle arrest. *Front Biosci* **5**, D479-487, doi:10.2741/sangfelt (2000).
- 47 Wang, R. *et al.* Gut stem cell necroptosis by genome instability triggers bowel inflammation. *Nature* **580**, 386-390, doi:10.1038/s41586-020-2127-x (2020).
- 48 Cuevas, M. E. & Lindeman, T. E. In vitro cytotoxicity of 4'-OH-tamoxifen and estradiol in human endometrial adenocarcinoma cells HEC-1A and HEC-1B. *Oncol Rep* **33**, 464-470, doi:10.3892/or.2014.3565 (2015).
- 49 Bopp, S. K. & Lettieri, T. Comparison of four different colorimetric and fluorometric cytotoxicity assays in a zebrafish liver cell line. *BMC Pharmacol* **8**, 8, doi:10.1186/1471-2210-8-8 (2008).

REVIEWER COMMENTS

Reviewer #1 (Remarks to the Author):

In the revised manuscript, the authors performed additional experiments to address the reviewer's critiques. These new results provide further support to the working model. I appreciate these efforts and most of my comments have been addressed. My remaining concerns are related to the new R-loop Cut&Tag data, which is quite confusing.

1. In Supp Fig. 16a, the results from heatmap analysis (bottom panels) do not seem to correlate with the cumulative plots (top panels). From the heatmap, there are clear R-loop peaks at TSSs, but they are not obvious in the cumulative plots. Also, based on the heatmap results, the C&T signal seems stronger in the Dhx9 flox sample in comparison to the KO. But the cumulative plots show opposite results.
2. What does the R-loop C&T signal/distribution look like on an annotated gene? The dot blot results suggest a global increase of R-loops upon Dhx9 KO. What are the regions that gained and/or lost R-loops?
3. It is unclear when the author stated that "it is important to note that the significantly changed genes in R-loops do not overlap with those in RNA-seq". Does this mean that the genes with differential R-loop levels do not overlap with genes that show differential expression? They should include the analyses to support this claim.
4. A few exemplary loci should be shown for R-loop peaks. They are expected to be diminished with RNase H treatment control.

Reviewer #2 (Remarks to the Author):

The authors did substantial work in improving the manuscript and I am largely convinced that the rigor of the work has been enhanced and merits publication. My one concern that wasn't sufficiently addressed is that the Lgr5Cre targets recombination sufficiently in the colon to be used in the DSS model. In response to this critique, the authors showed recombination efficiency in the duodenum, where the Cre is most robust, showing results consistent with published studies. However, they do not show a zoom out of colon crypts labeled with the Cre reporter. I recommend the authors either show extensive recombination in the colon, or remove the Lgr5-Cre based studies in the DSS experiments in favor of using the Villin-Cre DSS studies, which will be a more rigorous model.

Minor comment: There appears to be a notable shift in the enterocyte transcriptome in the new scRNAseq data. It would be useful for the authors to compare gene expression in this population (clusters 8-9).

Reviewer #3 (Remarks to the Author):

The authors have addressed this reviewer's concerns.

Response to reviewers

We thank the reviewers for their constructive comments and valuable suggestions. Over the past month, we have obtained more experimental evidence to support our conclusion in accordance with the reviewers' suggestions. Our point-by-point responses to the reviewers' comments are listed below. Text changes are highlighted with **yellow background** in our revised manuscript.

Reviewer #1 (Remarks to the Author):

In the revised manuscript, the authors performed additional experiments to address the reviewer's critiques. These new results provide further support to the working model. I appreciate these efforts and most of my comments have been addressed. My remaining concerns are related to the new R-loop Cut&Tag data, which is quite confusing.

Thank you for the constructive suggestions. We have re-performed the Cut&Tag experiment, specifically, included an RNase H control group to further support the conclusion.

1. In Supp Fig. 16a, the results from heatmap analysis (bottom panels) do not seem to correlate with the cumulative plots (top panels). From the heatmap, there are clear R-loop peaks at TSSs, but they are not obvious in the cumulative plots. Also, based on the heatmap results, the C&T signal seems stronger in the Dhx9 flox sample in comparison to the KO. But the cumulative plots show opposite results.

Response: Thank you for raising a critical point. We apologize for any confusion caused. The inconsistency arises because the cumulative plots (top panels) data were normalized to IgG, while the heatmap (bottom panels) represents results from the entire enriched reads. Additionally, there was an offset in our cumulative plots results, and the peak of wave positions should indeed be centered around the TSS region. We genuinely appreciate your reminder. To address this, we have unified the approach in presenting signals consistently in the new data to ensure coherence between plotProfile (top panels) and TSS heatmap (bottom panels).

Notably, in response to your suggestion in question 4 regarding the RNase H treatment control group, we have re-conducted the Cut&Tag experiment and included the *Dhx9*^{ΔIEC} + RNase H control group. The results demonstrate a significantly stronger R-loop signal in the *Dhx9*^{ΔIEC} group compared to the *Dhx9*^{fl/fl} control. Moreover, RNase H treatment attenuated the R-loop signal in the *Dhx9*^{ΔIEC} samples. Thanks to your suggestion, these new results provide a more comprehensive understanding of our results (**Supplementary Fig. 16a**).

The new data can be found in **Supplementary Fig. 16**, and the description of the results are added in the text **line 357-359**.

Figure S16. Genomic distribution of R-loop peaks in *Dhx9*^{ΔIEC} IECs.

a, Heatmaps depicting the CUT&Tag-seq signal distribution around the transcription start sites (TSS) within a 5 kb range, both upstream and downstream. Signals are shown for S9.6 antibody in *Dhx9*^{fl/fl} IECs, S9.6 antibody in *Dhx9*^{ΔIEC} IECs, S9.6 antibody in *Dhx9*^{ΔIEC} IECs that treated with RNase H, and IgG control in *Dhx9*^{fl/fl} IECs.

2. What does the R-loop C&T signal/distribution look like on an annotated gene? The dot blot results suggest a global increase of R-loops upon *Dhx9* KO. What are the regions that gained and/or lost R-loops?

Response: Thank you for your question. We appreciate your inquiry into the R-loop C&T signal/distribution on annotated genes. Our finding demonstrated a global increase in R-loop signals (peaks) across the entire genome in the *Dhx9*^{ΔIEC} IECs compared to *Dhx9*^{fl/fl} (5524 vs 1481), encompassing promoter, exon, intron, and intergenic regions. In response to your query regarding the regions that gained and/or lost R-loops, we have included additional statistical results detailing the numbers of R-loop peaks in promoter, exon, intron, and intergenic regions. The analysis indicates a comprehensive increase in R-loop peaks in each region, with a particularly notable rise in the promoter regions of the *Dhx9*^{ΔIEC} group (**Supplementary Fig. 16b**).

The new data can be found in **Supplementary Fig. 16b**, and the description of the results are added in the **text line 359-364**.

b
Figure S16. Genomic distribution of R-loop peaks in *Dhx9^{ΔIEC}* IECs.

b. Genomic distribution variation of R-loop peaks in *Dhx9^{fl/fl}* and *Dhx9^{ΔIEC}* IECs, as revealed by CUT&Tag-Seq data. The annotations highlight the distribution of these peaks across different genomic regions.

3. It is unclear when the author stated that “it is important to note that the significantly changed genes in R-loops do not overlap with those in RNA-seq”. Does this mean that the genes with differential R-loop levels do not overlap with genes that show differential expression? They should include the analyses to support this claim.

Response: We apologize for the possible confusion. Our intention was to convey that the genes showing significant alterations in the RNA-seq analysis of DHX9-deficient IECs are not directly regulated by R-loops. The changes in gene expression result from the overall increase in R-loops, leading to genomic DNA damage.

To support this assertion, we conducted an intersection analysis between genes associated with R-loop signals and those significantly upregulated or downregulated in RNA-seq analysis of DHX9-deficient IECs. The results indicate that approximately 1/10 of the genes exhibiting significant changes in RNA-seq analysis display R-loop signals. This proportion aligns with the overall percentage (approximately 10%) of genes in the genome that harbor R-loop signals (Our study identified around 3,000 genes with R-loop-enriched signals in DHX9-deficient IECs, constituting about 1/10 of the total protein-coding genes in mice (approximately 30,000)). We hope this clarification addresses your concern (**Supplementary Fig. 16d**).

The new data can be found in **Supplementary Fig. 16**, and the description of the results are added in the text **line 364-371**.

Figure S16. Genomic distribution of R-loop peaks in *Dhx9*^{ΔIEC} IECs.

d, Venn diagrams showing the overlap of R-loop accumulated genes and transcriptionally upregulated or downregulated genes in RNA-seq analysis of DHX9-deficient IECs.

4. A few exemplary loci should be shown for R-loop peaks. They are expected to be diminished with RNase H treatment control.

Response: Thank you for your suggestion. We have provided representative R-loop signals at selected genomic loci to illustrate the varied distribution of signals across different gene regions. Specifically, *Vil*, exhibits R-loop peaks at multiple regions (promoter, exon, and intron); *Tef*, R-Loop peaks at promoter regions; *Peak1*, R-Loop peaks at intron regions; *Lars2*, R-Loop peaks at exon regions; *Fus*, an R-loop signal in the promoter region was previously reported. Notably, R-loop signals in the *Dhx9*^{ΔIEC} group are more pronounced than those in the *Dhx9*^{fl/fl} group. In response to your concern, we performed RNase H treatment as a control, revealing a significant reduction in R-loop signals (**Supplementary Fig. 16c**). These observations support the conclusion that DHX9 depletion induces increased R-loop formation.

The new data can be found in **Supplementary Fig. 16**, and the description of the results are added in the text **line 362-364**.

Figure S16. Genomic distribution of R-loop peaks in *Dhx9*^{ΔIEC} IECs.

c, UCSC genome browser tracks of CUT&Tag signals at the *Vil*, *Tef*, *Peak1*, *Peak1*, *Lars2*, and *Fus* loci.

Reviewer #2 (Remarks to the Author):

The authors did substantial work in improving the manuscript and I am largely convinced that the rigor of the work has been enhanced and merits publication. My one concern that wasn't sufficiently addressed is that the *Lgr5*Cre targets recombination sufficiently in the colon to be used in the DSS model. In response to this critique, the authors showed recombination efficiency in the duodenum, where the Cre is most robust, showing results consistent with published studies. However, they do not show a zoom out of colon crypts labeled with the Cre reporter. I recommend the authors either show extensive recombination in the colon, or remove the *Lgr5*-Cre based studies in the DSS experiments in favor of using the Villin-Cre DSS studies, which will be a more rigorous model.

Response: Thank you for your constructive suggestions. In response to your concern regarding *Lgr5*-Cre recombination efficiency in the colon, we have conducted a dedicated imaging of colon from *Lgr5-EGFP-Cre^{ERT2}; Rosa26^{lsl-tdTomato}* mice. The results demonstrate that the *Lgr5*-Cre efficiency in the colon is comparable to that observed in the small intestine. The expression is higher at the front end, and the overall efficiency level is approximately 40%. We wholeheartedly agree with your assessment. The Villin-Cre DSS studies indeed represent a more rigorous model, and the *Lgr5*-Cre-based studies in the DSS experiments serve as supplementary data.

The new data can be found in **Supplementary Fig. 11**, and the description of the results are added in the text **line 279-286**.

Figure S11. The knockout efficiency of *Lgr5-EGFP-Cre^{ERT2}* was tracked by tdTomato fluorescence.

c, Representative image of whole intestine and colon from *Lgr5-EGFP-Cre^{ERT2}; Rosa26^{lsl-tdTomato}* mice, which were treated with tamoxifen for 6 days at 8 weeks of age. tdTomato are derived from endogenous fluorescence. Nuclei are stained with DAPI (blue). n = 4 per group.

Minor comment: There appears to be a notable shift in the enterocyte transcriptome in the new scRNAseq data. It would be useful for the authors to compare gene expression in this population (clusters 8-9).

Response: Thank you for your valuable suggestion. We have acknowledged the observed shift in the enterocyte transcriptome resulting from DHX9 depletion, signifying a deviation in the epithelial differentiation trajectory. Your suggestion to compare gene expression in clusters 8-9 is appreciated, and we will conduct further analyses to gain a more comprehensive understanding of the functional implications of DHX9 in maintaining epithelial cell homeostasis. We anticipate that these additional investigations will provide valuable insights into the specific molecular mechanisms driving the changes in the enterocyte population.

Reviewer #3 (Remarks to the Author):

The authors have addressed this reviewer's concerns.

Thank you once more for your invaluable feedback and guidance during the review of our manuscript.